# Benchmarking Distribution Shift in Tabular Data with TableShift

**Josh Gardner**[♮]  **Zoran Popović**[♮]  **Ludwig Schmidt**[♮,♭]

[♮] **University of Washington**  [♭] **Allen Institute for AI**
`{jpgard, zoran, schmidt}@cs.washington.edu`

## Abstract

Robustness to distribution shift has become a growing concern for text and image models as they transition from research subjects to deployment in the real world. However, high-quality benchmarks for distribution shift in *tabular* machine learning tasks are still lacking despite the widespread real-world use of tabular data and differences in the models used for tabular data in comparison to text and images. As a consequence, the robustness of tabular models to distribution shift is poorly understood. To address this issue, we introduce TABLESHIFT, a distribution shift benchmark for tabular data. TABLESHIFT contains 15 binary classification tasks in total, each with an associated shift, and includes a diverse set of data sources, prediction targets, and distribution shifts. The benchmark covers domains including finance, education, public policy, healthcare, and civic participation, and is accessible using only a few lines of Python code via the TABLESHIFT API. We conduct a large-scale study comparing several state-of-the-art tabular data models alongside robust learning and domain generalization methods on the benchmark tasks. Our study demonstrates (1) a linear trend between in-distribution (ID) and out-of-distribution (OOD) accuracy; (2) domain robustness methods can reduce shift gaps but at the cost of reduced ID accuracy; (3) a strong relationship between shift gap (difference between ID and OOD performance) and shifts in the label distribution.[1]

## 1 Introduction

Modern machine learning models have achieved near- or even super-human performance on many tasks. This has contributed to deployments of models across critical domains, including finance, public policy, and healthcare. However, in tandem with the growing deployment of machine learning models, researchers have also demonstrated concerning model performance drops under *distribution shift* – when the test/deployment data are not drawn from the same distribution as the training data. Analyses of these performance drops have primarily been confined to the domains of vision and language modeling (e.g. [38, 50], where effective benchmarks for distribution shift exist. Despite the widespread use of tabular data, the impact of distribution shift on *tabular* data has not been thoroughly investigated. While there are existing benchmarks for IID tabular classification, none of these focus on distribution shifts [36, 33].

This is particularly concerning in light of the known differences between tabular data and the modalities mentioned above (images, text, audio). First, in contrast to these modalities, where large neural models are the undisputed state-of-the-art, there is considerable debate about whether deep

---

[1]The benchmark data, Python package, model implementations, and more information about TABLESHIFT are available at `https://github.com/mlfoundations/tableshift` and `https://tableshift.org`.

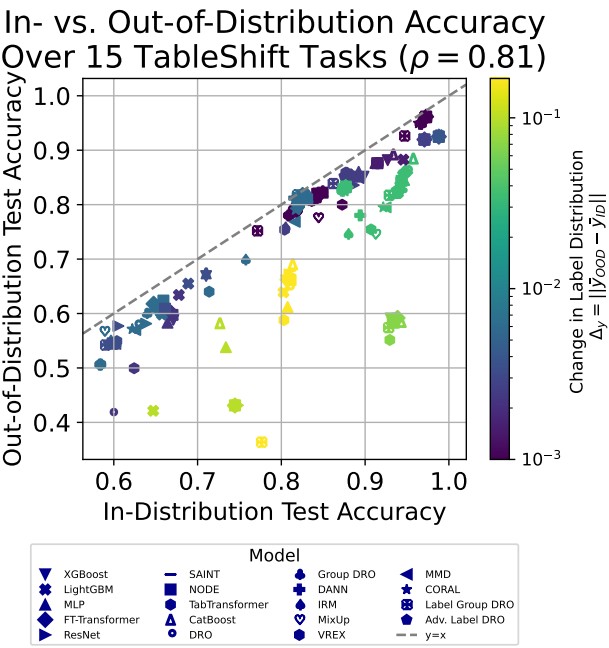

Figure 1: In-domain (ID) and out-of-domain (OOD) accuracy show a linear trend across 15 TableShift tasks and 19 model types ($\rho = 0.81$). ID accuracy ($x$-axis values) and change in the label distribution $\Delta_y$ (color) together explain 99% of the variance in OOD accuracy ($R^2 = 0.993$). For exact results see Section E.3.

learning models improve performance over non-neural baselines (e.g. XGBoost, LightGBM) for tabular data, even without the presence of distribution shift [46, 15, 78]. Second, tabular data tends to contain structured features extracted from raw data (e.g. counts of activities, coded responses to questions), as opposed to the raw signals (e.g. activity event streams, pixel values, audio of responses) where modern machine learning methods perform well and where previous studies of distribution shift have focused. Third, tabular data requires fundamentally different preprocessing procedures, and the impact of these decisions is not widely understood, despite being known to have empirical impact [31]. Finally, high-quality tabular datasets can be difficult to access [35]; for example, due to the personal nature of many tabular datasets, tabular data cannot simply be scraped at Internet scale as many text and image datasets are. This makes finding high-quality tabular *distribution shift* datasets particularly challenging.

Thus, the machine learning research community currently lacks not only (1) an empirical understanding of the impact of distribution shift on tabular data models, but also (2) a shared set of accessible and high-quality benchmarks to enable such investigations. We address both gaps in this work. Our main contributions are:

**TableShift Benchmark:** we introduce a curated, high-quality set of publicly-accessible benchmark tasks for (binary) tabular data classification under distribution shift. We describe the tasks in §3.1 and the API in §3.2. TableShift includes a set of real-world tabular datasets from domains including finance [30], public policy [24], healthcare [19, 47, 18, 74, 87], and civic participation [5]. We select these datasets to ensure a diversity of tasks, distribution shifts, and dataset sizes.

**Large-scale empirical study of distribution shift in tabular data:** We conduct a large-scale study in §4, including state-of-the-art tree-based tabular models, tabular neural networks, distributional robustness methods, domain generalization methods, and label shift robustness methods. Our findings show (1) a strong linear trend between in-distribution (ID) and out-of-distribution (OOD) accuracy across benchmark tasks and models that was not previously identified for tabular data; (2) that no model consistently outperforms baseline methods, and (3) a correlation between the shift gap and the shift in label distribution, which is not ameliorated by label shift robustness methods included in our study.

**Accessible TableShift API and baselines:** We release a Python API for constructing rich datasets directly from their raw public forms. The API provides built-in documentation of data types and feature codings, alongside standardized preprocessing and transformation pipelines, making the datasets accessible in multiple formats suitable for training tabular models (e.g. in scikit-learn and PyTorch). We also release the set of baseline model implementations (including both state of the art tabular data models, robust learning models, and domain generalization methods) and end-to-end training code in order to facilitate future research on distribution shift in tabular data.

## 2   Setup, Task, and Notation

### 2.1   Task and Setting

Consider a dataset composed of examples $(x, y, d) \sim P_d$ where $x$ is the input, $y$ is the prediction target, and $d$ the domain from which that example is drawn. All examples drawn from $P_d$ have domain label $d$. We can view the overall data distribution as a mixture of domains $\mathcal{D} = \{d_1, \ldots, d_D\}$, where $D \geq 2$. Training examples are drawn from the training distribution $P^{\text{train}} = \sum_{d \in D} q_d^{\text{train}} P_d$, and testing examples from $P^{\text{test}} = \sum_{d \in D} q_d^{\text{test}} P_d$, with domain weights $q_d \in [0, 1]$. We can define the training and testing domains as $\mathcal{D}^{\text{train}} = \{d \in \mathcal{D} : q_d^{\text{train}} > 0\}$ and $\mathcal{D}^{\text{test}} = d \in \mathcal{D} : q_d^{\text{test}} > 0$, respectively. We refer to cases where $|\mathcal{D}^{\text{train}}| \geq 2$ as "domain generalization" tasks, because domain generalization models require at least two subdomains in the training data.

In a standard (IID) setting, our goal is to learn a classifier $f_\theta$ that accurately predicts $y$ using examples from $\mathcal{D}^{\text{train}}$. A *distribution shift* (or *domain shift*) occurs due to the fact that $P^{\text{train}} \neq P^{\text{test}}$. As a consequence of this shift, the joint distributions $p^{\text{train}}(x, y) \neq p^{\text{test}}(x, y)$ differ in training and testing. This difference can be composed of one or more changes to the underlying data generating process. This includes covariate shift, where $p(x)$ changes; label shift, where $p(y)$ changes, and concept shift, where $p(y|x)$ changes. In almost all real-world scenarios, distribution shifts are composed of an unknown mixture of all three forms of shift[2]. For a fixed classifier $f_\theta$, we refer to

$$\Delta_{\text{Acc}} = \text{Acc}(f_\theta, \mathcal{D}^{\text{test}}) - \text{Acc}(f_\theta, \mathcal{D}^{\text{train}}) \tag{1}$$

as the "shift gap" (where both metrics are computed on examples not seen at training time). Note that the shift gap can be affected by changes in $p(y)$, $p(y|x)$, and $p(x)$. While disentangling the effects of these forms of shift is not a focus of the current work, we provide initial exploratory results on the impact of changes in $p(y)$, $p(y|x)$, and $p(x)$ over the benchmark tasks in Sections 5 and E.

In our setting, we assume that no information about the target $\mathcal{D}^{\text{test}}$ is available – i.e., there is no knowledge of the change in $p(y)$, $p(y|x)$, and $p(x)$, and no unlabeled data from the target domain.

### 2.2   Related Work

Here we provide a brief overview of related work necessary to contextualize our benchmark and main results. For a detailed overview of related work, see Section D.

Our work is closely related to the literature on distribution shift in machine learning. A series of recent works have demonstrated that even state-of-the-art models experience significant performance drops under distribution shift in tasks including vision, language modeling, and question answering [61, 62, 38, 50, 9]. This has led to the development of methods to mitigate susceptibility to such shifts [76, 53, 1, 6, 90, 89, 54, 46]. High-quality benchmarks, specifically tailored to distribution shift, have been essential in both measuring these gaps and assessing progress toward closing them [38, 50]. The use of tabular data is widespread in practice [15, 46, 78], including the use of sensitive personal data (race, gender, age) and for important tasks (credit scoring, medical diagnosis). However, the impact of distribution shift in the tabular domain has received little attention in the research literature. In particular, benchmarks containing *tabular* distribution shifts are lacking (one notable exception is Shifts [57] and Shifts 2.0 [57], a multimodal benchmark of five tasks, two of which are tabular; for a more detailed overview of domain shift benchmarks and a comparison to TABLESHIFT, see Section G).

---

[2]We note that this is a slight abuse of the terminology, as e.g. "label shift" typically refers to the case where *only* $p(y)$ changes.

# 3 Tableshift: A Distribution Shift Benchmark for Tabular Data

This work introduces the TABLESHIFT benchmark. TABLESHIFT contains a set of 15 curated tasks designed to be a rigorous, challenging, diverse, and reliable benchmarking suite for tabular data under distribution shift, and we encapsulate them within a Python API.

## 3.1 TableShift Benchmark Tasks

To select tasks for TABLESHIFT, we identified datasets meeting the following formal criteria:

**Open source:** datasets must be publicly available, including data dictionaries documenting the source of the data (i.e. conditions of its collection), definitions of variables, and any preprocessing applied.

**Real-world:** does not contain simulated data.

Table 1: Summary of TABLESHIFT tasks and their associated distribution shifts. For details on each task, see Section E. "Domain Generalization" indicates whether there are multiple training subdomains ($|\mathcal{D}^{\text{train}}| \geq 2$) and thus whether domain generalization models can be applied to this task. "Baseline gap" gives the "shift gap" $\Delta_{\text{Acc}}$ (difference between ID and OOD test accuracy, see Equation (1)) of the tuned XGBoost or LightGBM model with the best validation accuracy after following our hyperparameter tuning procedure (§4.2).

| Task | Target | Shift | Domain Generalization | Baseline Gap $\Delta_{\text{Acc}}$ | SE($\Delta_{\text{Acc}}$) |
|------|--------|-------|:---------------------:|:----------------------------------:|:-------------------------:|
| **ASSISTments** | Next Answer Correct | School | ✓ | $-34.49$ % | 0.011 |
| **College Scorecard** | Low Degree Completion Rate | Institution Type | ✓ | $-11.16$ % | 0.010 |
| **ICU Hospital Mortality** | ICU patient expires in hospital during current visit | Insurance Type | ✓ | $-6.30$ % | 0.008 |
| **Hospital Readmission** | 30-day readmission of diabetic hospital patients | Admission source | ✓ | $-5.94$ % | 0.002 |
| **Diabetes** | Diabetes diagnosis | Race | ✓ | $-4.48$ % | 0.001 |
| **ICU Length of Stay** | Length of stay >= 3 hrs in ICU | Insurance Type | ✓ | $-3.39$ % | 0.015 |
| **Voting** | Voted in U.S. presidential election | Geographic Region | ✓ | $-2.58$ % | 0.016 |
| **Food Stamps** | Food stamp recipiency in past year for households with child | Geographic Region | ✓ | $-2.39$ % | 0.002 |
| **Unemployment** | Unemployment for non-social security-eligible adults | Education Level | ✓ | $-1.28$ % | 0.001 |
| **Income** | Income >= 56k for employed adults | Geographic Region | ✓ | $-1.25$ % | 0.002 |
| **FICO HELOC** | Repayment of Home Equity Line of Credit loan | Est. third-party risk level | | $-22.58$ % | 0.029 |
| **Public Health Insurance** | Coverage of non-Medicare eligible low-income individuals | Disability Status | | $-14.46$ % | 0.001 |
| **Sepsis** | Sepsis onset within next 6hrs for hospital patients | Length of Stay | | $-6.05$ % | 0.001 |
| **Childhood Lead** | Blood lead levels above CDC Blood Level Reference Value | Poverty level | | $-5.12$ % | 0.005 |
| **Hypertension** | Hypertension diagnosis for high-risk age (50+) | BMI Category | | $-4.36$ % | 0.003 |

**Sufficient dimensionality and size:** contains at least three features (in all cases, our benchmark datasets contain many more than three features) and at least 1000 observations. In particular, having large test sets is critical for making reliable statistical comparisons between models.

**Heterogeneous:** contains features of mixed types.

**Binary Classification:** supports a meaningful binary classification task (regression tasks are not included).

**Shift Gap:** We explicitly select datasets where strong hyperparameter-tuned tabular baselines display a statistically significant shift gap ($\Delta_{\text{Acc}} \neq 0$, see Eqn. (1)).

In addition to these criteria, we selected benchmark tasks and data sources that were *diverse*. TABLESHIFT includes tasks from many domains (finance, policy, civic participation, medical diagnosis) and from a variety of raw data sources (electronic health records, surveys/questionnaires, etc.) and with a diversity of shift gap ($\Delta_{\text{Acc}}$) magnitudes.

A summary of the benchmark tasks is shown in Table 1. We give a detailed overview of each task, including background and motivation, information on the data source, and distribution shifts, in Section B. Datasets and each individual feature of each task are also documented in the Python package. One important aspect of TableShift's diversity, shown in Table 1, is that not all real-world tasks support domain generalization (i.e. not all tasks have multiple training subdomains, $|\mathcal{D}^{\text{train}} \geq 2|$). To reflect this, we include both types of tasks in the TableShift benchmark.

While the intended use of TableShift is for distribution shift, the package is also likely to be of high utility to all researchers studying tabular data modeling due to the data quality, detailed documentation, flexible preprocessing, and ease of use of the datasets in TableShift.

## 3.2  TableShift API

Successful existing benchmarks for distribution/domain shift in machine learning (e.g. WILDS, DomainBed) not only include high-quality datasets, but also make the data *accessible* by providing a high-quality API as an interface to the otherwise-disparate sources. This section describes the TableShift API. Providing this API for tabular data is particularly important, for several reasons.

First, the input and output of tabular data pipelines differ from other modalities: tabular datasets are stored in different formats from image and text datasets, and are used with a greater variety of machine learning tools (e.g. `scikit-learn`). Second, the preprocessing operations used in tabular data differ significantly from other data modalities. These preprocessing operations also require unique feature-level metadata such as data types (i.e. categorical vs. numeric; numeric values for categorical features are a common encoding scheme in practice) and codings for categorical variables. Finally, raw sources used to build tabular datasets can be difficult to access. Datasets are often scattered across hundreds or even thousands of files (e.g., the Sepsis task dataset is constructed from over $40k$ data files; the Childhood Lead dataset is joined from nearly 100 files containing disjoint feature sets provided by the National Health and Nutrition Examination Survey (NHANES)).

The TableShift API addresses each of these issues. It defines a set of primitives which allow for the construction of data pipelines which go from raw data sources to preprocessed data of any TableShift benchmark task in a few lines of Python code[3]. The resulting data is documented – each feature in the benchmark includes metadata which describes the feature and any encodings. The API natively supports a set of common data transformations, including one-hot and label encoding for categorical data; scaling and binning of numeric data; and handling of missing values. TableShift provides native output in a variety of data formats, including PyTorch DataLoaders, Pandas DataFrames, and Ray Datasets. Finally, *any* dataset in the TableShift benchmark can be loaded with default preprocessing parameters with an identical call to the API, providing a unified interface.

We provide a a detailed comparison between TableShift and related existing benchmarks in Section G. However, we emphasize that there is *no* existing benchmark suite for distribution shift in tabular data, and existing distribution shift benchmarks are incompatible with the unique constraints of tabular data discussed above.

---

[3]See https://tableshift.org and https://github.com/mlfoundations/tableshift

# 4 Experiment Setup

We conduct a set of experiments to demonstrate the potential insights to be gained from using TableShift. As previously mentioned, there has been considerable debate about whether tree-based models (XGBoost, LightGBM, etc.) or specialized deep learning-based models (i.e. ResNet- and Transformer-based architectures) are more effective for tabular data modeling. However, previous investigations have not explored how these models perform under *distribution shift* in tabular data. Additionally, many methods have been proposed for robust learning and domain generalization but also not rigorously evaluated on tabular data. We present a series of experiments to evaluate 19 distinct methods using the TABLESHIFT benchmark.

## 4.1 Tabular Data Classification Techniques in our Comparison

We train and evaluate a set of tabular data classifiers from several families. For each, we give additional details and description in Section F, and the full hyperparameter grids in Table 19. Implementations of these classifiers, including the hyperparameter tuning framework used to tune them, are available in the TABLESHIFT API. The classifiers compared in our experiments are:

**Baseline Models:** These models do not include any intervention for robustness to domain shift, but are generally effective for tabular data in the IID setting. We evaluate multilayer perceptrons (MLP), XGBoost [20], LightGBM [48], and CatBoost [25] as baseline methods. While we refer to these as "baselines" for convenience, we note that the methods based on gradient-boosted trees (XGBoost, LightGBM, CatBoost) are still considered state-of-the-art on many tasks [37].

**Tabular Neural Networks:** We also include a set of state-of-the-art methods for modeling tabular data. The models we use are SAINT [79], TabTransformer [43], NODE [70], FT-Transformer, and tabular ResNet (the latter two via [36]).

**Domain Robustness Models:** These models attempt to ensure good performance on distributions close to the training data. These models attempt to optimize an objective over a worst-case distribution with bounded distance from the training data. We evaluate distributionally robust optimization (DRO) with both $\chi^2$ and CVaR geometry [53], and group DRO (where the groups are domains) [76]. Both the DRO and group DRO models are parameterized over MLPs, as in both original works.

**Label Shift Robustness Models:** These models attempt to ensure good performance when the label distribution $P(y)$ changes. We evaluate Group DRO (where the groups are class labels) and the adversarial label robustness method of [92].

**Domain Generalization Models:** These are models designed with the goal of achieving low error rates on unseen test domains. In practice, this is achieved by achieving low error *disparity* across the subdomains in $\mathcal{D}^{\text{train}}$. These methods require domain labels at training time, and training data drawn from multiple different domains ($|\mathcal{D}^{\text{train}}| \geq 2$). Domain generalization models in our study are: Domain-Adversarial Neural Networks (DANN) [1], Invariant Risk Minimization (IRM) [6], Domain MixUp [90, 89], Risk Extrapolation (VReX) [52], DeepCORAL [82] and MMD [54].

We note that our goal of the current study is not to propose novel methods for distributionally robust learning; it is to conduct a comprehensive comparison of a large set of existing methods, many of which have not been previously compared to each other, on a high-quality benchmark. For example, while domain generalization models have been applied to image and text classification tasks (e.g. [50, 38]), to our knowledge these methods have not been previously investigated for mitigating distribution shift in *tabular* data in a large-scale benchmarking study. Indeed, we are aware of no prior applications of many of these domain generalization methods to tabular data. As a result, it is not clear *a priori* how these methods might compare to existing robustness or baseline methods due to the aforementioned differences between tabular data and these other data modalities.

The experiments described above cover both model architectures (different functional forms for the predictor $f_\theta$) and loss functions (different objective functions $\mathcal{L}$ used to train the model by attempting to find $\min_\theta \mathcal{L}(f_\theta(\mathcal{D}^{\text{train}}))$). In order to train a classifier with gradient-based training, both are required. Except where noted otherwise, any method requiring gradient-based training (MLP, Tabular Neural Networks, Domain Generalization Models) is trained with standard empirical risk minimization and cross-entropy loss. Similarly, any method which itself is a loss function (i.e. all variants of DRO) is

trained with $f$ parameterized as an MLP, as is standard in prior works implementing and comparing these methods (e.g. [53, 76, 33]).

## 4.2 Methods

For each task, we conduct the following procedure.

First, we split the full dataset into $\mathcal{D}^{\text{train}}$ and $\mathcal{D}^{\text{test}}$. We summarize the domain splits in Tables 1,1 and describe the splitting for each task in detail, along with background and motivation for each task domain split, in Section B. Within each domain, we have both a validation and a test set. We use the same domain splits, data preprocessing, and train/validation/test splits for all models and training runs, except where explicitly noted.

Second, we then conduct a hyperparameter sweep for each model described in Section 4.1. We use HyperOpt [13] to sample from the model hyperparameter space, in accordance with previous works (e.g. [36, 46]) which largely use adaptive hyperparameter optimization due to the variability in effective hyperparameter settings between datasets. We only train on the training set, and use the in-domain validation accuracy for hyperparameter tuning. We give the complete grid for each model in §I. Each model is tuned for 100 trials.

Finally, we evaluate the trained models on the test splits of each dataset. As recommended in [50], we use in-domain and out-of-domain *test* accuracy (not in-domain train accuracy) to evaluate the models. For all results shown, we use the best model selected according to (in-domain) validation accuracy; this follows the selection procedure used to study domain generalization in the image domain in [38].

# 5 Empirical Results

**ID and OOD Accuracy are Correlated.** Our results show that, across all models and tasks, in-distribution (ID) and out-of-distribution (OOD) accuracy are correlated: as ID performance improves, OOD performance also tends to improve (see Figure 1; $\rho = 0.81$). This linear trend holds *across datasets and model classes*. We note that, while this is consistent with findings for image [62] and question answering [61] models, the relationship between ID accuracy and OOD accuracy on tabular data was previously unknown. This result suggests that, for a wide variety of tabular data tasks, improving models' ID performance is likely to improve their OOD performance.

**No Model Consistently Outperforms Baselines.** While many models have been proposed for both (a) improving general performance on tabular data tasks over established baselines such as XGBoost and LightGBM, and (b) improving robustness to distribution shift, our results show that no model consistently outperforms the standard tabular baselines of XGBoost, LightGBM, or CatBoost in either respect. Figure 4a shows that, on average across all datasets, no model consistently achieves better performance (as measured as a fraction of the maximum OOD accuracy achieved by any model) compared to baseline methods. This finding has not been previously demonstrated in tabular data due to the lack of an existing benchmark.

**No Method Eliminates Gaps.** We investigate the empirical performance of several methods designed to improve robustness to distribution shift (described in Section 4.1). Our results shows that, on the datasets where multiple training subdomains are available (and thus where domain generalization is viable), there is weak evidence that several techniques reduce gaps due to distribution shift, but no technique eliminates these gaps. However, it is important to note that this gap reduction comes at the cost of in-distribution performance: as Figure 4b shows, all robustness-enhancing models tend to shrink gaps by *reducing average ID performance, not by improving OOD performance*. This is shown in Figure 4b by the two parallel lines: one set of blue points representing baselines + tabular NNs, and another, shifted left, representing robustness-engancing and domain generalization models. Furthermore, we note that all domain generalization and domain robustness methods evaluated (excluding DRO) require additional information that is only present for some datasets – namely, the discrete variable over which a shift will occur (e.g. "race" for diabetes task) and data from at least 2 categories of this variable.

**Change in label distribution is correlated with shift gap.** We investigate the degree to which the three factors mentioned previously ($p(x)$, $p(y|x)$, $p(y)$) are related to model performance. Our results, in Figures 5 and 8, show that change in the label distribution $\Delta_y$ is correlated with shift gap

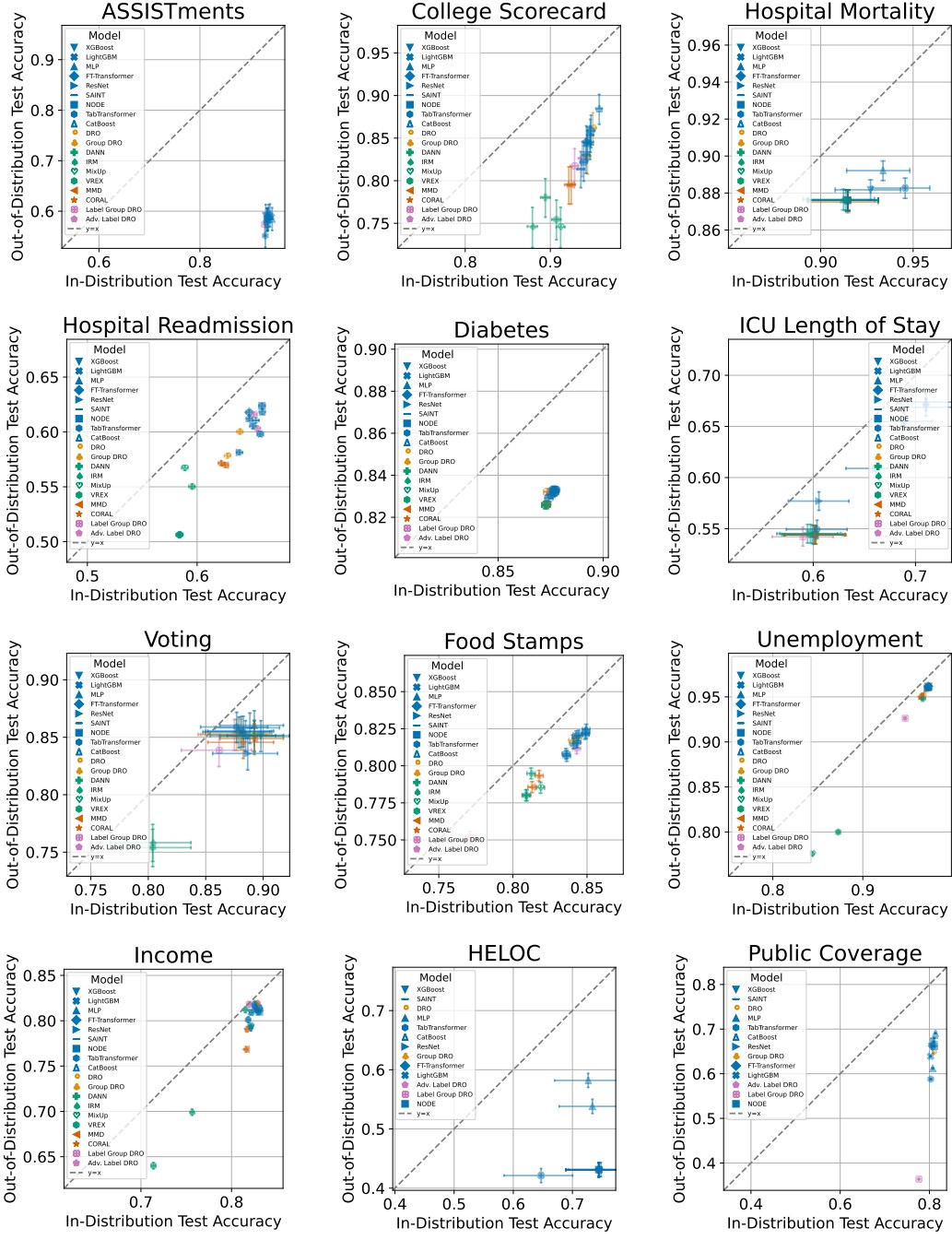

Figure 2: Results for baselines, robust learning, and domain generalization models across the 15 TABLESHIFT benchmark tasks. The $y = x$ line indicates a model with no shift gap, $\Delta_{\mathrm{Acc}} = 0$ (see Equation 1). Clopper-Pearson confidence intervals at $\alpha = 0.05$ shown for all points. Note that domain generalization models are only used on domain generalization tasks (cf. Table 1). Results for the remaining TABLESHIFT tasks are shown in Figure 3. For exact results see Section E.3.

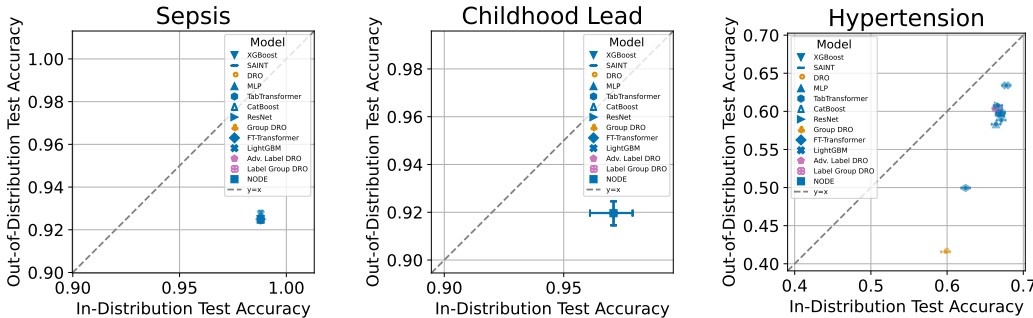

Figure 3: Additional results (cf. Figure 2). For exact results see Section E.3.

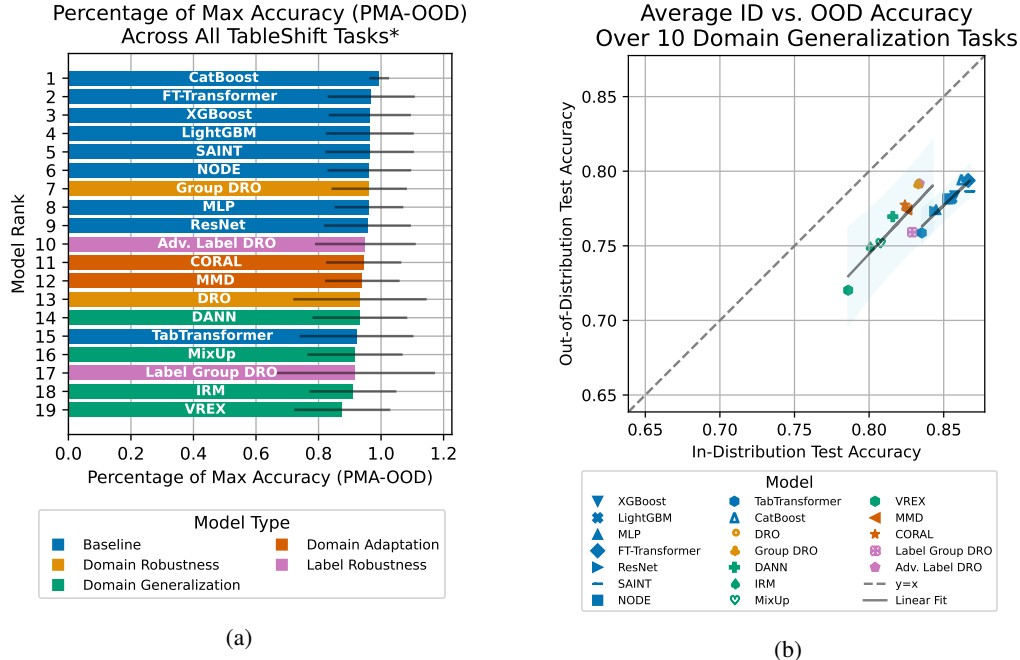

Figure 4: (a): Percentage of Maximum OOD Accuracy (PMA-OOD) across tasks (see Table 14 for exact values). *: domain generalization models and Group DRO can only be trained on the subset of 10 tasks with multiple training subdomains (see "Domain Generalization" column in Table 1). (b): Average ID and OOD accuracy by model across domain generalization tasks only. We only show domain generalization tasks in order to compare all models on the same set of tasks. See Figure 9 for results on all tasks. Exact values in Table 15.

$\Delta_{\text{Acc}}$ (Pearson correlation $\rho = 0.71$). This persists even after accounting for ID accuracy: a simple linear regression of OOD accuracy on [ID accuracy, $\Delta_y$] obtains $R^2 = 0.996$. This suggests that the change in the label distribution is an important factor in understanding tabular shifts (for example, the outliers in Figure 1 are from the four tasks with largest label shift: Public Coverage, HELOC, ASSISTments, College Scorecard; see Figures 2, 3 and Table 3). Label shift robustness methods in our study *did not* eliminate performance gaps under shift; in fact, label shift robustness methods often degraded both ID and OOD accuracy (e.g. Figure 4b). We provide similar analyses relating shift gap to $(i)$ covariate shift and $(ii)$ concept shift in Figure 8, but find that they are not clearly related.

**Changes in predictions are related to covariate shift.** As an exploratory finding, we find some evidence that changes in the predictions for OOD data are correlated with changes in $p(x)$, shown in Figure 7a ($\rho = 0.99$). This suggests that shift gaps in the benchmark datasets not explained by the combination of ID accuracy and $\Delta_y$ may be driven primarily by covariate shift (changes in $p(x)$) as opposed to concept shift (changes in $p(y|x)$). Further analysis is needed to confirm this exploratory finding. We note that relationships between other forms of shift showed much weaker correlation, roughly $\rho \approx -0.2$ (see Figure 7).

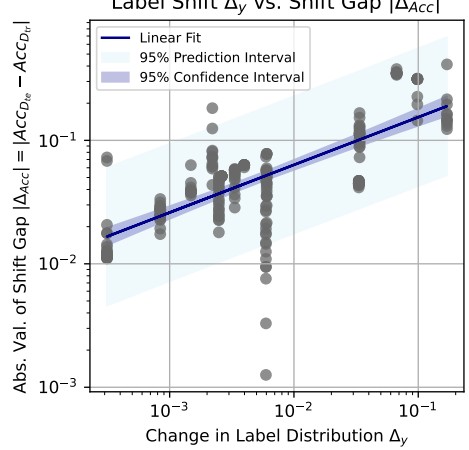

Figure 5: Label shift ($\Delta_y$, measured via Equation (3)) and absolute shift gap $\Delta_{\text{Acc}}$ show moderate correlation across datasets and models (Pearson correlation $\rho = 0.70$). Exact $\Delta_y$ values in Table 3.

## 6  Limitations

The conclusions in this study are limited to the specific datasets and models evaluated. While we intentionally selected a diverse suite of benchmark datasets along several axes (domain, distribution shift, dataset size, etc.), our conclusions can only be extended to other distribution shifts insofar as they are similar to the shifts in TABLESHIFT. More empirical validation is needed, including studies comparing our findings to other tabular shifts.

Our work does not exhaustively cover the space of all possible tabular data classifiers. In particular, "hybrid" methods combining some of the loss-based robustness interventions (i.e. Group DRO) with various tabular data-specific model architectures (e.g. FT-Transformer, ResNet) might lead to different results. Our initial exploratory evaluation of hybrid methods (see Section E.5), however, does not suggest that hybrid methods led to qualitative changes in our results, but these methods warrant a more extensive evaluation. Finally, our work does not establish *theoretical* connections between the factors analyzed (ID accuracy, OOD accuracy, $\Delta_y$).

## 7  Conclusion

We introduce the TABLESHIFT benchmark for studying distribution shift in tabular data. TABLESHIFT presents a diverse set of tasks for reliable study and benchmarking of tabular data models under distribution shift. We provide a Python API to access the datasets, along with implementations of several models including baselines, distributionally robust learners, and domain generalization methods. Finally, we present empirical results which form the first large-scale study of tabular data modeling under distribution shift.

Our results suggest multiple potential avenues for future work: First, improvements to *in-distribution* accuracy are likely to drive OOD accuracy gains. Second, improved robustness to *label shift* may reduce shift gaps. Third, *hybrid methods* which combine robustness-enhancing losses (such as Group DRO) with improved neural network architectures may be able to further improve OOD performance. Beyond these proposed directions, we hope that TableShift opens new research frontiers for tabular machine learning research beyond those addressed in the current work.

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

# A    Acknowledgements

This work was supported by a Microsoft Grant for Customer Experience Innovation. This work was also in part supported by the NSF AI Institute for Foundations of Machine Learning (IFML, CCF-2019844), Google, Open Philanthropy, and the Allen Institute for AI.

Our research utilized computational resources and services provided by the Hyak computing cluster at the University of Washington.

# B    Benchmark Task Details

This section provides details on each of the benchmark tasks in TableShift. While we describe the data source for each task, we emphasize that *TableShift does not host or distribute the data*; each data source is publicly available (some require training or authorization, but all are available to the public). Based on our review of the datasets, we believe that the datasets do not contain personally identifiable information, offensive content, or proprietary information. For data collected from human subjects, the conditions of collection and the ethics approval under which the data were collected are described in the documentation associated with each dataset.

## B.1    Food Stamps

**Background:** Food insecurity is a problem affecting more than 10% of households (13.5 million) across the United States in 2021[4]. Various programs exist to provide families and individuals with supplemental income to reduce food insecurity. However, diminished social support services in many U.S. states limit the ability of outreach providers to ensure all aligible individuals are receiving available benefits. Low-cost, low-friction screening tools powered by machine learning models might provide useful information whether an individual is receiving food stamps in order to identify lilely candidates for both food security programs ("food stamps") and as a proxy for eligibility and need for additional support services.

**Data Source:** We use person-level data from the American Community Survey (ACS)[5]. We filter the data for low-income adults aged 18-62 (i.e. selecting only adults below the social security eligibility age) in households with at least one child in the household. We use an income threshold of $30000 based on the U.S. poverty threshold for a family with one child.

**Distribution Shift:** In the United States, food stamps programs are managed at the state level. We apply domain shift over states, at the *regional* level. Specifically, we use the ACS census region as the split. The ACS includes 10 regions, which are: Puerto Rico; New England (Northeast region); Middle Atlantic (Northeast region); East North Central (Midwest region); West North Central (Midwest region); South Atlantic (South region); East South Central (South region); West South Central (South Region); Mountain (West region); Pacific (West region). We use East South Central (South region) as the holdout domain for this task.

This split parallels the case where a system is trained on a subset of states in a specific geographic area (perhaps in a localized study that draws participants or respondents from some geographic areas, but excludes other areas), and then applied to another. It also parallels the case where there is an interest in simulating the effect of a policy change. Finally, it mirrors the challenge of predicting an effect of a policy outcome (food stamps eligibility/recipiency) where differences in the underlying policy (different programs or eligibility across states) are a confounder.

## B.2    Income

**Background:** Income is a widely-used measure of social stability. In addition, income is often used as a criteria for various social support programs. For example, in the United States, income is used to measure poverty, and can be used determine eligibility for various social services such as food stamps and medicaid. Income prediction has obvious commercial utility. Finally, income prediction

---

[4]https://www.ers.usda.gov/topics/food-nutrition-assistance/food-security-in-the-u-s/key-statistics-graphics/
[5]https://www.census.gov/programs-surveys/acs/about.html

has a rich and unique history in the machine learning community, dating back to the "adult income" census dataset [51, 24].

**Data Source:** We use person-level data from the American Community Survey (ACS), as described in Task B.1. However, for the income prediction task, we use different filtering. We use the filtering described in [24], which filters for adults aged at least 16 years old, who report working more than zero hours in the past month with reported income at least $100.00. We use an income threshold of $56,000, which is the median income, as in [33].

**Distribution Shift:** Income patterns can vary in many ways. Here, we focus on domain shift at the *regional* level. We use the same splitting variable (US Census Region) described in Task B.1. However, for the income prediction task, we use New England (Northeast region) as the held-out domain.

## B.3  Public Coverage

**Background:** People use health-care services to diagnose, cure, or treat disease or injury; to improve or maintain function; or to obtain information about their health status and prognosis [67]. In the United States, health insurance is a critical component of individuals' ability to access health care. Public health insurance exists, among other reasons, to provide affordable and accessible health insurance options for individuals not willing or able to purchase insurance through the private insurance market. However, not all individuals have health insurance; only 88% of individuals in the U.S had health insurance in 2019 according to the National Health Interview Survey (NHIS). Increasing the proportion of people in the United States with health insurance is one of the four healthcare objectives of the U.S. Department of Health and Human Services "Healthy People 2030" initiative[6]. In this task, the goal is to predict whether an individual is covered by public health insurance.

**Data Source:** We use person-level data from the American Community Survey (ACS), as described in Task B.1. However, for this task, we filter the data to include only low-income individuals (those with income less than $30,000) who are below the age of 65 (at which age all persons in the United States are covered by Medicare). This is the same filtering used in [24, 33].

**Distribution Shift:** Many factors can influence individuals' ability to access or utilize health insurance and healthcare services. These include spoken language skills, mobility (whether an individual has recently relocated), education, ease of obtaining services, and discriminatory practices among providers [67]. We focus on *disability status*, as this is a widely-known factor in obtaining access to adequate health care [67]. Disability is also a particularly realistic factor in that disability status is likely to contribute to nonresponse to certain forms of data collection for many tabular data sources (including the four methods used to collect the ACS data: internet, mail, telephone, and in-person interviews) that can disadvantage persons with certain disabilities and decrease likelihood of participation or cause them to be excluded from study population.

For this task, the holdout domain $\mathcal{D}^{\text{test}}$ consists of persons with disabilities; the training domain $\mathcal{D}^{\text{train}}$ consists of persons who do not have disabilities. This simulates a situation where data collection practices excluded disabled persons, potentially through the factors described above.

## B.4  ACS Unemployment

**Background:** Unemployment is a key macroeconomic indicator and a measure of individual well-being. Unemployment is also linked to a variety of adverse outcomes, including socioeconomic, psychological, and health impacts [10, 16, 14, 63].

**Data Source:** We use person-level data from the American Community Survey (ACS), as described in Task B.1. However, for this task, we filter the data to include only individuals over the age of 18 and below the age of 62 (at which age persons in the United States are eligible to receive Social Security income).

**Distribution Shift:** Many factors are known to be related to unemployment. We focus on a form of subpopulation shift, and use *education level* as the domain split. We use individuals with educational

---

[6]https://health.gov/healthypeople/objectives-and-data/browse-objectives/health-care-access-and-quality/increase-proportion-people-health-insurance-ahs-01

attainment of GED (high school diploma equivalent) or higher as the training population $\mathcal{D}^{\text{train}}$, and individuals without high school-level education as $\mathcal{D}^{\text{test}}$. This simulates a survey collection with a biased sample that systematically excludes such persons.

### B.5 Diabetes

**Background:** Diabetes is a chronic disease that affects at least 37.7million people in the United States (11.3% of the U.S population); it is estimated that an additional 96 million adults have prediabetes.[7] Diabetes increases the risk of a variety of other health conditions, including stroke, kidney failure, renal complications, peripheral vascular disease, heart disease, and death. The economic cost of diabetes is also significant: The total estimated cost of diagnosed diabetes in 2017 is $327 billion [7]. Care for people with diagnosed diabetes accounts for 1 in 4 health care dollars in the U.S. – more than half of that expenditure is directly attributable to diabetes [7].

Early detection of diabetes thus stands to have a significant impact, allowing for clinical intervention and potentially reducing the prevalence of diabetes. Further, even prediabetes is ackowledged to have significant impacts both on health outcomes and quality of life [7], and early detection if high diabetes risk could serve to identify prediabetic individuals. There exists a considerable prior literature on models for early diabetes prediction, e.g. [88, 66, 42]

**Data Source:** We use data provided by the Behavioral Risk Factors Surveillance System (BRFSS)[8]. BRFSS is a large-scale telephone survey conducted by the Centers of Disease Control and Prevention. BRFSS collects data about U.S. residents regarding their health-related risk behaviors, chronic health conditions, and use of preventive services. BRFSS collects data in all 50 states as well as the District of Columbia and three U.S. territories. BRFSS completes more than 400,000 adult interviews each year, making it the largest continuously conducted health survey system in the world. BRFSS annual survey data from 2017-2021 is currently available from the CDC.

The BRFSS is composed of three components: 'fixed core' questions, asked every year, 'rotating core', asked every other year, and 'emerging core'. Since some of our features come from the rotating core, we only use every-other-year data sources; otherwise many features would be empty for the intervening years.

For the Diabetes prediction task, we use a set of features related to several known indicators for diabetes derived from [88]. These risk factors are general physical health, high cholesterol, BMI/obesity, smoking, the presence of other chronic health conditions (stroke, coronary heart diseas), diet, alcohol consumption, exercise, household income, marital status, time since last checkup, education level, health care coverage, and mental health. For each risk factor, we extract a set of relevant features from the BRFSS foxed core and rotating core questionnaires. We also use a shared set of demographic indicators (race, sex, state, survey year, and a question related to income level). The prediction target is a binary indicator for whether the respondent has ever been told they have diabetes.

**Distribution Shift:** While diabetes affects a large fraction of the overall population, diabetes risk varies according to several demographic factors. One such factor is race/ethnicity [42, 17], with all other race-ethnicity groups reported in the 2022 CDC National Diabetes Statistics Report displaying higher risk than 'White non-Hispanic' individuals[17]. Compounding this issue, it has been widely acknowledged that health studiy populations tend to be biased toward white European-Americans [23, 68, 26, 44]. As a result, these studies have tended to focus on risk factors affecting white populations at the expense of identifying risk factors for nonwhite populations [44], despite distinct differences in how these populations are affected by various disease risk factors, differences in individuals' genetic factors, and differences in how they respond to medication across racial and ethnic populations. This disparity is a contributing factor to race-based disparities in treatment for diabetes [21].

In order to simulate the domain gap induced by these real-world differences in study vs. deployment populations, we partition the benchmark task by race/ethnicity. We use "White non-Hispanic"-identified individuals as the training domain, and all other race/ethnicity groups as the target domain.

---

[7] https://www.cdc.gov/diabetes/health-equity/diabetes-by-the-numbers.html
[8] https://www.cdc.gov/brfss/index.html

## B.6   Hypertension

**Background:** Hypertension, or systolic blood pressure (typically systolic pressure 130 mm Hg or higher or diastolic 80 or higher) affects nearly half of Americans [3]. Hypertension is sometimes called a "silent killer" because in most cases, there are no obvious symptoms of hypertension [3]; this would make an accurate at-risk model of hypertension useful. When left untreated, hypertension is associated with the strongest evidence for causation of all risk factors for heart attack and other cardiovascular disease [32]. Hypertension also increases the risk of stroke, kidney damage, vision loss, insulin resistance, and other adverse outcomes [4]. While existing tools have attempted to predict blood pressure without the use of a cuff (the gold-standard measurement of blood pressure), these tools are still significantly less accurate (see e.g. [77, 28]), and there is an ongoing need for effective blood pressure measurement.

**Data Source:** We use BRFSS as the raw data source, as described in Task B.5 above. However, for the hypertension prediction task, we use features related to the following set of risk factors for hypertension via [64]: Age, family history and genetics, other medical conditions (e.g. diabetes, various forms of cancer), race/ethnicity, sex, and social and economic factors (income, employment status). We collect all survey questions related to these risk factors and use them as the predictors for this task, along with a shared set of demographic indicators (race, sex, state, survey year, and a question related to income level).

**Distribution Shift:** We use BMI category as the domain splitting variable. Individuals with BMI identified as "overweight" or "obese" are in the held-out domain, and those identified as "underweight" or "normal weight" are in the training domain. This simulates a model being deployed under subpopulation shift, where the target population has different (higher) BMI than the training population.

## B.7   Voting

**Background** Understanding participation in elections is a critical task for policymakers, politicians, and those with an interest in democracy. In the 2020 United States presidential election, for example, voter turnout reached record levels, but it is estimated that only 66.8% of eligible individuals voted according to the U.S. Census[9]. Additionally, so-called "likely voter models," that predict which individuals will vote in an electio, are widely acknowledged as critical to polling and campaigning in U.S. politics. Predicting whether an individual will vote is notoriously difficult; one reason for this challenge is that domain shift is a fundamental reality of such modeling (presidential elections only occur every four years, after which significant political and demographic changes occur prior to the next presidential election).

The prediction target for this dataset is to determine whether an individual will vote in the U.S presidential election, from a detailed questionnaire.

**Data Source** We use data from the American National Election Studies (ANES)[10]. Since 1948, ANES has conducted surveys, usually administered as in-person interviews, during most years of national elections. This series of studies, known as the ANES "Time Series," constitutes a pre-election interview and a post-election interview during years of Presidential elections, along with other data sources. Topics cover voting behavior and the elections, together with questions on public opinion and attitudes.

We use features derived from the ANES Time Series. From the pool of over 500 questions in the ANES Time Series, we extract a set of features related to Americans' voting behavior, including their social and political attitudes, opinions about elected leaders, and media consumption habits.

**Domain Shift** We introduce a domain split by geographic region. We use the ANES Census Region feature, where the out-of-domain region is the region representing the southern United States (AL, AR, DE, D.C., FL, GA, KY, LA, MD, MS, NC, OK, SC,TN, TX, VA, WV). This simulates a study in which voter data is collected in one part of the country, and the goal is to infer voting behavior in another geographic region; this is a common occurence with polling data, particularly during the U.S. primaries, which occur over a period of several weeks at the state level.

---

[9]https://www.census.gov/library/stories/2021/04/record-high-turnout-in-2020-general-election.html

[10]https://electionstudies.org/

## B.8 Childhood Lead Exposure

In this task, the goal is to identify children 18 or younger with elevated lead blood levels.

**Background:** Lead is a known environmental toxin that has been shown to affect deleteriously the nervous, hematopoietic, endocrine, renal, and reproductive systems[11]. In young children, lead exposure is a particular hazard because children more readily absorb lead than adults, and children's developing nervous systems also make them more susceptible to the effects of lead. However, most children with any lead in their blood have no obvious immediate symptoms.[12] The risk for lead exposure is disproportionately higher for children who are poor, non-Hispanic black, living in large metropolitan areas, or living in older housing.

The CDC sets a national standard for blood lead levels in children. This value was established in 2012 to be 3.5 micrograms per deciliter ($\mu$g/dL) of blood.[13] This value, called the blood lead reference value (BLRV) for children, corresponds to the 97.5 percentile and is intended to identify lead exposure in order to allow parents, doctors, public health officials, and communities to act early to reduce harmful exposure to lead in children. Thus, early prediction of childhood lead exposure, as well as accurate just-in-time prediction for children where obtaining actual laboratory blood test results is too costly or infeasible, is of high utility to many stakeholders.

Early detection of lead exposure can trigger many potentially impactful interventions, including: environmental and home analysis for early identification of sources of lead; testing and treatment for nutritional factors influencing susceptibility to lead exposure (such as calcium and iron intake); developmental analysis and support; and additional medical diagnostic tests.[14]

Using the laboratory blood test results from the NHANES (see 'Data Source' below), the task is to identify whether a respondents' blood level exceeds the BLRV *using only questionnaire data*. We use respondents of age 18 or younger as the target population (note that respondent data for ages 1-5 is restricted and thus not available to our benchmarking study). This simulates the prediction of expensive and time-consuming laboratory testing using a quick and inexpensive questionnaire. Laboratory testing is conducted by the CDC at the National Center for Environmental Health, Centers for Disease Control and Prevention, Atlanta, GA[15]

**Data Source:** The data are drawn from the CDC National Health and Nutrition Examination Survey (NHANES)[16], a program of the National Center for Health Statistics (NCHS) within the Centers for Disease Control and Prevention (CDC). NHANES is a program of studies designed to assess the health and nutritional status of adults and children in the United States. The survey is unique in that it combines extensive interviews with physical examinations and high-quality laboratory testing. The NHANES interview includes demographic, socioeconomic, dietary, and health-related questions. The survey examines a nationally representative sample of about 5,000 persons each year. The examination component consists of medical, dental, and physiological measurements, as well as laboratory tests administered by highly trained medical personnel.

Findings from NHANES are used to determine the prevalence of major diseases and risk factors for diseases; to assess nutritional status and its association with health promotion and disease prevention; and are the basis for national standards for such measurements as height, weight, and blood pressure. Data from this survey are widely used in epidemiological studies and health sciences research.

We use only questionnaire-based NHANES features as the predictors, but use a prediction target from the NHANES' lab-based component. This simulates the development of a screening questionnaire to predict blood lead levels.

**Distribution Shift:** We use poverty as a domain-splitting variable. Children from low-income households and those who live in housing built before 1978 are at the greatest risk of lead exposure[17].

---

[11]https://wwwn.cdc.gov/Nchs/Nhanes/2017-2018/P_PBCD.htm

[12]https://www.cdc.gov/nceh/lead/prevention/blood-lead-levels.htm

[13]https://www.cdc.gov/nceh/lead/data/blood-lead-reference-value.htm

[14]https://www.cdc.gov/nceh/lead/advisory/acclpp/actions-blls.htm

[15]A detailed description of the methods and procedures used for laboratory testing for lead in the 2017-2018 NHANES survey is given at https://wwwn.cdc.gov/Nchs/Nhanes/2017-2018/P_PBCD.htm; similar descriptions are available for each year of data collection.

[16]https://wwwn.cdc.gov/Nchs/Nhanes/

[17]https://www.cdc.gov/nceh/lead/prevention/populations.htm

However, due to factors mentioned above, impoverished populations can be less likely to be included in medical studies, including those that may involve in-person visits for blood laboratory testing, which is the primary method for lead exposure detection. We use the poverty-income ratio (PIR) measurement in NHANES. The PIR is calculated by dividing total annual family (or individual) income by the poverty guidelines specific to the survey year. The Department of Health and Human Services (HHS) poverty guidelines are used as the poverty measure to calculate this ratio. These guidelines are issued each year, in the Federal Register, for determining financial eligibility for certain federal programs, such as Head Start, Supplemental Nutrition Assistance Program (SNAP), Special Supplemental Nutrition Program for Women, Infants, and Children (WIC), and the National School Lunch Program. The poverty guidelines vary by family size and geographic location (with different guidelines for the 48 contiguous states and the District of Columbia; Alaska; and Hawaii).

The training domain is composed of individuals with PIR of at least 1.3; persons with PIR $\leq$ 1.3 are in the held-out domain. The threshold of 1.3 is selected based on the PIR categorization used in NHANES, where PIR $\leq$ 1.3 is the lowest level.

### B.9 Hospital Readmission

**Background:** Effective management and treatment of diabetic patients admitted to the hospital can have a significant impact on their health outcomes, both short-term and long-term [83]. Several factors can affect the quality of treatment patients receive [81]. One of the costliest and potentially most adverse outcomes after a patient is released from the hospital is for that patient to be *readmitted* soon after their initial release; this can both be a sign of a condition that is not improving, and, at times, ineffective initial treatment. Thus, predicting the readmission of patients is a priority from both a medical and economic perspective.

In this task, the goal is to predict whether a diabetic patient is *readmitted* to the hospital within 30 days of their initial release.

**Data Source:** We use the dataset provided by [81][18]. The dataset represents 10 years (1999-2008) of clinical care at 130 US medical facilities, including hospitals and other networks. It includes over 50 features representing patient and hospital outcomes. The dataset includes observations for records which meet the following criteria: (1) It is an inpatient encounter (a hospital admission). (2) It is a diabetic encounter, that is, one during which any kind of diabetes was entered to the system as a diagnosis. (3) The length of stay was at least 1 day and at most 14 days. (4) Laboratory tests were performed during the encounter. (5) Medications were administered during the encounter.

The data contains such attributes as patient number, race, gender, age, admission type, time in hospital, medical specialty of admitting physician, number of lab test performed, HbA1c test result, diagnosis, number of medication, diabetic medications, number of outpatient, inpatient, and emergency visits in the year before the hospitalization, etc. We use the full set of features in the initial dataset, which is described in [81].

**Distribution Shift:** Patients can be (re)admitted to hospitals from a variety of sources. The source of a patient admission canbe correlated with many demographic and other risk factors known to be related to health outcomes (e.g. race, income level, etc.).

We use the "admission source" as the domain split for TableShift. There are 21 distinct admission sources in the dataset, including "transfer from a hospital", "physician referral", etc. After conducting a sweep over various held-out values, we use "emergency room" as the held-out domain split. This matches a potential scenario where a model is constructed using a variety of admission sources, but a patient from a novel source is added; it is also possible e.g. that data from emergent patients could not be collected when training a readmission model. We note that this domain split provides 20 unique training subdomains (the other admission sources), which is the largest $|\mathcal{D}^{\text{train}}|$ in TableShift.

### B.10 Sepsis

**Background:** Sepsis is a life-threatening condition that arises when the body's response to infection causes injury to its own tissues and organs. Sepsis is a major public health concern with significant

---

[18]https://archive.ics.uci.edu/ml/datasets/Diabetes+130-US+hospitals+for+years+1999-2008

morbidity, mortality, and healthcare expenses; each year, 1.7 million adults in America develop sepsis, of which at least $350,000$ die during their hospitalization or are discharged to hospice. The CDC estimates that 1 in 3 people who dies in a hospital had sepsis during that hospitalization[19].

Early detection and antibiotic treatment of sepsis improve patient outcomes. While advances have been made in early sepsis prediction, there is a fundamental unmet clinical need for improved prediction [74]. The goal in this task is to predict, from a set of fine-grained ICU data (including laboratory measurements, sensor data, and patient demographic information), whether a patient will experience sepsis onset within the next 6 hours.

**Data Source:** We use the data source from the PhysioNet/Computing in Cardiology Challenge [74], which was designed by clinicians and other healthcare experts to facilitate the development of automated, open-source algorithms for the early detection of sepsis from clinical data. The dataset is derived from ICU patient records for over $60,000$ patients from two hospitals with up to 40 clinical variables collected during each hour of the patient's ICU stay.

**Distribution Shift:** We explored multiple domain shifts for this dataset; we note that, in particular, splitting domains by *hospital* did *not* lead to a shift gap for tuned baseline models (although there is a third, held-out hospital that was used in the original challenge for this dataset, it is not publicly available and is not part of the TableShift benchmark). Instead, we use "length of stay" as a domain shift variable. We bifurcate the dataset based on how long a patient has been in the ICU, with patients having been in ICU for $\leq 47$ hours in the training domain, and patients having been in ICU more than 47 hours in the test domain. This matches a scenario where a medical model is trained only on observed stays of a fixed duration (no more than two full days), but then used beyond its initial observation window to predict sepsis in patients with longer stays. We note that length of stay of 47 hours corresponds to the 80th percentile of the data for that feature.

### B.11 ICU Patient Length-of-Stay

**Background:** According to [72], length of hospital stay is, along with patient mortality, "the most important clinical outcome" for an ICU admission. Accurately predicting the length of stay of a patient can aid in assessment of the severity of a patient's condition. Of particular clinical relevance, making these predictions *early* and with a *non-zero time gap* between the prediction and the outcome is of real-world importance: predictions must be made sufficiently early such that a patient's treatment can be adjusted to potentially avoid a negative outcome. The importance of this prediction task for real-world clinical care is underscored by the many previous works in the medical literature addressing this prediction topic (see e.g. [40, 72, 87].

In our benchmark, the specific task is to predict, from the first 24 hours of patient data, an ICU patient's stay will exceed 3 days (a binary indicator for whether length of stay $> 3$). We note that this is directly adopted from MIMIC-extract.

**Data Source:** We use the MIMIC-extract dataset [87]. MIMIC-extract is an open-source pipeline for transforming raw electronic health record (EHR) data from the Medical Information Mart for Intensive Care (MIMIC-III) dataset [45].

MIMIC-III, the underlying data source, captures over a decade of intensive care unit (ICU) patient stays at Beth Israel Deaconess Medical Center in Boston, USA. An individual patient might be admitted to the ICU at multiple times in the dataset; however, MIMIC-extract focuses on each subject's first UCI visit only, since those who make repeat visits typically require additional considerations with respect to modeling and care [87]. MIMIC-extract includes all patient ICU stays in the MIMIC-III database that where the following criteria are met: $(i)$ the subject is an adult (age of at least 15 at time of admission), $(ii)$ the stay is the first known ICU admission for the subject, and $(iii)$ the total duration of the stay is at least 12 hours and less than 10 days.

MIMIC-extract is designed by EHR domain experts with clinical validity (of data) and relevance (of prediction tasks) in mind. In addition to the filtering described above, MIMIC-extract's pipeline includes steps to standardize units of measurement, detect and correct outliers, and select a curated set of features that reduce data missingness in the preprocessed data; for details on the steps taken by the original authors to achieve this, see [87]. We use the preprocessed version of MIMIC-extract made

---

[19]https://www.cdc.gov/sepsis/what-is-sepsis.html

available by the authors [20]. This includes the static demographic variables, alongside the time-varying vitals and labs described in [45]. Because event he preprocessed data contains missing values, we use the authors' default methods for handling missing data.

The resulting dataset contains approximately $24,000$ observations.

**Distribution Shift:** We split the domains by health insurance type. We train on patients with all insurance types except Medicare, and use patients with Medicare insurance as the target domain.

### B.12 ICU Patient In-Hospital Mortality

**Background:** As discussed in the background of §B.11, hospital mortality is considered to be one of the most important outcomes for ICU patients. The clinical relevance of hospital mortality is perhaps even more clear than for length-of-stay prediction, as preventing patient mortality is one of the primary goals for many patients. Again, as discussed in §B.11, making this prediction *early* is of particular importance, as early predictions can provide a proxy for overall patient risk and can be used to intervene to avoid mortality.

We note that in this task, we are predicting *hospital* morality (that the patient dies at any point during this visit, even if they are discharged from the ICU to another unit in the hospital). Hospital mortality events are distinct from (and a superset of) ICU mortality events. As mentioned above, the importance of this prediction task for real-world clinical care is underscored by the many previous works addressing this prediction topic (see e.g. [40, 72, 87]).

**Data Source:** This task uses the same data source and feature set from MIMIC-extract described above in §B.11.

**Distribution Shift:** We split the domains by health insurance type. We train on patients with all insurance types except { Medicare, Medicaid } and use patients with { Medicare, Medicaid } insurance as the target domain.

### B.13 FICO Home Equity Line of Credit (HELOC)

**Background:** FICO (legal name: Fair Isaac Corporation) is a US-based company that provides credit scoring services. The FICO score, a measure of consumer credit risk, is a widely used risk assessment measure for consumer lending in the united states.

The Home Equity Line of Credit (HELOC) is a line of credit, secured by the applicant's home. A HELOC provides access to a revolving credit line to use for large expenses or to consolidate higher-interest rate debt on other loans such as credit cards. A HELOC often has a lower interest rate than some other common types of loans. To assess an applicant's suitability for a HELOC, a lender evaluates an applicants' financial background, including credit score and financial history. The lender's goal is to predict, using this historical customer information, whether a given applicant is likely to repay a line of credit and, if so, how much credit should be extended.

In addition to desiring accurate credit risk predictions for their overall utility for both lenders and borrowers, lending institutions are incentivized (and, in some cases, legally required) to use models which achieve some degree of robustness: institutions can face severe penalties when borrowers are not treated equitably (e.g. [84]).

**Data Source:** We use the dataset from the FICO Commmunity Explainable AI Challenge[21], an open-source dataset containing features derived from anonymized credit bureau data. The binary prediction target is an indicator for whether a consumer was 90 days past due or worse at least once over a period of 24 months from when the credit account was opened. The features represent various aspects of an applicant's existing financial profile, including recent financial activity, number of various transactions and credit inquiries, credit balance, and number of delinquent accounts.

**Distribution Shift:** It is widely acknowledged that the dominant approach to credit scoring using financial profiles can unintentionally discriminate against historically marginalized groups (credit bureau data do not include explicit information about race [58]). For example, since FICO scores are

---

[20]The publicly-accessible dataset (which requires credentialed MIMIC-III access through PhysioNet due to privacy restrictions) is described at `https://github.com/MLforHealth/MIMIC_Extract`

[21]`https://community.fico.com/s/explainable-machine-learning-challenge`

based on payment history and credit use and many marginalized groups in the United States have lower or less reliable incomes, these marginalized groups can suffer from systematically lower credit scores [60, 71, 8, 58]; this has been referred to as the "credit gap" [49, 22]. In particular, debt and savings level play a role in credit scores and can systematically disadvantage Black and Hispanic applicants, even when demographic data are not formally used in the credit rating process [60, 58].

For this task, we partition the dataset based on the 'External Risk Estimate', a feature in the dataset corresponding to the risk estimate assigned to an applicant by a third-party service. This estimate was identified in the original FICO explanable ML challenge [22]. We use individuals with a high external risk estimate (where "high" estimate is defined as exceeding an external risk estimate of 63, a threshold identified in the original challenge-winning model linked above) as the training domain, and individuals with estimate $\leq 63$ as the held-out domain.

## B.14 College Scorecard Degree Completion Rate

**Background:** Higher education is increasingly critical to securing strong job and income opportunities for persons in the United States. At the same time, the cost of obtaining a four-year college degree is extremely high: The average cost of college* in the United States is $35,551$ per student per year, including books, supplies, and daily living expenses and this cost has more than doubled in the 21st century alone, with an annual growth rate of $7.1\%$ [39].

However, not all institutions have similar outcomes for students. Graduation rates across institutions in the U.S. vary widely, and failure to complete a degree can leave a student with significant debt and a reduced ability to repay it. Understanding factors related to degree completion is an area of active research.

For this task, our goal is to predict whether an institution has a low completion rate, based on other characteristics of that institution. While the definition of a "low" completion rate is ultimately subjective and context-dependent, we use a thredhold of 50%, which is approximately equivalent to the median graduate rate across the institutions in the dataset. We use the completion rate for first-time, full-time students at four-year institutions (150% of expected time to completion/6 years).

**Data Source:** We use the College Scorecard[23]. The College Scorecard is an institution-level dataset compiled by the U.S. Department of Education from 1996-present. The College scorecard includes detailed institutional factors, including information about each institutions' student population, course offerings, and outcomes.

**Distribution Shift:** Institutions vary widely in their profiles, student populations, educational approach, and target industries or student pathways. We partition universities according to the CCBASIC variable[24], which gives the Carnegie Classification (Basic)[25]. This classification uses a framework developed by the Carnegie Commission on Higher Education in the early 1970s to support its research program. Partitioning our data according to this variable measures the robustness over institutional subpopulations, and is thus a form of subpopulation shift. We use the following set of institutions as the target domain (all other institutional types are in the training domain): 'Special Focus Institutions–Other special-focus institutions', 'Special Focus Institutions–Theological seminaries, Bible colleges, and other faith-related institutions', "Associate's–Private For-profit 4-year Primarily Associate's", 'Baccalaureate Colleges–Diverse Fields', 'Special Focus Institutions–Schools of art, music, and design', "Associate's–Private Not-for-profit", "Baccalaureate/Associate's Colleges", "Master's Colleges and Universities (larger programs)". Exact definitions of each institution class are available via the Carnegie Commission on Higher Education[26].

---

[22]https://community.fico.com/s/blog-post/a5Q2E0000001czyUAA/fico1670

[23]https://collegescorecard.ed.gov

[24]The data dictionary for the College Scorecard is available at https://collegescorecard.ed.gov/assets/CollegeScorecardDataDictionary.xlsx

[25]https://carnegieclassifications.acenet.edu

[26]https://carnegieclassifications.acenet.edu

### B.15 ASSISTments Tutoring System Correct Answer Prediction

**Background:** Machine learning systems are increasingly being adopted in digital learning tools for students of all ages. The ASSISTments tutoring platform[27] is a free, web-based, data-driven tutoring platform for students in grades 3-12. As of 2020, ASSISTments has been used by approximately 60,000 students with over 12 million problems solved [27]. ASSISTments also periodically releases open-source data snapshots from their platform to support educational research.

**Data Source:** We use the open-source ASSISTments 2012-2013 dataset. This is a dataset from school year 2012-2013 which contains submission-level features (each row in the dataset represents one submission by a student attempting to answer a problem on the ASSISTments tutoring platform). In addition to containing student-, problem-, and school-level features, the dataset also contains affect predictions for students based on an experimental affect detector implemented in ASSISTments. (These affect predictions are intended to be useful in identifying affective states such as boredom, confusion, frustration, and engaged problem-solving behavior).

**Distribution Shift:** We partition the datasets by school. Approximately 700 schools are in the training set, and 10 schools are used as the target distribution. This simulates the process of deploying ASSISTments at a new school.

## C  Dataset Availability

All datasets in TABLESHIFT meet the definition of "available and accessible" as described in [65]; namely, the data can be obtained without a personal request to the PI. All datasets are obtained from reliable, high-quality sources (United States government agencies, UCI Machine Learning Repository, Kaggle). We selected high-quality data sources which we expect to ensure keep the relevant data available for the foreseeable future. We provide a single script that can be used to download and preprocess TABLESHIFT data for all tasks in the git repository.

The data sources used to construct the TABLESHIFT benchmark datasets vary, and necessarily so do the restrictions or agreements required to access this data. All data sources have an established credentialization procedure that is open to the public, provides rapid access to the data, and is expected to be maintained for many years. An overview of the restrictions for each dataset is given below. A link to the data use agreement or credentialization procedure for each dataset marked "open credentialized access" is available in the README of our github repo; we will maintain this list over time if the access agreements change.

Table 2: Dataset availability.

| Task | Public Access | Open Credentialized Access | Source |
|------|---------------|----------------------------|--------|
| ASSISTments | ✓ | | Kaggle |
| College Scorecard | ✓ | | Department of Education |
| ICU Hospital Mortality | | ✓ | MIMIC Clinical Database |
| Hospital Readmission | ✓ | | UCI Machine Learning Repository |
| Diabetes | ✓ | | Centers for Disease Control/BRFSS |
| ICU Length of Stay | | ✓ | MIMIC Clinical Database |
| Voting | | ✓ | American National Election Survey |
| Food Stamps | ✓ | | American Community Survey |
| Unemployment | ✓ | | American Community Survey |
| Income | ✓ | | American Community Survey |
| FICO HELOC | | ✓ | FICO |
| Public Health Ins. | ✓ | | American Community Survey |
| Sepsis | | ✓ | PhysioNet |
| Childhood Lead | ✓ | | Centers for Disease Control/NHANES |
| Hypertension | ✓ | | Centers for Disease Control/BRFSS |

---

[27]https://new.assistments.org

# D  Related Work

## D.1  Distribution/Domain Shift

The (non)robustness of modern machine learning models to distribution shift has been extensively studied, but primarily in non-tabular domains, such as vision and language [62, 61]. Through the use of diverse and high-quality benchmarking suites, several recent works have demonstrated that many existing robust learning or domain generalization methods do not outperform standard supervised training such as SGD [38, 50]. Recent evidence has also suggested that in-distribution (ID) test performance is a very strong predictor of out-of-distribution (OOD) test performance in the domains of image classification [62], language modeling [55], and question answering [9], but whether these relationships hold for tabular data is unknown.

Several families of methods have been proposed to address this sensitivity to distribution shifts, including methods for distributional robustness [76, 53] and domain generalization [1, 6, 90, 89, 54, 46] . However, these methods are largely evaluated in non-tabular domains, and several "standard" domain generalization methods have never been applied to tabular data, to our knowledge. Formal analyses of robustness to any kind of shift in the tabular domain have been lacking [33].

## D.2  Tabular Data Modeling

Tabular data – data defined by structured, heterogeneous features – is common in many real-world applications, including medical diagnosis, finance, social science, and recommender systems [15, 46, 78]. In many respects, tabular data is different from the other modalities where deep learning models have had great success in the past decade. In contrast to these other modalities, where deep learning is the undisputed state of the art, deep learning-based models have tended to underperform on tabular data, and the state of the art is often considered to be tree-based ensemble models, such as XGBoost, LightGBM, or CatBoost [15, 36, 78, 33].

Deep learning-based models have been proposed for tabular data modeling, including carefully-regularized deep multilayer perceptrons (MLPs) [46], tabular variants of ResNet [36] and Transformer architectures [43, 36, 79], and differentiable tree-inspired models [70]. However, it is unclear whether there is any benefit from these sophisticated architectures, which are often derived from models which were designed for non-tabular tasks. Subsequent evaluations of deep learning-based tabular data models have often shown tree-based models to achieve superior performance [78, 15, 33]. However, their robustness to *distribution shift* has not been thoroughly evaluated (a notable exception is [33], which strictly evaluates subgroup robustness).

## D.3  Benchmarking for Machine Learning

Benchmarking – the use of standardized, publicly-available, high-quality datasets to evaluate performance on one or more tasks – is a critical practice contributing to progress the machine learning [56]. Distribution shift benchmarks in particular have been critical in assessing progress in the robustness of vision and language models, e.g. [50, 38, 80]. Because these benchmarks often require interfacing with many distinct data sources, successful and widely-used benchmarks also typically include a lightweight software API for interfacing with benchmarking datasets in a consistent manner[28]. In the IID setting, benchmark datasets have also been crucial to assessing and driving progress, such as ImageNet [29] for vision, LibriSpeech [69] for speech, AudioSet [34] for audio classification, or GLUE for NLP [86]. Critically, evaluations have shown that reuse of these high-quality benchmarks such as CIFAR-10, ImageNet, and even widely-used Kaggle datasets has not led to "overfitting" to performance on the benchmarks [75], and, in fact, progress on these benchmarks generalizes beyond the benchmark tasks [73].

High-quality benchmarks for *tabular* data are lacking, as has been noted in many previous works [36, 15, 33, 37, 78, 57, 35]. Existing datasets used for *de facto* tabular data "benchmarking" are often of low quality. For example, the German Credit dataset contains only $1k$ observations; the COMPAS and Adult datasets have data quality and bias issues [11, 12, 24]. While a small number of general tabular benchmarks have been proposed [15, 37], they have not seen widespread adoption, do not

---

[28]e.g.  DomainBed `https://github.com/facebookresearch/DomainBed`, WILDS `https://wilds.stanford.edu/`, BIG-bench `https://github.com/google/BIG-bench`

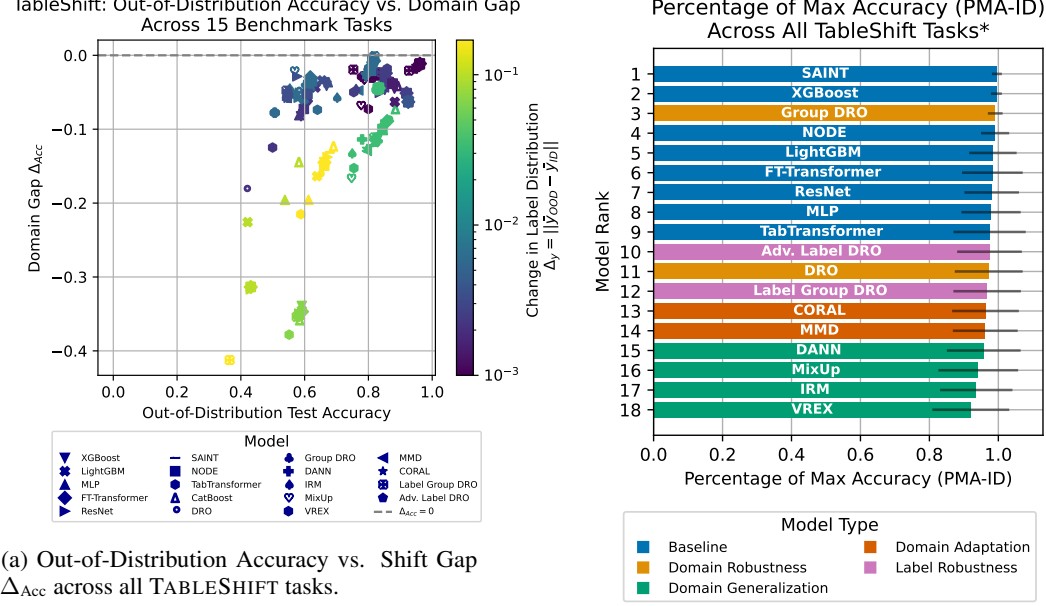

(a) Out-of-Distribution Accuracy vs. Shift Gap $\Delta_{\text{Acc}}$ across all TABLESHIFT tasks.

(b) Percentage of Maximum In-Distribution Accuracy (PMA-ID) across all TABLESHIFT tasks.

Figure 6: Additional results.

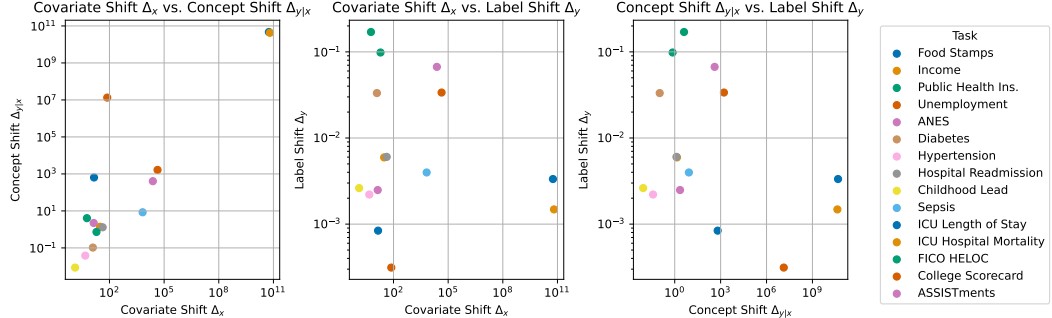

Figure 7: Pairwise scatterplots of shifts. Each point represents one dataset. Metrics are computed according to the domain split for each dataset (ID test vs. OOD test) according to the metric definitions for $\Delta_x$, $\Delta_{y|x}$, $\Delta_y$ in Section E. (a) left: Covariate shift $\Delta_x$ (computed via Optimal Transport Data Distance) vs. concept shift $\Delta_{y|x}$ (computed via Frechet Dataset Distance); $\rho = 0.99$. (b) center: Covariate shift $\Delta_x$ vs. label shift $\Delta_y$; $\rho = -0.20$. (c) left: Concept shift $\Delta_{y|x}$ vs. label shift $\Delta_y$; $\rho = -0.20$.

include the software utilities that have driven adoption of benchmarks in language and vision [50, 38], and do not contain distribution shifts (we make more detailed comparisons between TableShift and existing benchmarks in Section G). Critically, these tabular benchmarks also often lack feature-level documentation, which can be critical for tabular data.

Thus, while limited individual benchmarks do exist for tabular data modeling (without distribution shift) or for distribution shift (without tabular data), there is no existing benchmark that provides a high-quality set of tabular datasets *and* associated distribution shifts.

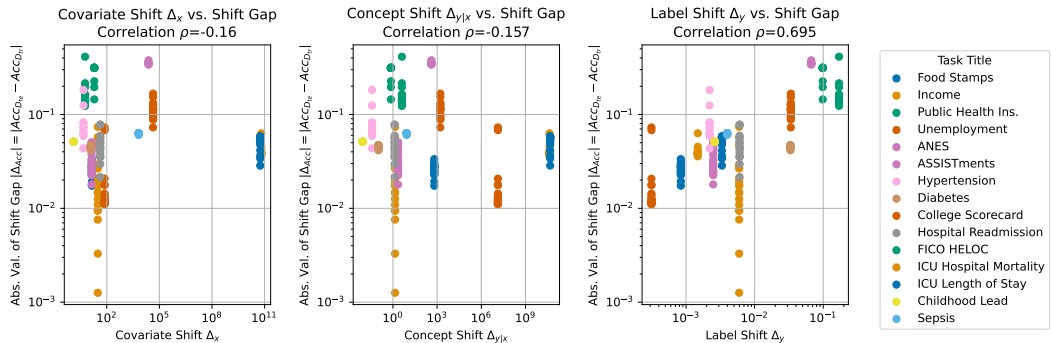

Figure 8: Domain shift metrics vs. (absolute) shift gap. Each point represents a tuned model on a given dataset. Only Label shift shows a strong correlation with shift gap ($\rho = 0.73$).

Figure 9: TableShift benchmark results, mean per model (left: non-domain generalization tasks, $\rho = 0.834$; right: domain generalization tasks, $\rho = 963$). In-domain and out-of-domain accuracy show a general linear trend. Baseline models (blue) consistently match or outperform domain robustness and domain generalization methods.

# E   Additional Dataset Details and Results

In this section we provide a brief tour of exploratory results regarding the domain shift datasets in TableShift, and additional experimental results.

## E.1   Domain Split Selection

For many tasks in TableShift, there exist clear motivations for selecting certain splitting variables, and for selecting which values of these variables to use as out-of-domain value(s) for our benchmarks. However, for oehters, there might bemultiple plausible splitting variables, or no obvious way to choose which specific value(s) to use as out of domain (e.g., any geographic region in ACS might be equallyplausible as a holdout domain for the Feed Stamps task).

For tasks where there were known domain splits that were likely to induce performance gaps that matches a real-world domain shift scenario, we began by selecting these. When tuned baselines (LightGBM and XGBoost) showed a shift gap $\Delta_{\text{Acc}}$ of at least 1%, we used that split. However, for tasks without a clear domain split or where mutliple plausible splitting values exist, we do the following. First, we identified a variable(s) that was likely to contribute to an actual shift in a real-world production through reviewing the relevant literature. Then, for each value $d \in \{d_1, \ldots d_D\} =$

Table 3: Summary of tasks in the TableShift benchmark and their associated distribution shifts.

| Task | $\Delta_x$ (Eqn. (2)) | $\Delta_{y|x}$ (Eqn (3)) | $\Delta_y$ (Eqn (3)) |
|---|---|---|---|
| **Food Stamps** | 14.20 | 640.82 | 0.0008 |
| **Income** | 30.60 | 1.40 | 0.0060 |
| **Public Health Ins.** | 5.79 | 4.06 | 0.1701 |
| **Unemployment** | 75.47 | 13,389,512.51 | 0.0003 |
| **ANES** | 13.60 | 2.23 | 0.0025 |
| **Diabetes** | 12.28 | 0.10 | 0.0332 |
| **Hypertension** | 4.69 | 0.04 | 0.0022 |
| **Hospital Readmission** | 42.37 | 1.30 | 0.0060 |
| **Childhood Lead** | 1.30 | 0.01 | 0.0026 |
| **Sepsis** | 6609.73 | 8.44 | 0.0040 |
| **ICU Length of Stay** | 56,439,324,672.00 | 47,042,729,585.25 | 0.0033 |
| **ICU Hospital Mortality** | 64,479,092,736.00 | 42,639,188,407.47 | 0.0015 |
| **FICO HELOC** | 19.35 | 0.73 | 0.0983 |
| **ASSISTments** | 24,054.59 | 1137.42 | 0.0670 |
| **College Scorecard** | 43,566.39 | 2116.63 | 0.0337 |

$\mathcal{D}$, we train on $\{\mathcal{D} \setminus d\}$ and evaluate on $d$. We select the split(s) that induced the highest performance gap in our baseline tree methods). We repeat this process for each dataset until a split that is both real-world relevant and also leads to a shift gap is found.

### E.2 Domain Shift Metrics (Covariate, Concept, and Label Shift)

As noted above, the domain shift $\Delta_{\text{Acc}}$ incurred when training a classifier is comprised of three distinct forms of shift: changes in $p(x)$ ("covariate shift"), changes in $p(y|x)$ ("concept shift"), and changes in $p(y)$ ("label shift"). It is not possible to measure the true shifts for any given dataset, because doing so would require knowing the true (ID, OOD) distributions. As a result, in order to still explore the influence of these various forms of shift on tabular data models, we propose metrics to approximately measure each form of shift.

We propose these metrics while noting that each is only an approximation of the actual degree of a certain form of shift in our dataset; measuring the actual underlying shift (e.g. the true change in $p(x)$ for covariate shift) is not possible from a finite sample. Thus, while these metrics can provide exploratory evidence of the relationship between a given type of shift (covariate, comcept, label) and model performance, they cannot provide direct evidence that any given shift type is (not) causing changes in model performance.

Table 13 gives the exact In- and Out-of-Distribution label proportions for each task, which are used to compute the label shift $\Delta_y$.

**Measuring covariate shift with OTDD**: We propose to use the following measure to approximate the degree of covariate shift between the (ID, OOD) test sets of a given task:

$$\Delta_x = \text{OTDD}(\mathcal{D}^{\text{train}}, \mathcal{D}^{\text{test}}) \tag{2}$$

where $\mathcal{D}^{\text{train}}, \mathcal{D}^{\text{test}}$ are the holdout (test) sets from the source and target domains, respectively. Here OTDD represents the Optimal Transport Dataset Distance with the Gaussian approximation as described in [2].

**Measuring concept shift with Frechet Dataset Distance (FDD):** We propose a straightforward measure of the change in $p(y|x)$ across two distributions. Inspired by measures of distributional difference widely used in the machine learning (Frechet Inception Distance, [41]) which leverage changes in the intermediate representations of a reference classifier for comparing distributions, we propose 'Frechet dataset distance" (FDD) for comparing two distributions.

This metric is computed as follows: First, we train a classifier on the source domain using the best tuned hyperparameters from our hyperparameter sweep to obtain a fixed classifier $f_\theta$. Then, for each

domain, we compute $\tilde{x} := f_{\theta[i]}(x)$, for each $x \in \mathcal{D}$, where $i$ indicates that we compute the activations at the $i^{\text{th}}$ layer of the model (this is sometimes referred to as the coding vector or feature vector for an input). Finally, we compute the Frechet dataset distance, which measures the distance between these two distributions (also called the Wasserstein-2 distance), as:

$$\text{DFD}(\mathcal{D}^{\text{train}}, \mathcal{D}^{\text{test}}) = ||\mu_{\mathcal{D}^{\text{train}}} \check{} \mu_{\mathcal{D}^{\text{test}}}||^2 + Tr(\Sigma_{\mathcal{D}^{\text{train}}} + \Sigma_{\mathcal{D}^{\text{test}}} \check{} 2 * \sqrt{\Sigma_{\mathcal{D}^{\text{train}}} * \Sigma_{\mathcal{D}^{\text{test}}}})$$

where $\mu_{\mathcal{D}}$ indicates the set of feature vectors from dimain $\mathcal{D}$ and $\Sigma_{\mathcal{D}}$ indicates the covariance matrix of $\mu_{\mathcal{D}}$. We refer to this measure as $\Delta_{y|x}$ below. A lower FDD score indicates a smaller distance between $x_i : x_i \in \mathcal{D}^{\text{train}}$ and $x_j : x_j \in \mathcal{D}^{\text{test}}$.

We parameterize the models used for FDD as MLPs. For each dataset, we use the MLP hyperparameters associated with the best validation accuracy for that model over our experiments; the model trained using these parameters is used for computing the feature activations for FDD.

**Measuring label shift:** We propose a simple measure of label shift. While label shift is clearly one factor influencing shift gaps and is perhaps the most straightforward to empirically estimate, it receives surprisingly little attention in existing literature on domain shift. We use the following measure to quantify the label shift between the source and target distributions:

$$\Delta_y = ||\bar{y}_{\mathcal{D}^{\text{train}}} - \bar{y}_{\mathcal{D}^{\text{test}}}||^2 \tag{3}$$

where $\bar{y}_{\mathcal{D}} = \frac{1}{|\mathcal{D}|} \sum_{i \in \mathcal{D}} y_i$ is the empirical sample mean of a given domain. Since all tasks in TableShift are binary classification tasks, this measures the $L_2$ difference in the base rates across the two domains.

Using these metrics, we provide one perspective on the amount of each respective form of shift in Table 3. Additionally, we provide scatter plots showing the pairwise relationships between these metrics in Figure 7, and scatter plots showing the relationship between each individual metric and the shift gap in Figure 8 (see also Figure 5 discussed in Section 5).

### E.3 Detailed Results Per Task

We provide detailed task-specific results and data in this section. In particular, we list the complete set of main results for the (In-Distribution, Out-Of-Distribution) scatter plots shown in Figures 1, 2, 3, along with the 95% Clopper-Pearson confidence intervals for these results, in Tables 4, 5, 6, 8, 9, 7, 10, 11. We also give summary metrics describing the size of each dataset split in Table 12.

Table 4: Best (In-Distribution and Out-Of-Distribution) accuracy pair observed on each benchmark task. Note that domain generalization models can only be trained on datasets with more than one training subdomain (see Table 1 for domain generalization datasets and Section 4.1 for a list of domain generalization models). ⋆: domain generalization models cannot be trained when only one training subdomain is present. See also Figures 1, 2,3. ◇: the large number of training subdomains (over 700 for ASSISTments) makes training domain generalization models impractical. □: the large dataset size makes training adversarial label DRO models impractical (since per-example gradients must be computed). We leave these experiments to future work.

| Estimator | ASSISTments | | | | Childhood Lead | | | |
|---|---|---|---|---|---|---|---|---|
| | ID Acc. (95% CI) | | OOD Acc. (95% CI) | | ID Acc. (95% CI) | | OOD Acc. (95% CI) | |
| Adv. Label DRO | □ | □ | □ | □ | 0.971 | (0.961, 0.979) | 0.92 | (0.915, 0.925) |
| CatBoost | 0.943 | (0.942, 0.944) | 0.584 | (0.562, 0.607) | 0.971 | (0.961, 0.979) | 0.92 | (0.914, 0.925) |
| DRO | 0.932 | (0.931, 0.933) | 0.583 | (0.561, 0.606) | 0.971 | (0.961, 0.979) | 0.92 | (0.915, 0.925) |
| FT-Transformer | 0.939 | (0.938, 0.94) | 0.592 | (0.569, 0.614) | 0.971 | (0.961, 0.979) | 0.92 | (0.915, 0.925) |
| Label Group DRO | 0.928 | (0.927, 0.929) | 0.574 | (0.551, 0.596) | 0.971 | (0.961, 0.979) | 0.92 | (0.915, 0.925) |
| LightGBM | 0.936 | (0.935, 0.937) | 0.591 | (0.568, 0.613) | 0.971 | (0.961, 0.979) | 0.92 | (0.915, 0.925) |
| MLP | 0.933 | (0.932, 0.934) | 0.583 | (0.561, 0.606) | 0.971 | (0.961, 0.979) | 0.92 | (0.915, 0.925) |
| NODE | 0.935 | (0.934, 0.936) | 0.583 | (0.561, 0.606) | 0.971 | (0.961, 0.979) | 0.92 | (0.915, 0.925) |
| ResNet | 0.933 | (0.932, 0.934) | 0.583 | (0.561, 0.606) | 0.971 | (0.961, 0.979) | 0.92 | (0.915, 0.925) |
| SAINT | 0.935 | (0.934, 0.936) | 0.584 | (0.562, 0.607) | 0.971 | (0.961, 0.979) | 0.92 | (0.915, 0.925) |
| TabTransformer | 0.93 | (0.929, 0.93) | 0.551 | (0.529, 0.574) | 0.971 | (0.961, 0.979) | 0.92 | (0.915, 0.925) |
| XGBoost | 0.93 | (0.929, 0.931) | 0.591 | (0.568, 0.613) | 0.971 | (0.961, 0.979) | 0.92 | (0.914, 0.925) |
| CORAL | ◇ | ◇ | ◇ | ◇ | ⋆ | ⋆ | ⋆ | ⋆ |
| DANN | ◇ | ◇ | ◇ | ◇ | ⋆ | ⋆ | ⋆ | ⋆ |
| Group DRO | ◇ | ◇ | ◇ | ◇ | ⋆ | ⋆ | ⋆ | ⋆ |
| IRM | ◇ | ◇ | ◇ | ◇ | ⋆ | ⋆ | ⋆ | ⋆ |
| MMD | ◇ | ◇ | ◇ | ◇ | ⋆ | ⋆ | ⋆ | ⋆ |
| MixUp | ◇ | ◇ | ◇ | ◇ | ⋆ | ⋆ | ⋆ | ⋆ |
| VREX | ◇ | ◇ | ◇ | ◇ | ⋆ | ⋆ | ⋆ | ⋆ |

Table 5: Best (In-Distribution and Out-Of-Distribution) accuracy pair observed on each benchmark task. Note that domain generalization models can only be trained on datasets with more than one training subdomain (see Table 1 for domain generalization datasets and Section 4.1 for a list of domain generalization models). ⋆: domain generalization models cannot be trained when only one training subdomain is present. See also Figures 1, 2,3.

| Estimator | College Scorecard | | | | Diabetes | | | |
|---|---|---|---|---|---|---|---|---|
| | ID Acc. (95% CI) | | OOD Acc. (95% CI) | | ID Acc. (95% CI) | | OOD Acc. (95% CI) | |
| Adv. Label DRO | 0.937 | (0.933, 0.942) | 0.826 | (0.805, 0.846) | 0.877 | (0.875, 0.878) | 0.832 | (0.83, 0.833) |
| CatBoost | 0.957 | (0.954, 0.961) | 0.885 | (0.866, 0.901) | 0.877 | (0.876, 0.879) | 0.833 | (0.831, 0.835) |
| DRO | 0.95 | (0.946, 0.954) | 0.862 | (0.842, 0.88) | 0.876 | (0.875, 0.878) | 0.832 | (0.83, 0.834) |
| FT-Transformer | 0.948 | (0.944, 0.952) | 0.859 | (0.839, 0.877) | 0.877 | (0.875, 0.879) | 0.832 | (0.831, 0.834) |
| Label Group DRO | 0.928 | (0.924, 0.933) | 0.817 | (0.796, 0.838) | 0.876 | (0.874, 0.878) | 0.831 | (0.83, 0.833) |
| LightGBM | 0.939 | (0.935, 0.943) | 0.822 | (0.8, 0.841) | 0.876 | (0.874, 0.878) | 0.833 | (0.831, 0.835) |
| MLP | 0.947 | (0.942, 0.95) | 0.845 | (0.825, 0.864) | 0.877 | (0.875, 0.879) | 0.832 | (0.83, 0.833) |
| NODE | 0.944 | (0.939, 0.948) | 0.844 | (0.823, 0.863) | 0.877 | (0.875, 0.879) | 0.833 | (0.832, 0.835) |
| ResNet | 0.947 | (0.943, 0.951) | 0.854 | (0.834, 0.872) | 0.874 | (0.872, 0.876) | 0.829 | (0.828, 0.831) |
| SAINT | 0.936 | (0.931, 0.94) | 0.814 | (0.792, 0.834) | 0.877 | (0.875, 0.879) | 0.833 | (0.831, 0.834) |
| TabTransformer | 0.942 | (0.938, 0.946) | 0.845 | (0.825, 0.864) | 0.875 | (0.873, 0.877) | 0.83 | (0.829, 0.832) |
| XGBoost | 0.942 | (0.938, 0.946) | 0.83 | (0.809, 0.85) | 0.877 | (0.875, 0.879) | 0.832 | (0.83, 0.834) |
| CORAL | 0.922 | (0.917, 0.926) | 0.795 | (0.773, 0.816) | 0.874 | (0.872, 0.875) | 0.832 | (0.83, 0.834) |
| DANN | 0.894 | (0.889, 0.9) | 0.78 | (0.757, 0.802) | 0.873 | (0.871, 0.875) | 0.826 | (0.824, 0.827) |
| Group DRO | 0.944 | (0.939, 0.948) | 0.829 | (0.808, 0.849) | 0.877 | (0.875, 0.879) | 0.832 | (0.83, 0.833) |
| IRM | 0.879 | (0.873, 0.885) | 0.746 | (0.721, 0.769) | 0.873 | (0.871, 0.875) | 0.826 | (0.824, 0.827) |
| MMD | 0.925 | (0.92, 0.929) | 0.795 | (0.773, 0.816) | 0.873 | (0.871, 0.875) | 0.826 | (0.825, 0.828) |
| MixUp | 0.912 | (0.907, 0.917) | 0.746 | (0.721, 0.769) | 0.873 | (0.871, 0.875) | 0.826 | (0.824, 0.827) |
| VREX | 0.907 | (0.902, 0.912) | 0.754 | (0.731, 0.777) | 0.873 | (0.871, 0.875) | 0.826 | (0.824, 0.827) |

Table 6: Best (In-Distribution and Out-Of-Distribution) accuracy pair observed on each benchmark task. Note that domain generalization models can only be trained on datasets with more than one training subdomain (see Table 1 for domain generalization datasets and Section 4.1 for a list of domain generalization models). ⋆: domain generalization models cannot be trained when only one training subdomain is present. See also Figures 1, 2,3.

| Estimator | FICO HELOC | | | | Food Stamps | | | |
|---|---|---|---|---|---|---|---|---|
| | ID Acc. (95% CI) | | OOD Acc. (95% CI) | | ID Acc. (95% CI) | | OOD Acc. (95% CI) | |
| Adv. Label DRO | 0.745 | (0.689, 0.795) | 0.431 | (0.419, 0.443) | 0.843 | (0.84, 0.846) | 0.812 | (0.808, 0.815) |
| CatBoost | 0.727 | (0.67, 0.778) | 0.582 | (0.57, 0.594) | 0.849 | (0.847, 0.852) | 0.825 | (0.821, 0.828) |
| DRO | 0.745 | (0.689, 0.795) | 0.431 | (0.419, 0.443) | 0.844 | (0.841, 0.846) | 0.819 | (0.815, 0.822) |
| FT-Transformer | 0.745 | (0.689, 0.795) | 0.431 | (0.419, 0.443) | 0.843 | (0.841, 0.846) | 0.816 | (0.812, 0.819) |
| Label Group DRO | 0.745 | (0.689, 0.795) | 0.431 | (0.419, 0.443) | 0.771 | (0.768, 0.774) | 0.752 | (0.748, 0.756) |
| LightGBM | 0.647 | (0.584, 0.7) | 0.421 | (0.409, 0.433) | 0.836 | (0.833, 0.838) | 0.808 | (0.805, 0.812) |
| MLP | 0.734 | (0.678, 0.785) | 0.538 | (0.526, 0.55) | 0.841 | (0.838, 0.844) | 0.815 | (0.812, 0.819) |
| NODE | 0.745 | (0.689, 0.795) | 0.431 | (0.419, 0.443) | 0.849 | (0.847, 0.852) | 0.822 | (0.819, 0.825) |
| ResNet | 0.748 | (0.693, 0.798) | 0.431 | (0.42, 0.443) | 0.843 | (0.84, 0.845) | 0.82 | (0.817, 0.824) |
| SAINT | 0.745 | (0.689, 0.795) | 0.431 | (0.419, 0.443) | 0.849 | (0.846, 0.851) | 0.821 | (0.818, 0.825) |
| TabTransformer | 0.745 | (0.689, 0.795) | 0.431 | (0.419, 0.443) | 0.836 | (0.834, 0.839) | 0.807 | (0.803, 0.81) |
| XGBoost | 0.745 | (0.689, 0.795) | 0.431 | (0.419, 0.443) | 0.844 | (0.842, 0.847) | 0.82 | (0.817, 0.824) |
| CORAL | ⋆ | ⋆ | ⋆ | ⋆ | 0.818 | (0.815, 0.82) | 0.793 | (0.79, 0.797) |
| DANN | ⋆ | ⋆ | ⋆ | ⋆ | 0.809 | (0.806, 0.812) | 0.78 | (0.776, 0.784) |
| Group DRO | ⋆ | ⋆ | ⋆ | ⋆ | 0.84 | (0.838, 0.843) | 0.817 | (0.814, 0.821) |
| IRM | ⋆ | ⋆ | ⋆ | ⋆ | 0.812 | (0.81, 0.815) | 0.795 | (0.791, 0.798) |
| MMD | ⋆ | ⋆ | ⋆ | ⋆ | 0.813 | (0.81, 0.816) | 0.786 | (0.782, 0.789) |
| MixUp | ⋆ | ⋆ | ⋆ | ⋆ | 0.819 | (0.816, 0.821) | 0.785 | (0.782, 0.789) |
| VREX | ⋆ | ⋆ | ⋆ | ⋆ | 0.809 | (0.806, 0.812) | 0.78 | (0.776, 0.784) |

Table 7: Best (In-Distribution and Out-Of-Distribution) accuracy pair observed on each benchmark task. Note that domain generalization models can only be trained on datasets with more than one training subdomain (see Table 1 for domain generalization datasets and Section 4.1 for a list of domain generalization models). ⋆: domain generalization models cannot be trained when only one training subdomain is present. See also Figures 1, 2,3.

| Estimator | Hospital Readmission | | | | Hypertension | | | |
|---|---|---|---|---|---|---|---|---|
| | ID Acc. (95% CI) | | OOD Acc. (95% CI) | | ID Acc. (95% CI) | | OOD Acc. (95% CI) | |
| Adv. Label DRO | 0.655 | (0.641, 0.669) | 0.603 | (0.599, 0.607) | 0.666 | (0.66, 0.672) | 0.601 | (0.6, 0.603) |
| CatBoost | 0.659 | (0.645, 0.674) | 0.618 | (0.614, 0.623) | 0.67 | (0.665, 0.676) | 0.599 | (0.597, 0.6) |
| DRO | 0.628 | (0.613, 0.642) | 0.578 | (0.574, 0.582) | 0.598 | (0.592, 0.604) | 0.416 | (0.414, 0.417) |
| FT-Transformer | 0.648 | (0.633, 0.662) | 0.618 | (0.614, 0.622) | 0.666 | (0.661, 0.672) | 0.604 | (0.603, 0.605) |
| Label Group DRO | 0.652 | (0.637, 0.666) | 0.616 | (0.612, 0.62) | 0.665 | (0.659, 0.671) | 0.604 | (0.603, 0.605) |
| LightGBM | 0.658 | (0.643, 0.672) | 0.598 | (0.594, 0.602) | 0.678 | (0.672, 0.683) | 0.634 | (0.633, 0.635) |
| MLP | 0.648 | (0.633, 0.662) | 0.612 | (0.608, 0.617) | 0.664 | (0.658, 0.67) | 0.583 | (0.582, 0.584) |
| NODE | 0.659 | (0.645, 0.673) | 0.624 | (0.62, 0.628) | 0.67 | (0.664, 0.676) | 0.597 | (0.596, 0.599) |
| ResNet | 0.639 | (0.624, 0.653) | 0.581 | (0.577, 0.586) | 0.667 | (0.661, 0.672) | 0.608 | (0.606, 0.609) |
| SAINT | 0.654 | (0.639, 0.668) | 0.61 | (0.606, 0.615) | 0.669 | (0.664, 0.675) | 0.595 | (0.594, 0.596) |
| TabTransformer | 0.584 | (0.569, 0.599) | 0.507 | (0.502, 0.511) | 0.624 | (0.618, 0.63) | 0.499 | (0.498, 0.501) |
| XGBoost | 0.651 | (0.636, 0.665) | 0.605 | (0.601, 0.61) | 0.671 | (0.665, 0.677) | 0.588 | (0.587, 0.59) |
| CORAL | 0.622 | (0.607, 0.637) | 0.571 | (0.567, 0.576) | ⋆ | ⋆ | ⋆ | ⋆ |
| DANN | 0.584 | (0.569, 0.599) | 0.506 | (0.502, 0.51) | ⋆ | ⋆ | ⋆ | ⋆ |
| Group DRO | 0.639 | (0.624, 0.653) | 0.6 | (0.596, 0.605) | ⋆ | ⋆ | ⋆ | ⋆ |
| IRM | 0.595 | (0.58, 0.61) | 0.55 | (0.546, 0.555) | ⋆ | ⋆ | ⋆ | ⋆ |
| MMD | 0.626 | (0.611, 0.64) | 0.57 | (0.565, 0.574) | ⋆ | ⋆ | ⋆ | ⋆ |
| MixUp | 0.589 | (0.574, 0.604) | 0.567 | (0.563, 0.572) | ⋆ | ⋆ | ⋆ | ⋆ |
| VREX | 0.584 | (0.569, 0.599) | 0.506 | (0.502, 0.51) | ⋆ | ⋆ | ⋆ | ⋆ |

Table 8: Best (In-Distribution and Out-Of-Distribution) accuracy pair observed on each benchmark task. Note that domain generalization models can only be trained on datasets with more than one training subdomain (see Table 1 for domain generalization datasets and Section 4.1 for a list of domain generalization models). ⋆: domain generalization models cannot be trained when only one training subdomain is present. See also Figures 1, 2,3. ♡: the large feature dimensionality of both ICU datasets makes training Transformer-based models impractical (e.g. even a single-layer SAINT model requires >13B trainable parameters on both ICU datasets)

| Estimator | ICU Hospital Mortality | | | | ICU Length of Stay | | | |
| | ID Acc. (95% CI) | | OOD Acc. (95% CI) | | ID Acc. (95% CI) | | OOD Acc. (95% CI) | |
| --- | --- | --- | --- | --- | --- | --- | --- | --- |
| Adv. Label DRO | 0.915 | (0.893, 0.931) | 0.876 | (0.87, 0.882) | 0.602 | (0.572, 0.631) | 0.544 | (0.535, 0.553) |
| CatBoost | 0.934 | (0.914, 0.948) | 0.892 | (0.887, 0.897) | 0.71 | (0.682, 0.737) | 0.674 | (0.665, 0.682) |
| DRO | 0.915 | (0.893, 0.931) | 0.876 | (0.87, 0.882) | 0.601 | (0.571, 0.63) | 0.544 | (0.535, 0.553) |
| FT-Transformer | ♡ | ♡ | ♡ | ♡ | ♡ | ♡ | ♡ | ♡ |
| Label Group DRO | 0.915 | (0.893, 0.931) | 0.876 | (0.87, 0.882) | 0.59 | (0.56, 0.619) | 0.542 | (0.533, 0.551) |
| LightGBM | 0.946 | (0.928, 0.959) | 0.883 | (0.877, 0.888) | 0.689 | (0.66, 0.716) | 0.655 | (0.646, 0.663) |
| MLP | 0.912 | (0.891, 0.929) | 0.877 | (0.871, 0.882) | 0.599 | (0.569, 0.628) | 0.544 | (0.535, 0.553) |
| NODE | 0.915 | (0.893, 0.931) | 0.876 | (0.87, 0.882) | 0.661 | (0.632, 0.689) | 0.609 | (0.6, 0.618) |
| ResNet | 0.915 | (0.893, 0.931) | 0.876 | (0.87, 0.882) | 0.606 | (0.576, 0.635) | 0.577 | (0.568, 0.586) |
| SAINT | ♡ | ♡ | ♡ | ♡ | ♡ | ♡ | ♡ | ♡ |
| TabTransformer | 0.915 | (0.893, 0.931) | 0.876 | (0.87, 0.882) | 0.604 | (0.574, 0.633) | 0.549 | (0.54, 0.558) |
| XGBoost | 0.927 | (0.908, 0.943) | 0.882 | (0.876, 0.887) | 0.71 | (0.682, 0.737) | 0.669 | (0.66, 0.677) |
| CORAL | 0.915 | (0.893, 0.931) | 0.875 | (0.869, 0.881) | 0.603 | (0.573, 0.632) | 0.544 | (0.535, 0.553) |
| DANN | 0.915 | (0.893, 0.931) | 0.876 | (0.871, 0.882) | 0.594 | (0.564, 0.624) | 0.545 | (0.536, 0.554) |
| Group DRO | 0.915 | (0.893, 0.931) | 0.876 | (0.87, 0.882) | 0.602 | (0.572, 0.631) | 0.544 | (0.535, 0.553) |
| IRM | 0.915 | (0.893, 0.931) | 0.876 | (0.87, 0.882) | 0.601 | (0.571, 0.63) | 0.544 | (0.535, 0.553) |
| MMD | 0.915 | (0.893, 0.931) | 0.876 | (0.87, 0.882) | 0.602 | (0.572, 0.631) | 0.544 | (0.535, 0.553) |
| MixUp | 0.915 | (0.893, 0.931) | 0.876 | (0.87, 0.882) | 0.602 | (0.572, 0.631) | 0.544 | (0.535, 0.553) |
| VREX | 0.913 | (0.893, 0.931) | 0.876 | (0.87, 0.882) | 0.597 | (0.567, 0.627) | 0.545 | (0.536, 0.554) |

Table 9: Best (In-Distribution and Out-Of-Distribution) accuracy pair observed on each benchmark task. Note that domain generalization models can only be trained on datasets with more than one training subdomain (see Table 1 for domain generalization datasets and Section 4.1 for a list of domain generalization models). ⋆: domain generalization models cannot be trained when only one training subdomain is present. □: the large dataset size makes training adversarial label DRO models impractical (since per-example gradients must be computed). We leave these experiments to future work. See also Figures 1, 2,3.

| Estimator | Income | | | | Public Health Ins. | | | |
|---|---|---|---|---|---|---|---|---|
| | ID Acc. (95% CI) | | OOD Acc. (95% CI) | | ID Acc. (95% CI) | | OOD Acc. (95% CI) | |
| Adv. Label DRO | 0.829 | (0.827, 0.83) | 0.819 | (0.816, 0.822) | □ | □ | □ | □ |
| CatBoost | 0.832 | (0.83, 0.834) | 0.814 | (0.811, 0.817) | 0.814 | (0.812, 0.815) | 0.69 | (0.689, 0.691) |
| DRO | 0.828 | (0.826, 0.83) | 0.818 | (0.816, 0.821) | 0.809 | (0.808, 0.81) | 0.647 | (0.646, 0.648) |
| FT-Transformer | 0.825 | (0.823, 0.827) | 0.818 | (0.815, 0.821) | 0.807 | (0.806, 0.808) | 0.662 | (0.661, 0.663) |
| Label Group DRO | 0.819 | (0.817, 0.821) | 0.818 | (0.815, 0.821) | 0.776 | (0.775, 0.777) | 0.364 | (0.363, 0.365) |
| LightGBM | 0.822 | (0.82, 0.824) | 0.809 | (0.806, 0.812) | 0.803 | (0.802, 0.804) | 0.639 | (0.638, 0.64) |
| MLP | 0.828 | (0.826, 0.829) | 0.813 | (0.81, 0.816) | 0.808 | (0.806, 0.809) | 0.612 | (0.611, 0.613) |
| NODE | 0.831 | (0.829, 0.833) | 0.81 | (0.807, 0.813) | 0.811 | (0.81, 0.812) | 0.662 | (0.661, 0.663) |
| ResNet | 0.826 | (0.824, 0.828) | 0.815 | (0.812, 0.818) | 0.81 | (0.809, 0.811) | 0.672 | (0.671, 0.673) |
| SAINT | 0.829 | (0.827, 0.831) | 0.81 | (0.807, 0.812) | 0.811 | (0.81, 0.812) | 0.68 | (0.679, 0.681) |
| TabTransformer | 0.818 | (0.816, 0.82) | 0.801 | (0.798, 0.804) | 0.803 | (0.802, 0.804) | 0.588 | (0.587, 0.589) |
| XGBoost | 0.821 | (0.819, 0.823) | 0.792 | (0.789, 0.795) | 0.805 | (0.804, 0.806) | 0.661 | (0.66, 0.662) |
| CORAL | 0.817 | (0.815, 0.819) | 0.791 | (0.788, 0.793) | ⋆ | ⋆ | ⋆ | ⋆ |
| DANN | 0.815 | (0.813, 0.817) | 0.812 | (0.809, 0.815) | ⋆ | ⋆ | ⋆ | ⋆ |
| Group DRO | 0.827 | (0.826, 0.829) | 0.813 | (0.81, 0.815) | ⋆ | ⋆ | ⋆ | ⋆ |
| IRM | 0.756 | (0.754, 0.758) | 0.699 | (0.696, 0.702) | ⋆ | ⋆ | ⋆ | ⋆ |
| MMD | 0.816 | (0.814, 0.818) | 0.768 | (0.765, 0.771) | ⋆ | ⋆ | ⋆ | ⋆ |
| MixUp | 0.821 | (0.819, 0.823) | 0.794 | (0.791, 0.797) | ⋆ | ⋆ | ⋆ | ⋆ |
| VREX | 0.714 | (0.712, 0.716) | 0.64 | (0.637, 0.644) | ⋆ | ⋆ | ⋆ | ⋆ |

Table 10: Best (In-Distribution and Out-Of-Distribution) accuracy pair observed on each benchmark task. Note that domain generalization models can only be trained on datasets with more than one training subdomain (see Table 1 for domain generalization datasets and Section 4.1 for a list of domain generalization models). ⋆: domain generalization models cannot be trained when only one training subdomain is present. See also Figures 1, 2,3.

| Estimator | Sepsis | | | | Unemployment | | | |
|---|---|---|---|---|---|---|---|---|
| | ID Acc. (95% CI) | | OOD Acc. (95% CI) | | ID Acc. (95% CI) | | OOD Acc. (95% CI) | |
| Adv. Label DRO | 0.988 | (0.987, 0.989) | 0.925 | (0.924, 0.926) | 0.972 | (0.971, 0.973) | 0.96 | (0.959, 0.961) |
| CatBoost | 0.988 | (0.987, 0.989) | 0.925 | (0.923, 0.926) | 0.973 | (0.973, 0.974) | 0.962 | (0.961, 0.963) |
| DRO | 0.988 | (0.987, 0.989) | 0.925 | (0.924, 0.926) | 0.973 | (0.972, 0.973) | 0.961 | (0.96, 0.962) |
| FT-Transformer | 0.988 | (0.987, 0.989) | 0.925 | (0.924, 0.926) | 0.973 | (0.972, 0.974) | 0.962 | (0.961, 0.962) |
| Label Group DRO | 0.988 | (0.987, 0.989) | 0.925 | (0.924, 0.926) | 0.947 | (0.946, 0.948) | 0.926 | (0.925, 0.927) |
| LightGBM | 0.988 | (0.987, 0.989) | 0.928 | (0.926, 0.929) | 0.973 | (0.972, 0.974) | 0.96 | (0.96, 0.961) |
| MLP | 0.988 | (0.987, 0.989) | 0.925 | (0.923, 0.926) | 0.973 | (0.972, 0.973) | 0.96 | (0.959, 0.961) |
| NODE | 0.988 | (0.987, 0.989) | 0.925 | (0.924, 0.926) | 0.973 | (0.972, 0.974) | 0.962 | (0.961, 0.963) |
| ResNet | 0.988 | (0.987, 0.989) | 0.925 | (0.924, 0.926) | 0.972 | (0.971, 0.972) | 0.959 | (0.958, 0.96) |
| SAINT | 0.988 | (0.987, 0.989) | 0.925 | (0.924, 0.926) | 0.973 | (0.972, 0.974) | 0.962 | (0.961, 0.963) |
| TabTransformer | 0.988 | (0.987, 0.989) | 0.925 | (0.924, 0.926) | 0.972 | (0.971, 0.973) | 0.961 | (0.96, 0.962) |
| XGBoost | 0.988 | (0.987, 0.989) | 0.925 | (0.923, 0.926) | 0.973 | (0.972, 0.973) | 0.961 | (0.961, 0.962) |
| CORAL | ⋆ | ⋆ | ⋆ | ⋆ | 0.964 | (0.963, 0.965) | 0.95 | (0.949, 0.951) |
| DANN | ⋆ | ⋆ | ⋆ | ⋆ | 0.966 | (0.965, 0.967) | 0.948 | (0.947, 0.95) |
| Group DRO | ⋆ | ⋆ | ⋆ | ⋆ | 0.971 | (0.97, 0.972) | 0.958 | (0.957, 0.959) |
| IRM | ⋆ | ⋆ | ⋆ | ⋆ | 0.966 | (0.965, 0.967) | 0.948 | (0.947, 0.95) |
| MMD | ⋆ | ⋆ | ⋆ | ⋆ | 0.966 | (0.966, 0.967) | 0.953 | (0.952, 0.954) |
| MixUp | ⋆ | ⋆ | ⋆ | ⋆ | 0.844 | (0.842, 0.846) | 0.776 | (0.774, 0.778) |
| VREX | ⋆ | ⋆ | ⋆ | ⋆ | 0.873 | (0.871, 0.874) | 0.8 | (0.798, 0.802) |

Table 11: Best (In-Distribution and Out-Of-Distribution) accuracy pair observed on each benchmark task. See also Figures 1, 2,3.

| Estimator | Voting | | | |
|---|---|---|---|---|
| | ID Acc. (95% CI) | | OOD Acc. (95% CI) | |
| Adv. Label DRO | 0.875 | (0.843, 0.902) | 0.852 | (0.839, 0.865) |
| CatBoost | 0.883 | (0.852, 0.909) | 0.855 | (0.842, 0.868) |
| DRO | 0.881 | (0.85, 0.907) | 0.853 | (0.839, 0.866) |
| FT-Transformer | 0.879 | (0.848, 0.906) | 0.855 | (0.841, 0.868) |
| Label Group DRO | 0.862 | (0.829, 0.89) | 0.839 | (0.825, 0.852) |
| LightGBM | 0.881 | (0.85, 0.907) | 0.855 | (0.841, 0.868) |
| MLP | 0.892 | (0.862, 0.918) | 0.86 | (0.847, 0.873) |
| NODE | 0.885 | (0.854, 0.911) | 0.851 | (0.838, 0.864) |
| ResNet | 0.887 | (0.856, 0.912) | 0.836 | (0.822, 0.849) |
| SAINT | 0.888 | (0.858, 0.914) | 0.858 | (0.845, 0.871) |
| TabTransformer | 0.877 | (0.846, 0.904) | 0.859 | (0.845, 0.872) |
| XGBoost | 0.898 | (0.869, 0.923) | 0.851 | (0.838, 0.864) |
| CORAL | 0.883 | (0.852, 0.909) | 0.846 | (0.832, 0.859) |
| DANN | 0.892 | (0.862, 0.918) | 0.852 | (0.838, 0.865) |
| Group DRO | 0.877 | (0.846, 0.904) | 0.852 | (0.839, 0.865) |
| IRM | 0.804 | (0.767, 0.837) | 0.758 | (0.742, 0.774) |
| MMD | 0.892 | (0.862, 0.918) | 0.849 | (0.835, 0.862) |
| MixUp | 0.892 | (0.862, 0.918) | 0.851 | (0.837, 0.864) |
| VREX | 0.804 | (0.767, 0.837) | 0.754 | (0.737, 0.77) |

Table 12: Sample sizes by split. In particular, large *test* sizes are desirable for benchmarking, as they reduce the statistical uncertainty of comparing model performance.

| Task | ID Test | OOD Test | OOD Validation | Train | Validation | Total |
|---|---|---|---|---|---|---|
| **Food Stamps** | 78,628 | 48,878 | 5431 | 629,018 | 78,627 | 840,582 |
| **Income** | 158,016 | 75,911 | 8435 | 1,264,123 | 158,015 | 1,664,500 |
| **Public Coverage** | 500,782 | 817,877 | 90,876 | 4,006,249 | 500,781 | 5,916,565 |
| **Unemployment** | 161,365 | 163,611 | 18,180 | 1,290,914 | 161,364 | 1,795,434 |
| **Voting** | 520 | 2772 | 309 | 4159 | 520 | 8280 |
| **Hypertension** | 27,052 | 518,622 | 57,625 | 216,411 | 27,051 | 846,761 |
| **Diabetes** | 121,154 | 209,375 | 23,264 | 969,229 | 121,154 | 1,444,176 |
| **Readmission** | 4287 | 50,968 | 5664 | 34,288 | 4286 | 99,493 |
| **HELOC** | 278 | 6914 | 769 | 2220 | 278 | 10,459 |
| **ICU Length of Stay** | 1080 | 11,835 | 1316 | 8634 | 1079 | 23,944 |
| **ICU Hospital Mortality** | 890 | 13,544 | 1505 | 7116 | 889 | 23,944 |
| **Sepsis** | 140,288 | 134,402 | 14,934 | 1,122,299 | 140,287 | 1,552,210 |
| **Childhood Lead** | 1476 | 11,466 | 1274 | 11,807 | 1476 | 27,499 |
| **ASSISTments** | 266,566 | 1906 | 212 | 2,132,526 | 266,566 | 2,667,776 |
| **College Scorecard** | 12,320 | 1352 | 151 | 98,556 | 12,320 | 124,699 |

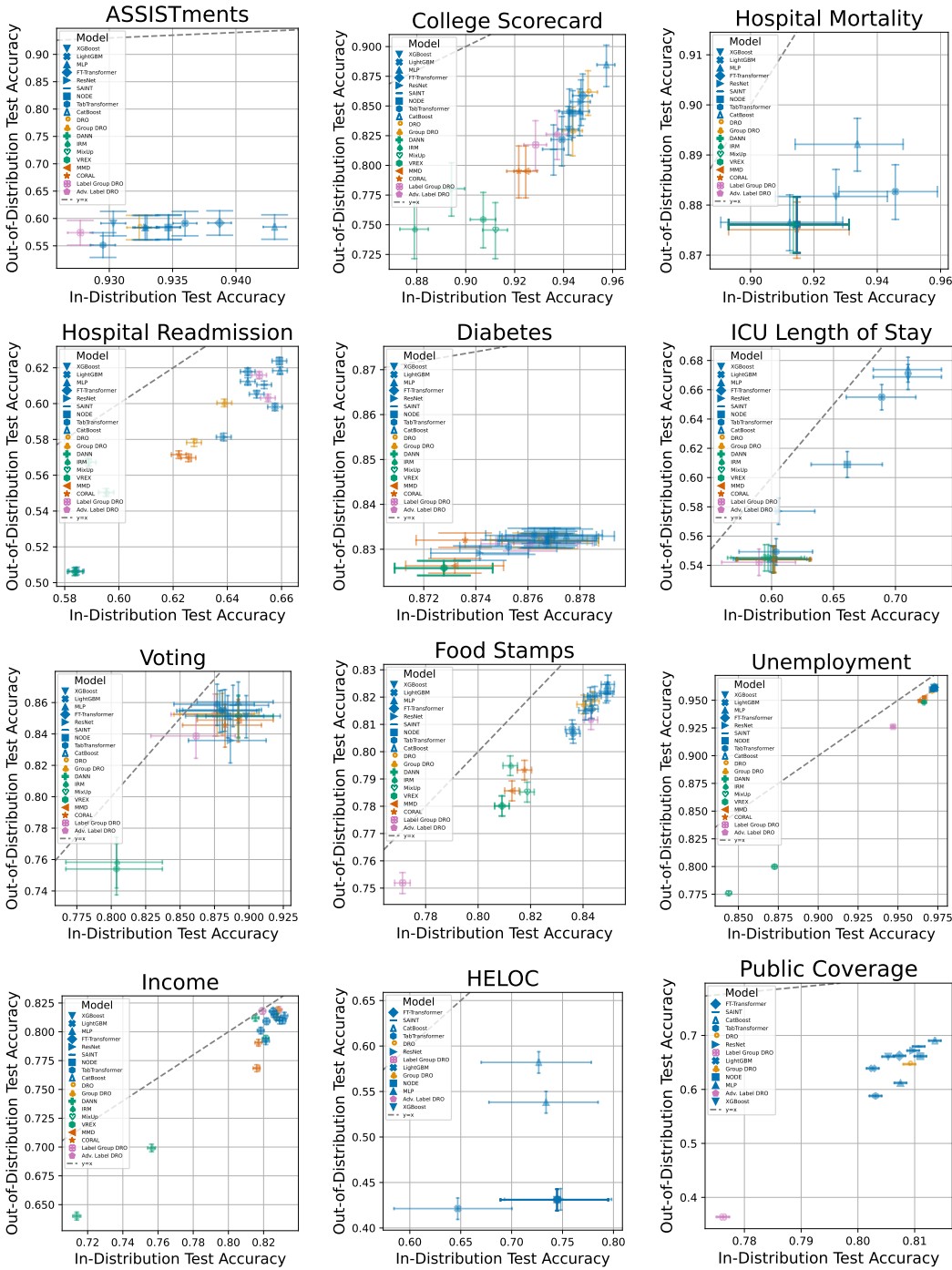

Figure 10: Alternate version of Figure 2 with adjusted scaling for increased detail.

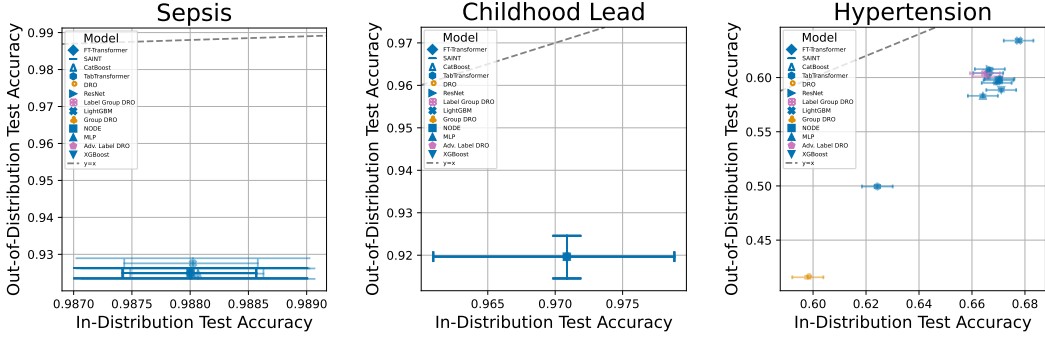

Figure 11: Alternate version of Figure 3 with adjusted scaling for increased detail.

Table 13: Label summary statistics for test sets in TABLESHIFT. $\bar{Y}$ gives the proportion of positive labels for the respective split, and $\text{Var}(\bar{Y})$ the variance of the sample proportion.

| | ID | | OOD | |
| Task | $\bar{Y}_{\text{ID}}$ | $\text{Var}(\bar{Y}_{\text{ID}})$ | $\bar{Y}_{\text{OOD}}$ | $\text{Var}(\bar{Y}_{\text{OOD}})$ |
|---|---|---|---|---|
| **Voting** | 0.804 | 0.017 | 0.754 | 0.008 |
| **ASSISTments** | 0.695 | 0.001 | 0.437 | 0.011 |
| **Childhood Lead** | 0.029 | 0.004 | 0.080 | 0.003 |
| **College Scorecard** | 0.127 | 0.003 | 0.311 | 0.013 |
| **Diabetes** | 0.127 | 0.001 | 0.174 | 0.001 |
| **FICO HELOC** | 0.255 | 0.026 | 0.569 | 0.006 |
| **Food Stamps** | 0.191 | 0.001 | 0.220 | 0.002 |
| **Hospital Readmission** | 0.416 | 0.008 | 0.494 | 0.002 |
| **Hypertension** | 0.402 | 0.003 | 0.584 | 0.001 |
| **ICU Hospital Mortality** | 0.085 | 0.009 | 0.124 | 0.003 |
| **ICU Length of Stay** | 0.398 | 0.015 | 0.456 | 0.005 |
| **Income** | 0.321 | 0.001 | 0.398 | 0.002 |
| **Public Health Ins.** | 0.224 | 0.001 | 0.636 | 0.001 |
| **Sepsis** | 0.012 | 0.000 | 0.075 | 0.001 |
| **Unemployment** | 0.034 | 0.000 | 0.052 | 0.001 |

Table 14: PMA-OOD results (cf. Figure ) and standard deviation of PMA-OOD over benchmark tasks. (Cf. Figure 4a.)

| Estimator | PMA-OOD Mean | PMA-OOD Std. |
|---|---|---|
| VREX | 0.876 | 0.076 |
| IRM | 0.910 | 0.069 |
| Label Group DRO | 0.915 | 0.129 |
| MixUp | 0.917 | 0.076 |
| TabTransformer | 0.922 | 0.091 |
| DANN | 0.932 | 0.075 |
| DRO | 0.932 | 0.107 |
| MMD | 0.940 | 0.059 |
| CORAL | 0.944 | 0.059 |
| Adv. Label DRO | 0.950 | 0.080 |
| ResNet | 0.956 | 0.069 |
| MLP | 0.961 | 0.054 |
| Group DRO | 0.962 | 0.059 |
| NODE | 0.963 | 0.066 |
| SAINT | 0.964 | 0.070 |
| LightGBM | 0.964 | 0.070 |
| XGBoost | 0.964 | 0.065 |
| FT-Transformer | 0.968 | 0.069 |
| CatBoost | 0.994 | 0.014 |

Table 15: Label summary statistics for test sets in TABLESHIFT. $\bar{Y}$ gives the proportion of positive labels for the respective split, and $\mathrm{Var}(\bar{Y})$ the variance of the sample proportion. (Cf. Figure 4b.)

| Method | ID | | OOD | |
| --- | --- | --- | --- | --- |
| | ID Accuracy | Std. | OOD Accuracy | Std. |
| Adv. Label DRO | 0.834 | 0.125 | 0.792 | 0.132 |
| CatBoost | 0.862 | 0.105 | 0.794 | 0.126 |
| DANN | 0.816 | 0.137 | 0.770 | 0.148 |
| CORAL | 0.824 | 0.129 | 0.777 | 0.134 |
| DRO | 0.843 | 0.129 | 0.773 | 0.147 |
| FT-Transformer | 0.866 | 0.103 | 0.794 | 0.126 |
| Group DRO | 0.832 | 0.129 | 0.791 | 0.133 |
| IRM | 0.800 | 0.130 | 0.749 | 0.136 |
| Label Group DRO | 0.829 | 0.123 | 0.759 | 0.134 |
| LightGBM | 0.856 | 0.108 | 0.781 | 0.124 |
| MixUp | 0.807 | 0.125 | 0.752 | 0.118 |
| MLP | 0.845 | 0.126 | 0.774 | 0.141 |
| MMD | 0.825 | 0.130 | 0.774 | 0.135 |
| NODE | 0.853 | 0.110 | 0.781 | 0.129 |
| ResNet | 0.844 | 0.126 | 0.773 | 0.139 |
| SAINT | 0.868 | 0.099 | 0.787 | 0.127 |
| TabTransformer | 0.835 | 0.136 | 0.759 | 0.160 |
| VREX | 0.786 | 0.127 | 0.720 | 0.128 |
| XGBoost | 0.857 | 0.105 | 0.783 | 0.122 |

### E.4 Results with Additional Random Seeds

Our experiments on each model-dataset pair comprise a single run of 100 rounds of our hyperparameter tuning protocol described in Section 4.2. Here, we provide the results of additional experiments conducted using different random seeds, in order to evaluate the sensitivity of our results to the random variation inherent in the training and hyperparameter tuning process.

For these experiments, we conduct an identical procedure to the experiments described in the main text of our paper, but only change the random seed. This process affects the random initialization of model weights, random initialization of hyperparameter tuning, and training data shuffling, among other procedures. We note that it does *not* affect the train/test splitting in our datasets, as the train/test splits are defined by distribution shifts and are fixed to ensure comparability of the benchmark across experiments.

The results are shown in Table 16. Table 16 shows that, across the five models and three datasets evaluated, there is minimal variation in performance due to random seeds. Of the 90 measurements covering 45 trials represented in Table 16, the 95% Clopper-Pearson CIs for both ID and OOD accuracy overlap in all cases, with only four exceptions (LightGBM, iteration 0, Food Stamps ID and OOD accuracy; LightGBM, iteration 2, Hypertension OOD accuracy; MLP, Hypertension, iteration 0, OOD accuracy; FT-Transformer, iteration 0, OOD accuracy). These results provide evidence that our results are robust to variation due to random seed.

### E.5 Results with Hybrid Methods

Our main study design is focused on benchmarking existing previously-proposed methods for tabular modeling. The methods we evlauate span *models*, which prescribe the functional form of a predictor $f_\theta$, and also *objective functions*, which describe the loss $\mathcal{L}$ to be minimized while learning the parameters $\theta$ of a fixed predictor $f$. Concretely, for example, FT-Transformer or MLP specify the form of $f$, while some robustness interventions, such as Group DRO, specify an objective that can be monimized over any smooth continuous function.

Our study does not explore potential combinations of different models and objective functions from the preexisting literature. In this section, we conduct an exploratory investigation into whether "hybrid methods" – combinations of different models and objective functions explored in our study – might improve robustness, for the best-performing compatible combinations of models and objective functions in our study.

In particular, our hybrid model study explores the use of the Group DRO objective function, in combination with three models from our study: FT-Transformer, NODE, and ResNet. Group DRO was selected as it is the highest-performing objective-based technique in our study (see Figure 4a), and the three models were selected as they are the highest-performing Transformer-based model, tree-based model, and baseline supervised model, repsectively, in our study. We note that Group DRO cannot be easily combined with CatBoost, XGBoost, or LightGBM, as these are not smooth differentiable continuous functions, which is a requirement for the use of the Group DRO objective.

Our methodology in this section is as follows: for each estimator (FT-Transformer, NODE, ResNet), we train the model with both ERM (the standard procedure used in our main experiments above) and Group DRO. We follow the same hyperparameter tuning procedure as described in Section 4.2 above. We use the same hyperparameter grid defined in Section I for each model, but also include a full sweep over the Group DRO step size parameter, using the Group DRO grid described in Section I (thus, for model X, we take the union of the two hyperparameter grids: { grid(X) ∪ grid(Group DRO) } ). We conduct this procedure for five benchmark datasets: Childhood Lead, College Scorecard, Food Stamps, Hypertension, and Voting.

The results of our hybrid model experiments are shown in Table 17. The results show little or no evidence that Group DRO reduces shift gaps for the models evaluated, as indicated by the fact that OOD test accuracy intervals tend to be overlapping, or *higher*, for ERM relative to Group DRO. Keeping in mind that Group DRO was parameterized over MLP models in our main experiments (as all prior works only use Group DRO with MLP), the results in Table 17 suggest that Group DRO may primarily improve weak (MLP) models but does not improve robustness for stronger models, explaining the improvements for Group DRO over vanilla MLP models in the main text.

Table 16: Results with additional random seeds. Varying random seeds has a minimal impact on the final results of our hyperparameter tuning procedure, indicating that our findings are robust to variation due to random seeds. See Section E.4 for details on experimental design.

| Task | Base Estimator | Iteration | ID Test Accuracy Value | ID Test Accuracy 95% CI | OOD Test Accuracy Value | OOD Test Accuracy 95% CI |
|---|---|---|---|---|---|---|
| College Scorecard | CatBoost | 0 | 0.957 | (0.954, 0.961) | 0.885 | (0.866, 0.901) |
| | | 1 | 0.959 | (0.955, 0.962) | 0.879 | (0.861, 0.896) |
| | | 2 | 0.959 | (0.956, 0.963) | 0.882 | (0.863, 0.898) |
| | FT-Transformer | 0 | 0.948 | (0.944, 0.952) | 0.859 | (0.839, 0.877) |
| | | 1 | 0.946 | (0.942, 0.95) | 0.850 | (0.83, 0.868) |
| | | 2 | 0.940 | (0.936, 0.945) | 0.830 | (0.809, 0.85) |
| | LightGBM | 0 | 0.939 | (0.935, 0.943) | 0.822 | (0.8, 0.841) |
| | | 1 | 0.943 | (0.938, 0.947) | 0.839 | (0.819, 0.859) |
| | | 2 | 0.943 | (0.939, 0.947) | 0.837 | (0.816, 0.856) |
| | MLP | 0 | 0.947 | (0.942, 0.95) | 0.845 | (0.825, 0.864) |
| | | 1 | 0.949 | (0.944, 0.952) | 0.859 | (0.84, 0.878) |
| | | 2 | 0.945 | (0.941, 0.949) | 0.859 | (0.839, 0.877) |
| | XGBoost | 0 | 0.942 | (0.938, 0.946) | 0.830 | (0.809, 0.85) |
| | | 1 | 0.946 | (0.942, 0.95) | 0.842 | (0.821, 0.861) |
| | | 2 | 0.947 | (0.943, 0.951) | 0.845 | (0.824, 0.864) |
| Food Stamps | CatBoost | 0 | 0.849 | (0.847, 0.852) | 0.825 | (0.821, 0.828) |
| | | 1 | 0.850 | (0.847, 0.852) | 0.824 | (0.821, 0.827) |
| | | 2 | 0.849 | (0.847, 0.852) | 0.824 | (0.82, 0.827) |
| | FT-Transformer | 0 | 0.843 | (0.841, 0.846) | 0.816 | (0.812, 0.819) |
| | | 1 | 0.848 | (0.846, 0.851) | 0.824 | (0.82, 0.827) |
| | | 2 | 0.844 | (0.842, 0.847) | 0.817 | (0.814, 0.82) |
| | LightGBM | 0 | 0.836 | (0.833, 0.838) | 0.808 | (0.805, 0.812) |
| | | 1 | 0.844 | (0.841, 0.846) | 0.818 | (0.814, 0.821) |
| | | 2 | 0.843 | (0.84, 0.846) | 0.817 | (0.814, 0.821) |
| | MLP | 0 | 0.841 | (0.838, 0.844) | 0.815 | (0.812, 0.819) |
| | | 1 | 0.845 | (0.842, 0.847) | 0.817 | (0.814, 0.821) |
| | | 2 | 0.844 | (0.841, 0.846) | 0.811 | (0.808, 0.815) |
| | XGBoost | 0 | 0.844 | (0.842, 0.847) | 0.820 | (0.817, 0.824) |
| | | 1 | 0.843 | (0.84, 0.845) | 0.819 | (0.815, 0.822) |
| | | 2 | 0.845 | (0.842, 0.847) | 0.820 | (0.816, 0.823) |
| Hypertension | CatBoost | 0 | 0.670 | (0.665, 0.676) | 0.599 | (0.597, 0.6) |
| | | 1 | 0.671 | (0.665, 0.676) | 0.599 | (0.597, 0.6) |
| | | 2 | 0.671 | (0.666, 0.677) | 0.600 | (0.598, 0.601) |
| | FT-Transformer | 0 | 0.666 | (0.661, 0.672) | 0.604 | (0.603, 0.605) |
| | | 1 | 0.670 | (0.665, 0.676) | 0.594 | (0.593, 0.596) |
| | | 2 | 0.672 | (0.666, 0.677) | 0.595 | (0.594, 0.596) |
| | LightGBM | 0 | 0.678 | (0.672, 0.683) | 0.634 | (0.633, 0.635) |
| | | 1 | 0.672 | (0.666, 0.677) | 0.636 | (0.635, 0.637) |
| | | 2 | 0.672 | (0.667, 0.678) | 0.628 | (0.627, 0.629) |
| | MLP | 0 | 0.664 | (0.658, 0.67) | 0.583 | (0.582, 0.584) |
| | | 1 | 0.669 | (0.663, 0.674) | 0.597 | (0.596, 0.599) |
| | | 2 | 0.668 | (0.662, 0.673) | 0.598 | (0.597, 0.599) |
| | XGBoost | 0 | 0.671 | (0.665, 0.677) | 0.588 | (0.587, 0.59) |
| | | 1 | 0.669 | (0.664, 0.675) | 0.586 | (0.584, 0.587) |
| | | 2 | 0.669 | (0.664, 0.675) | 0.586 | (0.584, 0.587) |

Table 17: Hybrid method results. We compare Group DRO to standard ERM for the highest-performing Transformer, tree-based, and baseline models in our study (FT-Transformer, NODE, and ResNet, respectively) over five TABLESHIFT tasks, following our hyperparameter tuning procedure. There is little or no evidence that Group DRO reduces shift gaps for these models, indicating that Group DRO may primarily improve weak (MLP) models but does not improve robustness for stronger models. See Section E.5 for details on experimental design.

| Task | Base Estimator | Method | ID Test Accuracy | | OOD Test Accuracy | |
| | | | Value | 95% CI | Value | 95% CI |
|---|---|---|---|---|---|---|
| Childhood Lead | FT-Transformer | ERM | 0.971 | (0.961, 0.979) | 0.920 | (0.915, 0.925) |
| | | Group DRO | 0.971 | (0.961, 0.979) | 0.920 | (0.915, 0.925) |
| | NODE | ERM | 0.971 | (0.961, 0.979) | 0.920 | (0.915, 0.925) |
| | | Group DRO | 0.971 | (0.961, 0.979) | 0.920 | (0.915, 0.925) |
| | ResNet | ERM | 0.971 | (0.961, 0.979) | 0.920 | (0.915, 0.925) |
| | | Group DRO | 0.971 | (0.961, 0.979) | 0.920 | (0.915, 0.925) |
| College Scorecard | FT-Transformer | ERM | 0.948 | (0.944, 0.952) | 0.859 | (0.839, 0.877) |
| | | Group DRO | 0.935 | (0.93, 0.939) | 0.815 | (0.793, 0.835) |
| | NODE | ERM | 0.944 | (0.939, 0.948) | 0.844 | (0.823, 0.863) |
| | | Group DRO | 0.946 | (0.942, 0.95) | 0.835 | (0.814, 0.854) |
| | ResNet | ERM | 0.947 | (0.943, 0.951) | 0.854 | (0.834, 0.872) |
| | | Group DRO | 0.947 | (0.942, 0.95) | 0.824 | (0.803, 0.844) |
| Food Stamps | FT-Transformer | ERM | 0.843 | (0.841, 0.846) | 0.816 | (0.812, 0.819) |
| | | Group DRO | 0.826 | (0.823, 0.829) | 0.795 | (0.792, 0.799) |
| | NODE | ERM | 0.849 | (0.847, 0.852) | 0.822 | (0.819, 0.825) |
| | | Group DRO | 0.845 | (0.842, 0.847) | 0.822 | (0.819, 0.825) |
| | ResNet | ERM | 0.843 | (0.84, 0.845) | 0.820 | (0.817, 0.824) |
| | | Group DRO | 0.848 | (0.846, 0.851) | 0.818 | (0.815, 0.822) |
| Hypertension | FT-Transformer | ERM | 0.666 | (0.661, 0.672) | 0.604 | (0.603, 0.605) |
| | | Group DRO | 0.665 | (0.659, 0.67) | 0.608 | (0.607, 0.609) |
| | NODE | ERM | 0.670 | (0.664, 0.676) | 0.597 | (0.596, 0.599) |
| | | Group DRO | 0.671 | (0.665, 0.676) | 0.592 | (0.591, 0.593) |
| | ResNet | ERM | 0.667 | (0.661, 0.672) | 0.608 | (0.606, 0.609) |
| | | Group DRO | 0.663 | (0.658, 0.669) | 0.590 | (0.589, 0.592) |
| Voting | FT-Transformer | ERM | 0.879 | (0.848, 0.906) | 0.855 | (0.841, 0.868) |
| | | Group DRO | 0.894 | (0.865, 0.919) | 0.858 | (0.844, 0.87) |
| | NODE | ERM | 0.885 | (0.854, 0.911) | 0.851 | (0.838, 0.864) |
| | | Group DRO | 0.898 | (0.869, 0.923) | 0.860 | (0.847, 0.873) |
| | ResNet | ERM | 0.887 | (0.856, 0.912) | 0.836 | (0.822, 0.849) |
| | | Group DRO | 0.898 | (0.869, 0.923) | 0.847 | (0.833, 0.861) |

# F   Model Details

This section describes the models used in our study. For the hyperparameters used in our experiments, see Section I.

Our implementations of these models, along with associated code to train models with fixed hyperparameters or to tune hyperparameters at scale via the Ray framework, are available at https://github.com/mlfoundations/tableshift.

## F.1   Baseline Models

**XGBoost:** XGBoost is a popular library for learning gradient-boosted trees. We use the original XGBoost implementation [20]. XGBoost introduced column subsampling, weight regularization, and introduced major improvements in efficiency for training gradient boosted models on large or out-of-core datasets.

**LightGBM:** LightGBM is a library for learning gradient-boosted trees which extends the success of XGBoost in working fast and with large datasets [48]. LightGBM introduces novel techniques such as converting continuous features to histograms (for computational efficiency and for to reduce overfitting), combining certain features using Exclusive Feature Bundling (EFB), and through the use of Gradient-based One-Side Sampling (GOSS).

**CatBoost:** CatBoost [25] is a library for learning gradient-boosted trees which includes novel techniques for leveraging categorical features. This includes heuristics to replace numeric or one-hot encoding of categorical features with label-derived heuristics; "appearance" (count) features for categorical features; and efficient greedy feature recombination techniques.

**MLP:** We use standard multilayer perceptrons, via the implementation in RTDL[29]. MLPs have been shown to be highly effective models for tabular data, particularly when a large model search space is used and regularization is carefully tuned [46].

## F.2   Tabular Neural Networks

**FT-Transformer:** FT-Transformer is a transformer-based model that learns separate feature tokenizers for numeric and categorical data, and applies a transformer model [85] to the tokenized features.

**Tabular ResNet:** We use the version of Tabular ResNet proposed in [36]. We note that, despite the fact that this approach is shown to have competitive performance with many existing tabular data models in [36], it has not been widely used in the literature.

**NODE** Neural Oblivious Decision Ensembles (NODE) [70] is a method that leverages oblivious ensembling methods to train "tree-like" neural networks.

**TabTransformer:** TabTransformers [43] is a model that uses learned embeddings of categorical features, which are then passed through standard Transformer layers, alongside layer normalization of continuous features.

**SAINT:** SAINT [79] uses an enhanced embedding method for categorical features, alongside (optional) attention over both rows and columns, in a Transformer architecture. We note that, due to its use of featurewise feedforward layers, SAINT was impractical to use for our datasets with the largest numbers of features (ICU Hospital Mortality, ICU Length of Stay; both contain over 1000 features which resulted in over 13B parameters for even a single-layer SAINT model).

## F.3   Robustness Models

**Distributionally Robust Optimization (DRO):** We use two variants of DRO, both via [53]. For both methods, the model attempts to optimize a worst-care risk within a bounded distribution of the training data via a projected gradient descent procedure.

**Group DRO:** Originally introduced as a subgroup robustness method in [76], Group DRO is a DRO method which attempts to optimize the worst-group loss during training. Group DRO can also,

---

[29]https://github.com/Yura52/rtdl

however, be used as a domain robustness method by treating the domains as "group labels", which is how we use it in our study. We note that this use of Group DRO has been applied previously; e.g. [38].

### F.4 Domain Generalization Models

**Invariant Risk Minimization (IRM):** IRM [6] uses a modified training objective to learn models which a feature representation such that the optimal linear classifier on top of that representation matches across domains.

**MixUp:** Inter-Domain MixUp [90, 89] uses combinations of data points from random pairs of domains and their labels during training.

**Domain-Adversarial Neural Networks (DANN):** DANN [1] uses adversarial training to achieve domain robustness, where a discriminator attempts to predict the domain of a training example in order to match feature distributions across domains.

**Risk Extrapolation (REx):** REx [52] attempts to reduce differences in risk across training domains, in order to reduce a model's sensitivity to distributional shifts.

**CORAL:** CORAL (CORrelation ALignment) [82] attempts to ensure that feature activations are similar across domains; this can be used as either a domain generalization method or a domain adaptation method.

**MMD:** Similar to CORAL using a different kernel, MMD attempts to minimize the Maximum Mean Discrepancy (MMD) between domains.

### F.5 Label Shift Robustness Models

**Group DRO:** here, we use Group DRO [76] with class labels as the grouping attribute.

**Adversarial Label DRO:** This method, proposed in [92], uses a distributionally robust objective to optimize for the worst-case weighting over label groupings. We note that this approach is computationally expensive, requiring sample-level gradients even following the authors' original implementation, and so was not practical for our datasets with very large $n$ (ASSISTments, Public Health Insurance).

## G   Comparison To Other Benchmarking Toolkits

In this section, we provide a brief comparison of TableShift to other relevant benchmarking toolkits. We note that our goal in this section is *not* to fully characterize the functionality of other benchmarking platforms; it is only to compare and contrast their relevant attributes with TableShift and to motivate the creation of a novel benchmark and API for TableShift (as opposed to incorporating TableShift into an existing toolkit).

As noted above, there is *no* existing benchmark for domain shift in tabular data. However, in this section we compare to three main categories of relevant related toolkits: (1) domain shift benchmarks for non-tabular data (DomainBed, WILDS); (2) IID (non-domain-shift) benchmarks for tabular data ([37], OpenML); and (3) generic data-hosting platforms (Huggingface Datasets, TensorFlow Datasets. We briefly introduce and compare to each of these below.

### G.1   Domain Shift Benchmarks for Non-Tabular Data

**WILDS:** WILDS[30] is perhaps the closest benchmark to TableShift, but only uses non-tabular data. WILDS demonstrates a lightweight, useful set of programming abstractions for benchmarking models and sharing results across a diverse set of datasets for domain shift. WILDS interfaces with image and text datasets, and includes a rich variety of datasets with real-world sensitive attributes, carefully selected domain shifts, and has wide adoption in the robustness community. WILDS includes a high-quality Python API, which has led to wide integration with researchers' open-source code and widespread adoption. However, WILDS is currently not compatible with tabular datasets and does not

---

[30]See https://wilds.stanford.edu/ and [50].

include any tabular datasets in its benchmark suite. The needs for tabular datasets are different than the datasets currently used in WILDS (i.e. non-Torch models must be supported by the benchmark; subgroup and domain-shift information are handled differently for our use cases; data preprocessing is also different for tabular data as noted above).

**DomainBed:** DomainBed[31] is a benchmark that contains several reference implementations, including some that have been adapted for use in TableShift. In addition to these model implementations, DomainBed serves as an interface to several existing datasets through a Python API. However, DomainBed is specifically adapted to image data. It only supports image datasets with a specific folder structure (which would make extending to tabular datasets nontrivial) and includes many image-specific augmentation components of its pipelines. It also uses ResNet50/ResNet18 networks designed specifically for image classification, and therefore does not currently support either deep learning models suited to image data, nor (more importantly) the effective non-DL baselines described above such as XGBoost and LightGBM.

**Shift Happens:** The "shift happens" benchmark[32] is a community-built benchmark suite for image models. It specifically aims to feature datasets with domain shift, for tasks including image classification under domain shift, and out-of-distribution detection. The benchmark includes a Python API. This benchmark does not support tabular datasets, and is much less widely used, perhaps due to the community-driven effort (as opposed to benchmarks such as WILDS and DomainBed, which come packaged with preselected datasets and domain splits).

**Shifts 2.0:** Shifts ([57], recently upgraded to Shifts 2.0 [57]) is a collection of multimodal tasks with domain shifts. The Shifts benchmark is a part of the Shifts Project, an international collaboration of academic and industrial researchers dedicated to studying distributional shift.[33] Shifts 2.0, the current version, includes five tasks: tabular weather prediction, tabular marine cargo vessel power consumption prediction, machine translation, self-driving car vehicle motion prediction, and segmentation of white matter Multiple Sclerosis lesions in 3D magnetic resonance brain images. While shifts does contain two tabular data tasks, its relatively small number of tasks makes it a less reliable benchmark compared to the rich set of tabular datasets comprising TableShift. Shifts also does not include any tasks with real-world sensitive subgroups (such as age, race, or gender) which are of particular interest in many tabular classification tasks. Additionally, the domains represented in the tabular tasks of Shifts do not cover many critical domains widely recognized as using tabular data (e.g. finance, medicine, etc.; see Section D.2).

## G.2 IID Benchmarks for Tabular Data

**Benchmark of [37]:** The unnamed benchmark proposed in this work is intended to provide a consistent benchmarking suite for tabular dataset classification tasks, and was motivated by some of the same gaps described in this work. However, the datasets comprising the benchmark of [37] do not meet our specifications, for several reasons. First, the datasets are limited to be of *maximum* size $10k$ observations; this is too small for reliable and repeated benchmarking comparisons. Additionally, the datasets are label-balanced; in contrast, we use the naturaly-occurring label distributions for all datasets (and we show that these label distributions are importantly related to shift gap). Most critically, the benchmark datasets in [37] do not contain domain shifts; for most or all of the datasets, it is does not appear that a domain shift could be induced from splitting the existing data on an existing feature.

**OpenML:** OpenML has some overlap with the proposed functionality. However, OpenML both lacks functionality we seek to provide (subgroup robustness and domain shift utilities; a curated set of benchmarking datasets; lightweight and standardized control over common tabular preprocessing methods) and also provides extraneous functionality not needed for a lightweight tabular benchmarking library (tools for OpenML-hosted model/pipeline/evaluation sharing and collaboration; API/utilities for model training) . Additionally, OpenML is not yet widely used in the tabular data community, as demonstrated by the wide calls for effective tabular data benchmarking tools ([15], [37], [78]) and the lack of usage of OpenML in most robustness works, even recent works (e.g. [91]), which largely focus on canonical tabular datasets such as COMPAS and Adult.

---

[31]See https://github.com/facebookresearch/DomainBed and [38].
[32]https://shift-happens-benchmark.github.io/index.html
[33]https://shifts.ai/

Table 18: Comparison of relevant benchmarks. DB: DomainBed. OML: OpenML. GOV22: [37]. SH: Shift Happens. HFDS: Hugging Face Datasets. TFDS: TensorFlow Datasets.

| | | DB | OML | GOV22 | WILDS | SH | Shifts 2.0 | HFDS | TFDS | TableShift |
|---|---|---|---|---|---|---|---|---|---|---|
| **Output** | Supports tabular data input formats (e.g. .csv) | | ✓ | | | | ✓ | ✓ | ✓ | ✓ |
| | Supports tabular output formats (e.g. `pd.DataFrame`) | | | | | | ✓ | ✓ | ✓ | ✓ |
| | Support for large/out-of-core datasets | ✓ | | | ✓ | ✓ | | ✓ | ✓ | ✓ |
| **Preprocessing** | Supports tabular preprocessing: categorical encoding; missing value handling | | | | | | | | | ✓ |
| | Provides shared utilities for user-defined preprocessing per dataset | ✓ | | | ✓ | ✓ | | ✓ | ✓ | ✓ |
| **Metadata** | Feature-level metadata | | | | | | | | | ✓ |
| | Dataset-level metadata | | | | | | | ✓ | ✓ | ✓ |
| **Benchmark Tasks** | Includes domain shift (non-IID) splits | ✓ | | | ✓ | ✓ | ✓ | some | | ✓ |
| | Meets criteria in §3.1 | ✓ | | | ✓ | | | | | ✓ |
| | Large test sets ($\geq 10k$) | ✓ | | | | ✓ | | | | |
| | Includes label-imbalanced datasets | ✓ | ✓ | | ✓ | ✓ | ✓ | ✓ | ✓ | ✓ |
| | Includes real-world sensitive attributes | | some | | ✓ | | | some | some | all |

**DataPerf:** DataPerf is "a benchmark package for evaluating ML datasets and dataset working algorithms" [59]. Similar to WILDS and DomainBed, DataPerf covers many domains, not only tabular data. DataPerf has a much broader set of goals relative to TableShift, and includes a collection of tasks, metrics and rules that are intended to benchmark all stages of an ML pipeline, from raw data to test set selection amd model selection. However, DataPerf does *not* offer domain shifts, and while it is possible domain shifts could be integrated into DataPerf, it does not natively support the kind of benchmarking that we intend to support with TableShift.

### G.3    Other Data Hosting Platforms

**Hugging Face Datasets (HFDS):** HFDS is a generic dataset hosting utility provided by the company Hugging Face. It serves as a large, open dataset repository; however, these datasets are not curated for size, featurization, or quality. HFDS is a public platform where datasets can be contributed openly. However, of the tabular datasets on HFDS, few if any meet the specifications described in §3.1; in particular, most are not domain shift datasets.

**TensorFlow Datasets (TFDS):** TFDS is similar in many ways to HFDS. It is a public, open repository of datasets, and new datasets can be contributed via git. However, TFDS also has the same shortcomings as HFDS; in particular, its open format leads to a collection of datasets mostly not useful for tabular data benchmarking and almost no datasets with meaningful distribution shifts.

## H    Training Details

Neural network-based models were trained on GPU, either NVIDIA RTX 2080 Ti GPUs with 11GB of RAM, or NVIDIA Tesla M60 GPUs with 48 GB of RAM. We used a batch size of 4096 for training all models, except where this was not possible due to memory limitations (see code for details).

Where possible, gradient boosted tree models were trained using CPU (not GPU).

# I   Hyperparameter Grids

The hyperparameter tuning grid for each model is shown in Table 19. We make the full hyperparameter tuning code available as part of the release of this work, at https://github.com/mlfoundations/tableshift.

We made an effort to ensure our hyperparameter grids always included at least the full grid described in the original work(s) cited for each learning method used in our study. For some methods, our grid is a superset of the hyperparameter grid in the original study. This is to ensure, where possible, that we tune a similar range of certain parameters (i.e. learning rate) across all methods. For domain generalization methods, since we are not aware of any prior application to these methods to tabular data, we use the hyperparameter grids from [38].

Table 19: Hyperparameter grids used in all experiments. ♣: all MLP parameters also tuned. Continued in Table .

| Model | Hyperparameter | Values |
|---|---|---|
| **Baseline Methods** | | |
| ♣ MLP | Learning Rate | $\text{LogUniform}(1e^{-5}, 1e^{-1})$ |
| | Weight Decay | $\text{LogUniform}(1e^{-6}, 1)$ |
| | Num. Layers | $\{1, 2, \ldots, 8\}$ |
| | Hidden Units | $\{64, 128, 256, 512, 1024\}$ |
| | Num Epochs | $\text{QRandInt}(5, 100, 5)$ |
| | Dropout | $\text{Unif}(0, 0.5)$ |
| | Batch Size | $\{4096\}$ |
| XGBoost | Learning Rate | $\text{LogUniform}\{1e-5, 1\}$ |
| | Max. Depth | $\{3, \ldots, 10\}$ |
| | Min Child Weight | $\text{LogUniform}\{1e-8, 1e5\}$ |
| | Row Subsample | $\text{Uniform}\{0.5, 1.\}$ |
| | Column Subsample (Tree) | $\text{Uniform}\{0.5, 1.\}$ |
| | Column Subsample (Level) | $\text{Uniform}\{0.5, 1.\}$ |
| | $\gamma$ | $\text{LogUniform}\{1e-8, 1e2\}$ |
| | $\lambda$ | $\text{LogUniform}\{1e-8, 1e2\}$ |
| | $\alpha$ | $\text{LogUniform}\{1e-8, 1e2\}$ |
| | Max. Bins | $\{128, 256, 512\}$ |
| LightGBM | Learning Rate | $\text{LogUniform}\{1e-5, 1\}$ |
| | Min. Child Samples | $\{1, 2, 4, 8, 16, 32, 64\}$ |
| | Min. Child Weight | $\text{LogUniform}(1e-8, 1e5)$ |
| | Row Subsample | $\text{Uniform}\{0.5, 1.\}$ |
| | Max. Depth | $\{\text{None}, 1, 2, \ldots, 31\}$ |
| | Column Subsample (Tree) | $\text{Uniform}\{0.5, 1.\}$ |
| | Column Subsample (Level) | $\text{Uniform}\{0.5, 1.\}$ |
| | $\lambda$ | $\text{LogUniform}\{1e-8, 1e2\}$ |
| | $\alpha$ | $\text{LogUniform}\{1e-8, 1e2\}$ |
| CatBoost | Learning Rate | $\text{LogUniform}\{1e-3, 1\}$ |
| | Depth | $\text{QRandInt}\{3, 10\}$ |
| | Bagging Temp. | $\text{LogUniform}(1e-6, 1)$ |
| | $L_2$ Leaf Reg. | $\text{LogUniform}(1, 100)$ |
| | Leaf Estimation Iterations | $\text{QRandInt}\{1, 10\}$ |
| **Domain Generalization Methods** | | |
| DANN ♣ | $\text{LR}_G$ | $\text{LogUniform}\{1e-5, 1e-1\}$ |
| | $\text{WeightDecay}_G$ | $\text{LogUniform}\{1e-6, 1\}$ |
| | $\text{LR}_d$ | $\text{LogUniform}\{1e-5, 1e-1\}$ |
| | $\text{WeightDecay}_D$ | $\text{LogUniform}\{1e-6, 1\}$ |
| | D steps per G step | $\text{LogUniform}\{2, 2^3\}$ |
| | Grad Penalty | $\text{LogUniform}\{1e-2, 1e1\}$ |
| | Loss $\lambda$ | $\text{LogUniform}\{1e-2, 1e2\}$ |
| IRM ♣ | IRM $\lambda$ | $\text{LogUniform}\{1e-1, 1e5\}$ |
| | IRM Penalty Anneal Iters | $\text{LogUniform}\{1, 1e4\}$ |
| MixUp ♣ | MixUp $\alpha$ | $\text{Uniform}\{1e-1, 1e1\}$ |
| VReX ♣ | VReX $\lambda$ | $\text{LogUniform}\{1e-1, 1e5\}$ |
| | VReX Penalty Anneal Iters | $\text{LogUniform}\{1, 1e4\}$ |
| CORAL ♣ | MMD $\gamma$ | $\text{LogUniform}\{1e-1, 1e1\}$ |
| MMD ♣ | MMD $\gamma$ | $\text{Uniform}\{1e-1, 1e1\}$ |

| Model | Hyperparameter | Values |
|---|---|---|
| **Tabular Neural Networks** | | |
| ResNet ♣ | Num. Blocks | $\{1, 2, 3, 4\}$ |
| | $d$ main | RandInt$(64, 1024)$ |
| | Hidden Factor | RandInt$(1, 4)$ |
| | Dropout First | Uniform$(0, 0.5)$ |
| | Dropout Second | Uniform$(0, 0.5)$ |
| FT-Transformer ♣ | Num. Blocks | $\{1, 2, 3, 4\}$ |
| | Residual Dropout | Uniform$(0, 0.2)$ |
| | Attention Dropout | Uniform$(0, 0.5)$ |
| | FFN Dropout | Uniform$(0, 0.5)$ |
| | FFN Factor | Uniform$(2/3, 8/3)$ |
| | FFN Factor | $\{64, 128, 256, 512\}$ |
| TabTransformer | Num. Blocks | $\{1, 2, 3, 4\}$ |
| | Learning Rate | LogUniform$(1e^{-5}, 1e^{-1})$ |
| | Weight Decay | LogUniform$(1e^{-6}, 1)$ |
| | Num Epochs | QRandInt$(5, 100, 5)$ |
| | FFN Dropout | Uniform$(0, 0.5)$ |
| | Attention Dropout | Uniform$(0, 0.5)$ |
| | Model Dimension | $32, 64, 128, 256$ |
| | Depth | $3, 4, 5, 6$ |
| | Num. Heads | $2, 4, 8$ |
| NODE | Num. Epochs | $\{1, 2, 3, 4, 5\}$ |
| | Num. Layers | $2, 4, 8$ |
| | Total Tree Count | $1024, 2048$ |
| | Tree Depth | $6, 8$ |
| | Tree Output Dim. | $2, 3$ |
| | FFN Factor | $\{64, 128, 256, 512\}$ |
| | Learning Rate | LogUniform$(1e^{-5}, 1e^{-1})$ |
| | Weight Decay | LogUniform$(1e^{-6}, 1)$ |
| SAINT | Num. Epochs | $\{1, 2, 3, 4, 5\}$ |
| | Depth | $4, 6$ |
| | Model Dimension | $4, 8, 12, 16, 32$ |
| | Learning Rate | LogUniform$(1e^{-5}, 1e^{-1})$ |
| | Weight Decay | LogUniform$(1e^{-6}, 1)$ |
| | FFN Dropout | Uniform$(0.1, 0.8)$ |
| | Heads | $4, 8$ |
| | Attention Type | Row, Col, RowCol |
| **Domain Robustness Methods** | | |
| DRO ♣ | Uncertainty set size | LogUniform$\{1e-4, 1.\}$ |
| | Geometry | $\{$ CVaR, $\chi^2\}$ |
| Group DRO ♣ | Group weights step size | LogUniform$(1e^{-4}, 1)$ |
| **Label Shift Robustness Methods** | | |
| Label Group DRO ♣ | Group weights step size | LogUniform$(1e^{-4}, 1)$ |
| Adversarial Label DRO ♣ | Adv. Learning Rate $\eta_\pi$ | LogUniform$\{1e-4, 1e-1\}$ |
| | Adv. radius $r$ | LogUniform$\{1e-5, 0.5\}$ |
| | Clip max $r$ | LogUniform$\{1e-1, 10\}$ |
| | $\epsilon$ | LogUniform$\{1e-4, 1e-1\}$ |

Table 20: Hyperparameter grids used in all experiments. ♣: all MLP parameters also tuned. Continued from Table 19.

