| | Row Subsample | Uniform$\{0.5, 1.\}$ |
| | Column Subsample (Tree) | Uniform$\{0.5, 1.\}$ |
| | Column Subsample (Level) | Uniform$\{0.5, 1.\}$ |
| | $\gamma$ | LogUniform$\{1e-8, 1e2\}$ |
| | $\lambda$ | LogUniform$\{1e-8, 1e2\}$ |
| | $\alpha$ | LogUniform$\{1e-8, 1e2\}$ |
| | Max. Bins | $\{128, 256, 512\}$ |
| LightGBM | Learning Rate | LogUniform$\{1e-5, 1\}$ |
| | Min. Child Samples | $\{1, 2, 4, 8, 16, 32, 64\}$ |
| | Min. Child Weight | LogUniform $(1e-8, 1e5)$ |
| | Row Subsample | Uniform$\{0.5, 1.\}$ |
| | Max. Depth | $\{None, 1, 2, \ldots, 31\}$ |
| | Column Subsample (Tree) | Uniform$\{0.5, 1.\}$ |
| | Column Subsample (Level) | Uniform$\{0.5, 1.\}$ |
| | $\lambda$ | LogUniform$\{1e-8, 1e2\}$ |
| | $\alpha$ | LogUniform$\{1e-8, 1e2\}$ |
| CatBoost | Learning Rate | LogUniform$\{1e-3, 1\}$ |
| | Depth | QRandInt$\{3, 10\}$ |
| | Bagging Temp. | LogUniform $(1e-6, 1)$ |
| | $L_2$ Leaf Reg. | LogUniform $(1, 100)$ |
| | Leaf Estimation Iterations | QRandInt$\{1, 10\}$ |
| **Domain Generalization Methods** | | |
| DANN ♣ | $LR_G$ | LogUniform$\{1e-5, 1e-1\}$ |
| | WeightDecay$_G$ | LogUniform$\{1e-6, 1\}$ |
| | $LR_d$ | LogUniform$\{1e-5, 1e-1\}$ |
| | WeightDecay$_D$ | LogUniform$\{1e-6, 1\}$ |
| | D steps per G step | LogUniform$\{2, 2^3\}$ |
| | Grad Penalty | LogUniform$\{1e-2, 1e1\}$ |
| | Loss $\lambda$ | LogUniform$\{1e-2, 1e2\}$ |
| IRM ♣ | IRM $\lambda$ | LogUniform$\{1e-1, 1e5\}$ |
| | IRM Penalty Anneal Iters | LogUniform$\{1, 1e4\}$ |
| MixUp ♣ | MixUp $\alpha$ | Uniform$\{1e-1, 1e1\}$ |
| VReX ♣ | VReX $\lambda$ | LogUniform$\{1e-1, 1e5\}$ |
| | VReX Penalty Anneal Iters | LogUniform$\{1, 1e4\}$ |
| CORAL ♣ | MMD $\gamma$ | LogUniform$\{1e-1, 1e1\}$ |
| MMD ♣ | MMD $\gamma$ | Uniform$\{1e-1, 1e1\}$ |

| Model | Hyperparameter | Values |
|---|---|---|
| **Tabular Neural Networks** | | |
| ResNet ♣ | Num. Blocks | $\{1, 2, 3, 4\}$ |
| | $d$ main | RandInt$(64, 1024)$ |
| | Hidden Factor | RandInt$(1, 4)$ |
| | Dropout First | Uniform$(0, 0.5)$ |
| | Dropout Second | Uniform$(0, 0.5)$ |
| FT-Transformer ♣ | Num. Blocks | $\{1, 2, 3, 4\}$ |
| | Residual Dropout | Uniform$(0, 0.2)$ |
| | Attention Dropout | Uniform$(0, 0.5)$ |
| | FFN Dropout | Uniform$(0, 0.5)$ |
| | FFN Factor | Uniform$(2/3, 8/3)$ |
| | FFN Factor | $\{64, 128, 256, 512\}$ |
| TabTransformer | Num. Blocks | $\{1, 2, 3, 4\}$ |
| | Learning Rate | LogUniform$(1e^{-5}, 1e^{-1})$ |
| | Weight Decay | LogUniform$(1e^{-6}, 1)$ |
| | Num Epochs | QRandInt$(5, 100, 5)$ |
| | FFN Dropout | Uniform$(0, 0.5)$ |
| | Attention Dropout | Uniform$(0, 0.5)$ |
| | Model Dimension | $32, 64, 128, 256$ |
| | Depth | $3, 4, 5, 6$ |
| | Num. Heads | $2, 4, 8$ |
| NODE | Num. Epochs | $\{1, 2, 3, 4, 5\}$ |
| | Num. Layers | $2, 4, 8$ |
| | Total Tree Count | $1024, 2048$ |
| | Tree Depth | $6, 8$ |
| | Tree Output Dim. | $2, 3$ |
| | FFN Factor | $\{64, 128, 256, 512\}$ |
| | Learning Rate | LogUniform$(1e^{-5}, 1e^{-1})$ |
| | Weight Decay | LogUniform$(1e^{-6}, 1)$ |
| SAINT | Num. Epochs | $\{1, 2, 3, 4, 5\}$ |
| | Depth | $4, 6$ |
| | Model Dimension | $4, 8, 12, 16, 32$ |
| | Learning Rate | LogUniform$(1e^{-5}, 1e^{-1})$ |
| | Weight Decay | LogUniform$(1e^{-6}, 1)$ |
| | FFN Dropout | Uniform$(0.1, 0.8)$ |
| | Heads | $4, 8$ |
| | Attention Type | Row, Col, RowCol |
| **Domain Robustness Methods** | | |
| DRO ♣ | Uncertainty set size | LogUniform$\{1e-4, 1.\}$ |
| | Geometry | $\{$ CVaR, $\chi^2\}$ |
| Group DRO ♣ | Group weights step size | LogUniform$(1e^{-4}, 1)$ |
| **Label Shift Robustness Methods** | | |
| Label Group DRO ♣ | Group weights step size | LogUniform$(1e^{-4}, 1)$ |
| Adversarial Label DRO ♣ | Adv. Learning Rate $\eta_\pi$ | LogUniform$\{1e-4, 1e-1\}$ |
| | Adv. radius $r$ | LogUniform$\{1e-5, 0.5\}$ |
| | Clip max $r$ | LogUniform$\{1e-1, 10\}$ |
| | $\epsilon$ | LogUniform$\{1e-4, 1e-1\}$ |

Table 20: Hyperparameter grids used in all experiments. ♣: all MLP parameters also tuned. Continued from Table 19.