# OpenReview forum: "Benchmarking Distribution Shift in Tabular Data with TableShift"
_NeurIPS.cc/2023/Track/Datasets_and_Benchmarks — NeurIPS 2023 Datasets and Benchmarks Poster_

### Official Review · Reviewer_yyVo · 2023-06-27
**Everything everywhere all at once... yet lacks specificity**

**Rating:** 3
**Confidence:** 5
**Clarity:** The paper is well written.

**Strengths:**

The submission demonstrates a compelling motivation for benchmarking distribution shifts in tabular data, which is highly relevant to the research community.
Moreover, the inclusion of easy-to-use datasets and corresponding APIs is a notable strength of the submission. The provision of user-friendly datasets and APIs enhances the usability of the proposed benchmark and encourages wider adoption within the community.

**Additional Feedback:**

This paper addresses a timely and relevant topic concerning distribution shift robustness in tabular data. However, upon careful evaluation, it is evident that the paper lacks specificity and coherence in its approach. The utilization of datasets from both domain generalization (DG) benchmark setups and non-specific sources, along with comparing different model architectures and DG methods, contributes to a lack of clarity and complicates meaningful comparisons.

One key aspect for improvement is to focus on a specific problem formulation rather than attempting to cover a broad range of topics. By narrowing the scope, the authors can provide a more targeted analysis and enable a deeper understanding of the chosen problem domain. This would involve selecting datasets, model architectures, and evaluation methods that align with the defined problem formulation, facilitating a more coherent and interpretable study.

Furthermore, the excessive use of various figures to present different results further hampers the clarity and comparability of the findings. It is important to streamline the presentation of results and ensure that the chosen visualizations effectively convey the key insights without causing unnecessary confusion. Simplifying and consolidating the figures would greatly aid in the interpretation and synthesis of the experimental outcomes.

To enhance the overall quality and impact of the paper, it is crucial for the authors to refine their focus and provide a more focused analysis within a well-defined problem formulation. By addressing these concerns and improving the clarity and coherence of the research, the paper can be better positioned to make a meaningful contribution to the field.

Based on the aforementioned limitations, I regret to say that I am unable to recommend acceptance of the paper in its current form. However, I believe that with the appropriate revisions and enhancements, the authors have the potential to produce a more impactful and valuable contribution.

**Correctness:**

In the current submission, it is difficult to thoroughly evaluate the correctness of the paper due to the lack of refinement. Specifically, paragraphs (1) - (5) of the "Opportunities for improvement" section require further attention to provide a clearer understanding of the claims made, the soundness of the dataset construction (if applicable), and the appropriateness and correctness of the experiment design for benchmarks.

**Documentation:**

Please check (7) paragraph from from "Opportunities for improvement" section.

**Ethics:**

No ethical concerns were found.

**Limitations:**

In the current work, the authors have not explicitly addressed the limitations of their research. Given the inclusion of numerous tasks involving real-world sensitive subgroups, such as age, race, or gender, it would be beneficial for the authors to provide a thorough examination of the potential risks associated with the incorrect interpretation of the data and benchmarks.

**Opportunities For Improvement:**

**(1) In-depth Data Analysis:** The paper introduces new data sources for distribution shift benchmark but lacks extensive exploratory data analysis (EDA). Comprehensive EDA, detailing the presence of the shift, should be included in the main body of the paper or the appendix. Visual tools such as violin plots to showcase variations in target distributions across domains would enhance understanding. Regarding the authors' emphasis on label shifts between various domains in Table 4, it would be beneficial for a discerning reader to explore their significance in relation to the per-domain label variance.

**(2) Clarification of Dataset Selection:** The authors posit that "TABLESHIFT comprises 15 curated tasks designed for reliable benchmarking of tabular data under distribution shift." However, not all datasets reviewed bear a "domain generalization" label in Table 1, creating ambiguity about the shift type represented in them. If the datasets include subpopulation\[1\]/time-domain\[2\] shift, it's suggested that all Domain Generalisation methods be benchmarked on them. If they do not, clarity on the type of shift presented or the reasoning behind their usage within the distribution shift benchmark is needed.
- \[1\] - WILDS: A Benchmark of in-the-Wild Distribution Shifts
- \[2\] - WOODS: Benchmarks for Out-of-Distribution Generalization in Time Series

**(3) Improved Report Presentation:** The paper consistently presents results with accompanying figures. Although this graphical representation supports the authors' assertions, it hinders precise comparisons between different models/methods or potential future research. Further, standard deviation (std) values and their derivation methods should be included in Tables 1, and 5 for better comprehensibility. Lastly, a detailed explanation of the evaluation metric used – PMA-OOD and its justification is warranted.

**(4) Models vs. Methods Clarification:** The authors compare different models (MLP, ResNet, FT-Transformer) and methods (CORAL, MMD, DRO) without justifying why certain combinations weren't examined. Clarification on why different models weren't tested with different methods (FT-Transformer + CORAL) and the rationale behind evaluations (ERM?) of models like MLP/ResNet/FT-Transformer would enhance the paper's comprehensibility.

**(5) Hyperparameters Reporting:** The appendix provides the hyperparameters used but lacks reasoning behind the chosen ranges. Clarification on how MLP-based hyperparameters were tuned for FT-Transformer and why FT-Transformer and catboost hyperparameter spaces do not align with the original paper \[3\] would contribute to understanding.
- \[3\] Revisiting Deep Learning Models for Tabular Data

**(6) Expanded Related Work:** Despite asserting the novelty of their benchmark for distribution shifts in tabular data ("While there are existing benchmarks for IID tabular classification, none of these focus on distribution shifts." - line 28 and "However, previous investigations have not explored how these models (i.e. ResNet- and Transformer-based architectures) perform under distribution shift in tabular data." - line 177), the authors acknowledge existing benchmarks - Shifts \[4\], \[5\], which "does contain two tabular data tasks", line 1305 - in the appendix. Greater visibility of these related works in the main body of the paper could increase its credibility. Alternatively, consideration could be given to integrating these datasets into the TableShift benchmark.
- \[4\] Shifts: A dataset of real distributional shift across multiple large-scale tasks
- \[5\] Shifts 2.0: Extending the dataset of real distributional shifts

**(7) Codebase Refinement:** In testing the provided code, I encountered issues in downloading datasets due to a pandas error. To ensure reproducibility and public usability, a thorough review and correction of the codebase is necessary.

**(8) Appendix Refinement:** Section D.3 and Table 6.

**Relation To Prior Work:**

Please check (6) paragraph from from "Opportunities for improvement" section.

**Summary And Contributions:**

This paper introduces the TABLESHIFT benchmark, addressing the lack of robustness evaluation in tabular machine learning models under distribution shift. TABLESHIFT consists of 15 tasks with associated shifts, covering domains such as finance, education, public policy, healthcare, etc. The benchmark provides an accessible Python API for dataset access and includes implementations of many models and domain generalization methods. Through a large-scale study, the authors find a linear relationship between in-distribution and out-of-distribution accuracy, demonstrate the impact of domain robustness methods on shift gaps, and establish a strong connection between shift gaps and shifts in the label distribution.

---

> ### Author Response · Authors · 2023-08-22
> **Response to Reviewer yyVo27**
>
> Thank you to the reviewer for their thorough feedback. While we recognize that the reviewer has some concerns about the paper, we were also glad to see that they found it “well written” and addressed “ a timely and relevant topic concerning distribution shift robustness in tabular data”. We see a few clear themes in the review; we attempt to discuss them below, with references to the numbered points in the review. We believe that we can address this helpful feedback and improve the paper as a result.
>
> ### (1) In-depth Data Analysis:
> * We recognize the reviewers’ desire for a more detailed exploratory data analysis. Since our datasets are strictly discrete classification tasks, we are not able to generate meaningful violin plots of the label distributions. Instead, we have given the exact values of the label distribution for each ID/OOD dataset split, along with their variances, in the new supplementary Table 5. We agree with the reviewer that, given our emphasis on label shift, this provides useful information for readers.
> * We have also added standard error of the \Delta_acc metric to Table 1. To compute these values, we follow a standard procedure: compute the variance of the individual accuracies (again, using Clopper-Pearson intervals), and then sum these variances to obtain the variance of the difference.
>
> ### (2) Clarification of Dataset Selection:
>
> * The datasets in the TableShift benchmark were subject to a careful selection process. Some of our formal criteria are described in Section 3.1.
> * As we emphasize in the paper, we also made a point to construct a diverse and comprehensive benchmark, as other reviewers (e.g. XmM225, dJDE21) noted.  However, we believe that this is an asset of the benchmark, as all of these forms of shift are covered, and the shifts are clearly described in the paper and also explicitly defined in the accompanying code (if, for example, a user is only interested in subpopulation shift, they may use only those datasets for their study).
> * Having a diverse set of multiple types of shift is in line with other existing widely-used domain shift benchmarks, including those the reviewer cites (WILDS, SHIFTS). The WILDS paper emphasizes “a diverse range of distribution shifts that naturally arise in real-world applications”, which also describes our benchmark); SHIFTS 2.0 similarly emphasizes their benchmark’s diversity.
> * Our empirical results are also consistent across the shift types the reviewer mentions (e.g. there are no outliers or groups in Figure 1 depending on the nature of the shift). We believe these empirical results also demonstrate the usefulness of a diverse, comprehensive domain shift benchmark for tabular data -- all distribution shifts in our benchmark display similar empirical behavior.
>
> ### (3) Improved Report Presentation:
> * The reviewer correctly points out that it can be difficult to glean the exact data values from a crowded scatter plot. For every figure, have added a link in the caption to the table(s) in the appendix with the exact data values. This includes new supplementary Tables 6, 7, and 8.
> * With apologies for any confusion, we note that Table 5 in the supplement already contained the exact data values for every point in Figures 1, 2, and 3. We did not explain this in the paper, and are grateful to the reviewer for pointing this out. For Figure 4, the exact x-axis values are given in Table 4 (column for ∆y) and the x-values are computed via Table 5 (by taking OOD-ID as in Equation 1).
> * We are grateful for the reviewer’s point that we could do better to explain and motivate PMA-OOD. We will add a discussion to the paper with the extra camera-ready page. Briefly, PMA measures what percentage of the maximum observed accuracy (across all models on a task) is achieved by a given model on that task. We adopt the PMA metric (often used in other tabular data work, e.g. https://arxiv.org/abs/2212.13881) because it is dataset-invariant (allows for comparison across multiple datasets) and takes into account the magnitude of differences in performance (not simply rankings).
> * We use Clopper-Pearson confidence intervals at α= 0.05 for all of our accuracy measurements (shown in e.g. Figure 2 and accompanying caption).
> We have added standard error values to Table 1 as suggested.
> Due to space (and readability) constraints, we are not able to include CI values in Table 5, but instead added a reference in the Table 5 caption to where these intervals are shown in graphical form (Figures 2, 3). In this case, we believe that seeing the overlap of all CIs for a given dataset in the plot better supports useful inference about the results (instead of listing over 500 standard errors in Table 5), since all points within each dataset are similarly affected by the (ID, OOD) test set sizes. We note that even when accounting for the uncertainty from these intervals, our data still follow the trends described in our analysis.

---

> > ### Author Response · Authors · 2023-08-22
> > **Response Continued**
> >
> > ### (4) Models vs. Methods Clarification:
> >
> > * The reviewer correctly identifies an important gap in the existing literature: that there are many combinations of “models” and “methods” which have not been combined. We agree -- which is why we emphasize the need for future work on “hybrid methods which combine robustness-enhancing optimization objectives (such as Group DRO) with improved neural network architectures” in our conclusion. Since ours is a benchmarking study and is not intended to produce new algorithms, we focused our finite computational resources on the combinations of methods which we observed in prior works.
> > * We are running a small selection of experiments of hybrid methods, as the reviewer suggested, on a subset of tasks. We will report the results here in a follow-up soon.
> >
> > ### (5) Hyperparameters Reporting:
> >
> > * Related to (4), our hyperparameter grids are also drawn from prior work. However, we increased the size of hyperparameter grids when there were multiple relevant works tuning that method or for consistency with the other models in the benchmark. For example, we sweep over (1…8) layers for FT-Transformer, while the original paper [3] uses (1..4) or (1..6) depending on dataset, since we also used (1..8) layers for other nn-based models. We sweep over a larger range of learning rates and weight decay values for FT-Transformer, compared to [3], for the same reason. We have clarified this in the paper text (Supplementary Section H). For CatBoost, we use an identical grid to [3].
> >
> > ### (6) Expanded Related Work:
> >
> > * Relation to prior work (6): the reviewer notes that Shifts and Shifts 2.0 also contain two tabular data tasks; we will move our discussion of Shifts into the main text with the extra camera-ready page. We agree that it is important to recognize that this is perhaps the only example of tabular distribution shift in any formal benchmark.
> >
> > ### (7) Codebase Refinement:
> >
> > * We are grateful for the reviewers’ feedback on improving the codebase. We encourage the reviewer to see our detailed response to Reviewer XmM225, where we describe many improvements made to the code, software documentation, environment setup, and  technical reproducibility. We encourage the reviewer to try the setup commands described there and hope that they find their configuration issues have been addressed.
> >
> > ### (8) Appendix Refinement:
> >
> > * We have made the suggested refinements to Section D.3 and Table 6 (which is now Table 9).

---

> > > ### Comment · Reviewer_yyVo · 2023-08-25
> > >
> > > I sincerely thank the authors for their response and efforts to resolve some of my questions. However, I still think several major concerns remain:
> > >
> > > **(1) In-depth Data Analysis:** While you have provided label distribution exact values in the supplementary Table 6, might I suggest incorporating native visualizations, such as KDE plots, to offer clearer insights? Scrutinizing Table 6, it's somewhat challenging to gauge the significance of distribution shifts and their categorization as depicted in Table 1. Moreover, clarifying the task classification (Binary, Multiclass with 'N' classes) in Table 1 might offer readers enhanced comprehension, especially in the context of understanding "the proportion of positive labels" from Table 6.
> > >
> > > Additionally, could you elucidate the naming discrepancies between Table 1 and Table 6?
> > >
> > > Lastly, an extensive exploration of the underlying reasoning for achieving generalization across these shifts would provide valuable insights. Specifically, it appears logical to extend generalization to various healthcare devices employed across diverse hospitals, as they are inherently constructed, operated, and calibrated in a consistent manner. However, the applicability of such an approach to school assignments warrants deeper scrutiny. Identifying the shared attributes that would enable effective generalization in this context merits careful consideration.
> > >
> > > Regarding the addition of the standard error of the \Delta_acc metric in Table 1: the table caption could benefit from an accompanying description.
> > >
> > > **(2) Clarification of Dataset Selection:** Could you further detail the OOD shifts represented in the datasets of Table 1? While I appreciate the dataset diversity, I remain curious about the types of OOD shifts inherent in them, a point not yet addressed in your response.
> > >
> > > **(3) Improved Report Presentation:** There seems to be a discrepancy between Figure 1's ID-OOD accuracy representation and Table 5's **Max** ID-OOD accuracy. It would be beneficial to unify the chosen metric for benchmarking purposes and elucidate the rationale behind its selection.
> > >
> > > Moreover, when attempting to correlate the datasets between Figure 1 and Table 5, some inconsistencies arise. For instance, while Table 5 lists multiple datasets with a max OOD accuracy approximating 0.4 for DRO, Figure 1 depicts only one such observation.
> > >
> > > With reference to Figure 2, the choice of scale obscures the insights for some datasets like Assistments and Diabetes. Furthermore, a clarification on the location of results for datasets like FICO HELOC within Figure 2, relative to their presence in Table 5, would be helpful.
> > >
> > > Lastly, could you specify where the required standard deviation/variance measurements for the results can be found? Table 5, as I interpret it, encapsulates only mean values. The graphical representation, in its current state, neither facilitates reproducibility nor offers clarity due to its minuscule size. I believe refining Table 5 to encompass these vital results would be in line with the benchmarks set by the datasets and benchmark track.
> > >
> > > **(4) Models vs. Methods Clarification:** I urge you to update the results and tables to encapsulate this pivotal information. Comparing DRO with distinct backbones like MLP and FT-Transformer, under the generalized DRO category, might lead to misconceptions. Similarly, architectures such as MLP, FT-Transformer, NODE, and SAINT should clearly be indicated if trained under the ERM setup. Proper delineation between the architecture and the OOD generalization method employed during training is imperative for the paper's correctness.
> > >
> > > **(6) Expanded Related Work:** Incorporating such invaluable references within the review deadline would have significantly assisted in more accurately evaluating the work's merit.
> > >
> > > **(7) Codebase Refinement** and **(8) Appendix Refinement** require a more meaningful review; thus I will follow up on them later.

---

> > > ### Author Response · Authors · 2023-08-25
> > > **Updated response with requested additional results**
> > >
> > > As a follow up to our discussion on “(4) Models vs. Methods Clarification” in their original review, we have run some additional experiments to explore the use of “hybrid methods” which combine “models” and “methods” not explored in our work (nor in any published study, of which we are aware).
> > >
> > > Below, we selected three models:  the highest-performing (1) Transformer-based model (FT-Transformer), (2) tree-based model (NODE), and (3) “baseline” model (tabular ResNet). F We combined each model with the highest-performing robust learning method (Group DRO). Note that we are limited only to differentiable models for any robust learning method (since these are implemented as robust losses over continuous, differentiable functions only). These selections were sampled to combine a diverse set of models with a high-performing robustness method, and were based on the PMA-OOD averaged across all datasets (Figure 4).
> > >
> > > We provide the results in the table below. Additionally, for comparison, we provide a scatter plot for each dataset below, plus a summary violin plot summarizing the results. The scatter plots are formatted similarly to our main results from the paper, where we highlight the y=x line to show the case with zero shift gap. The table and scatter plot points have Clopper-Pearson CIs with alpha=0.05 computed for both the in- and out-of-distribution accuracy. The violin plot is drawn using the standard settings in matplotlib, and we overlay the actual data points as a reference. (We acknowledge that these plots can be difficult to parse, but are providing them in addition to the table in case they are useful for comparison to the paper.)
> > >
> > > Our results are shown in the figure and table below.
> > > * The impact of hybrid methods is minimal. While individual points may show some shifting, there is a great deal of variability over the experiments (since each reflects 100 hyperparameter tuning samples under HyperOpt), and we believe that it is best to interpret the results in terms of the overall trend. We show one such trend in the final subplot, which demonstrates that the overall distribution of the shift gap (∆_Acc; Equation (1)) does not change with hybrid vs. non-hybrid methods (in fact, hybrid methods have a larger average ∆_Acc: 0.0543 with ERM vs. 0.066 for Group DRO in the provided results).
> > > * Our main observations still hold. In particular, there is no change to the overall linear trend between in- and out-of-distribution accuracy, taken collectively. For each dataset, the new results fall along the trend lines for the original datasets shown in Figure 2 (there are no outliers with respect to the original observations, with ERM, for each model).
> > >
> > > We have also added these hybrid methods to the benchmark toolkit, so future work in the research community can also leverage these (and other) combinations, and are grateful to the reviewer for the suggestion to explore this direction.

---

> > > > ### Author Response · Authors · 2023-08-25
> > > > **Updated response with requested additional results [table]**
> > > >
> > > > | Task                 | Base Estimator  | Method    | ID Test Accuracy | ID Test Accuracy Interval | OOD Test Accuracy | OOD Test Accuracy Interval |
> > > > |----------------------|-----------------|-----------|------------------|---------------------------|------------------|---------------------------|
> > > > | College Scorecard    | FT-Transformer  | ERM       | 0.948            | (0.944, 0.952)            | 0.859            | (0.839, 0.877)            |
> > > > |                      |                 | Group DRO | 0.935            | (0.93, 0.939)             | 0.815            | (0.793, 0.835)            |
> > > > |                      | NODE            | ERM       | 0.944            | (0.939, 0.948)            | 0.844            | (0.823, 0.863)            |
> > > > |                      |                 | Group DRO | 0.946            | (0.942, 0.95)             | 0.835            | (0.814, 0.854)            |
> > > > |                      | ResNet          | ERM       | 0.947            | (0.943, 0.951)            | 0.854            | (0.834, 0.872)            |
> > > > |                      |                 | Group DRO | 0.947            | (0.942, 0.95)             | 0.824            | (0.803, 0.844)            |
> > > > | Food Stamps          | FT-Transformer  | ERM       | 0.843            | (0.841, 0.846)            | 0.816            | (0.812, 0.819)            |
> > > > |                      |                 | Group DRO | 0.826            | (0.823, 0.829)            | 0.795            | (0.792, 0.799)            |
> > > > |                      | NODE            | ERM       | 0.849            | (0.847, 0.852)            | 0.822            | (0.819, 0.825)            |
> > > > |                      |                 | Group DRO | 0.845            | (0.842, 0.847)            | 0.822            | (0.819, 0.825)            |
> > > > |                      | ResNet          | ERM       | 0.843            | (0.84, 0.845)             | 0.82             | (0.817, 0.824)            |
> > > > |                      |                 | Group DRO | 0.848            | (0.846, 0.851)            | 0.818            | (0.815, 0.822)            |
> > > > | Hypertension         | FT-Transformer  | ERM       | 0.666            | (0.661, 0.672)            | 0.604            | (0.603, 0.605)            |
> > > > |                      |                 | Group DRO | 0.665            | (0.659, 0.67)             | 0.608            | (0.607, 0.609)            |
> > > > |                      | NODE            | ERM       | 0.67             | (0.664, 0.676)            | 0.597            | (0.596, 0.599)            |
> > > > |                      |                 | Group DRO | 0.671            | (0.665, 0.676)            | 0.592            | (0.591, 0.593)            |
> > > > |                      | ResNet          | ERM       | 0.667            | (0.661, 0.672)            | 0.608            | (0.606, 0.609)            |
> > > > |                      |                 | Group DRO | 0.663            | (0.658, 0.669)            | 0.59             | (0.589, 0.592)            |
> > > > | Voting               | FT-Transformer  | ERM       | 0.879            | (0.848, 0.906)            | 0.855            | (0.841, 0.868)            |
> > > > |                      |                 | Group DRO | 0.894            | (0.865, 0.919)            | 0.858            | (0.844, 0.87)             |
> > > > |                      | NODE            | ERM       | 0.885            | (0.854, 0.911)            | 0.851            | (0.838, 0.864)            |
> > > > |                      |                 | Group DRO | 0.898            | (0.869, 0.923)            | 0.86             | (0.847, 0.873)            |
> > > > |                      | ResNet          | ERM       | 0.887            | (0.856, 0.912)            | 0.836            | (0.822, 0.849)            |
> > > > |                      |                 | Group DRO | 0.898            | (0.869, 0.923)            | 0.847            | (0.833, 0.861)            |
> > > > | Childhood Lead       | FT-Transformer  | ERM       | 0.971            | (0.961, 0.979)            | 0.92             | (0.915, 0.925)            |
> > > > |                      |                 | Group DRO | 0.971            | (0.961, 0.979)            | 0.92             | (0.915, 0.925)            |
> > > > |                      | NODE            | ERM       | 0.971            | (0.961, 0.979)            | 0.92             | (0.915, 0.925)            |
> > > > |                      |                 | Group DRO | 0.971            | (0.961, 0.979)            | 0.92             | (0.915, 0.925)            |
> > > > |                      | ResNet          | ERM       | 0.971            | (0.961, 0.979)            | 0.92             | (0.915, 0.925)            |
> > > > |                      |                 | Group DRO | 0.971            | (0.961, 0.979)            | 0.92             | (0.915, 0.925)            |

---

> > > > > ### Author Response · Authors · 2023-08-25
> > > > > **Updated response with requested additional results [link to figure]**
> > > > >
> > > > > https://github.com/mlfoundations/tableshift/blob/main/img/supplementary_figures_for_review.png

---

> > > > > > ### Author Response · Authors · 2023-08-26
> > > > > > **Follow up response to reviewer yyVo**
> > > > > >
> > > > > > We appreciate the reviewers’ thoughtful and detailed responses. We address each section below:
> > > > > >
> > > > > > * **(1) In-depth Data Analysis:** All of the prediction tasks in TableShift are binary prediction tasks. This may also clear up the reviewers’ concern about what is meant by “proportion of positive labels”. We would be happy to include KDE plots if the reviewer would clarify what kind of visualization they would find useful; it is currently unclear to us what a plot would add given that the label distribution is fully summarized with the values shown in Table 6. We have corrected the mismatched name between Table 1 and 6 (“Voting” was referred to as “ANES” in the previous version of Table 6; the data source for the Voting task is the American National Election Survey (ANES)).
> > > > > >
> > > > > >   We have added further description of the shift gap to the Table 1 caption as suggested.
> > > > > >
> > > > > >   The reviewer notes that further insight into “the underlying reasoning for achieving generalization across these shifts would provide valuable insights”. We agree, and this is an active area of ongoing research. However, developing methods to describe such shifts is beyond the scope of our benchmarking study -- but TableShift enables the research community to address these questions. We note that even quantifying and describing distribution shift is an open area of research; as one example, see the concurrent work https://arxiv.org/abs/2307.05284.
> > > > > >
> > > > > > * **(2) Clarification of Dataset Selection:** The shift for each dataset is described in supplementary section A. Each dataset is a separate subsection in Section A, and under the heading “Distribution Shift” you will find details regarding the exact nature of the shift. Additionally, the other headings (“Background”, “Data Source”) may also be useful to the reviewer in understanding the exact nature of the shifts and their motivation.
> > > > > >
> > > > > > * **(3) Improved Report Presentation:** We agree with the reviewer that a complete and easier-to-digest presentation of the (ID, O`OD) accuracy pairs would be useful. We acknowledge that our description of the previous Table 5 was unclear; the OOD accuracy values in that table did correspond to the points shown in Figure 1 (while they were described as “Max OOD Accuracy”, they should have been more clearly described as “Max OOD Accuracy associated with the best ID accuracy”; this not only matches the results shown in Figure 1 but reflects the task addressed in our work where a model is selected based on ID performance without knowledge of the OOD shift).
> > > > > >
> > > > > >   To address the reviewers’ feedback, we have added a new set of tables in Section D.3 (Tables 4-11) which give the complete set of (ID, OOD) pairs (as before) for each dataset, along with the 95% Clopper-Pearson confidence intervals for those metrics. As before, these pairs match exactly the data shown in Figures 1-3. We feel that this suggestion has improved the accessibility of the results and thank the reviewer.
> > > > > >
> > > > > >   The reviewer states that “while Table 5 lists multiple datasets with a max OOD accuracy approximating 0.4 for DRO, Figure 1 depicts only one such observation” -- we believe this may have been a misreading of the table. The results (previously in Table 5; now in Tables 4-11) show many *models* achieving OOD accuracy approximating 0.4 for DRO on the same *dataset*, FICO HELOC. This is visible in Table 6, and corresponds to the set of points near (0.74, 0.43) in Figure 1.
> > > > > >
> > > > > >   The reviewer notes that Figure 2 can be difficult to parse due to the scale. We acknowledge this concern -- our intention in Figure 2 is to use the same x-y scaling for all subfigures, to allow comparison across the subfigures -- and to address this, we have added an additional figure in the supplementary material, Figures 10/11 which contains the same data as Figures 2/3, but has more flexible scaling for each dataset to allow easier parsing of the results. Thank you for this suggestion.
> > > > > >
> > > > > >   We have also added the complete set of results (ID/OOD accuracy, plus confidence intervals) to the project github repo; these can be found in the `results` directory.
> > > > > >
> > > > > > * **(4) Models vs. Methods Clarification:** We have added this clarification to the paper in Section 4.1. To summarize, our experiments closely follow the methods of existing literature on DRO in only applying DRO to MLP models. All of the non-DRO models are trained with ERM. We are not aware of any works which apply DRO to these other model architectures, and thus felt it was most appropriate to follow this for our benchmarking study. Hopefully our results with the “hybrid methods”, posted after the reviewer’s follow-up response, also provide some evidence that the use of hybrid methods does not appear to have a substantial affect on robustness to the shifts in our benchmark.
> > > > > >
> > > > > > * **(6) Expanded Related Work:** We have added the requested references from the supplementary to the main text in Section 2.2.

---

> > > > > > > ### Comment · Reviewer_yyVo · 2023-08-28
> > > > > > >
> > > > > > > Thank you for your detailed response.
> > > > > > >
> > > > > > > **(0) Models x Methods Benchmark:** With regards to the new table discussing the Models x Methods benchmark, could you provide further clarity on your experimental design parameters? Specifically, details concerning the number of seeds and epochs would be beneficial.
> > > > > > >
> > > > > > > **(1) In-depth Data Analysis:** If every prediction task within TableShift pertains to binary predictions, then it's crucial to explicitly mention this in the manuscript. Neglecting such a vital piece of information could mislead readers, especially considering that applying the datasets to multiclass or regression tasks might yield different outcomes. The absence of this detail raises concerns about the specificity and scope of TableShift. Additionally, it would be valuable to know whether the API allows users to procure raw datasets for varied target specifications.
> > > > > > >
> > > > > > > **(3) Improved Report Presentation:** To navigate this section with clarity, I'd like to address concerns sequentially. Initially, Table 1 specifies "In-domain (ID) and out-of-domain (OOD) accuracy show a linear trend across **15 TableShift tasks** and **19 model types**." This assertion contrasts with the transition of content from Table 5 to Tables 4-11. The math indicates a shortfall in tasks (with 11-4 resulting in only 7 tasks covered). Further scrutiny of the previous version reveals that in Table 5, the DRO demonstrated ID/OOD performance metrics of (0.745, 0.431) on the FICO HELOC dataset and (0.598, 0.416) on the Hypertension dataset. Consequently, I would anticipate observing at least two DRO points proximate to the 0.4 OOD performance threshold, which is not available in the current representation. Could you elucidate this discrepancy? I must underscore that the former Table 5 encompassed 15 tasks and 19 models, which the current Tables 4-11 fail to encompass.

---

> > > > > > > > ### Author Response · Authors · 2023-08-28
> > > > > > > >
> > > > > > > > Once again, we sincerely appreciate the reviewers’ thoughtful engagement with our work and responses!
> > > > > > > >
> > > > > > > > * **(0) Models x Methods Benchmark:** For these experiments, we follow the exact same procedure as the rest of the experiments in the paper (described in Section 4.2). Specifically, we run 100 iterations of hyperparameter tuning, with the combined hyperparameter grid of the existing model, plus the Group DRO tunable parameter (the only tunable parameter for Group DRO is the group weights step size; the exact range and distribution of all hyperparameters is shown in Table 18). As in our other experiments, the number of epochs is treated as a tunable parameter.
> > > > > > > >
> > > > > > > > * **(1) In-depth Data Analysis:** We have added explicit mention of the binary nature of Tableshift’s classification tasks in the abstract, introduction, and section 3.1.
> > > > > > > >
> > > > > > > > * **(3) Improved Report Presentation:**
> > > > > > > >   * “This assertion contrasts with the transition of content from Table 5 to Tables 4-11. The math indicates a shortfall in tasks (with 11-4 resulting in only 7 tasks covered).”
> > > > > > > >
> > > > > > > >     Tables 4-11 contain two tasks each (separated by a vertical dividing line), except for Table 11 which contains the final task, to reduce the number of pages in the manuscript, for a total of 15 tasks. All 19 model types are shown in each Table 4-11. If the reviewer feels that another layout would be preferable, we are happy to reformat these tables as needed or adjust the captions, hyperlinks, etc to improve clarity, provided sufficient time remains in the response window (or during the camera-ready phase).
> > > > > > > >
> > > > > > > >   * “Further scrutiny of the previous version reveals that in Table 5, the DRO demonstrated ID/OOD performance metrics of (0.745, 0.431) on the FICO HELOC dataset and (0.598, 0.416) on the Hypertension dataset. Consequently, I would anticipate observing at least two DRO points proximate to the 0.4 OOD performance threshold, which is not available in the current representation. Could you elucidate this discrepancy?”
> > > > > > > >
> > > > > > > >     The reviewer asks about two different cases.
> > > > > > > >
> > > > > > > >       * First, they note that “the DRO demonstrated ID/OOD performance metrics of (0.745, 0.431) on the FICO HELOC dataset [...] I would anticipate observing at least two DRO points proximate to the 0.4 OOD performance threshold”. This is correct. Indeed, many models achieve very similar performance on the FICO HELOC dataset in our study, as is shown in the results table. This has the consequence that there are many points located in a very close region of the (ID, OOD) space -- this is visible in Figure 1 as many markers overlaid at approximately (0.745, 0.431). The results in the Figure match the table exactly (both in the previous revisions, and in the current version).
> > > > > > > >       * Second, the reviewer asks about the case of DRO achieving (0.598, 0.416) on the Hypertension dataset. That reading of results is correct, and this point is shown in the plot in Figure 1 at (0.598, 0.416). We do not see a discrepancy here -- but we are also happy to clarify further if this does not address the reviewers’ question.

---

> > > > > > > > > ### Comment · Reviewer_yyVo · 2023-08-29
> > > > > > > > >
> > > > > > > > > Thanks again to the authors for the detailed response!
> > > > > > > > > - **(0) Models x Methods Benchmark**: I observed that such a setup does not seem to incorporate the multi-seed evaluation protocol, which has been standard practice in other domain generalization studies [1,2,3,4]. Does the benchmark indeed evaluate each hyperparameter optimization trial only once?
> > > > > > > > > - **(1) In-depth Data Analysis**: Could you elucidate further on the API's capabilities? Specifically, is there flexibility in defining alternate targets within the TableShift framework?
> > > > > > > > > - **(3) Improved Report Presentation**: Upon closer examination, I noted that the performance metrics for the FICO HELOC dataset across all models are strikingly similar, corresponding to an indistinctive point in Figure 1. Could you perhaps shed light on any underlying reason for this uniformity? Furthermore, I would strongly advocate for the presentation of results for each dataset in separate tables. Consolidating results in this manner enhances clarity and facilitates a more straightforward discussion.
> > > > > > > > >
> > > > > > > > >
> > > > > > > > > [1] In Search of Lost Domain Generalization
> > > > > > > > >
> > > > > > > > > [2] WILDS: A Benchmark of in-the-Wild Distribution Shifts
> > > > > > > > >
> > > > > > > > > [3] Wild-Time: A Benchmark of in-the-Wild Distribution Shift over Time
> > > > > > > > >
> > > > > > > > > [4] WOODS: Benchmarks for Out-of-Distribution Generalization in Time Series

---

> > > > > > > > > > ### Author Response · Authors · 2023-08-31
> > > > > > > > > >
> > > > > > > > > > Thanks to the reviewer for the follow up and for the effort to fully understand our work.
> > > > > > > > > >
> > > > > > > > > > * **(0) Models x Methods Benchmark:**
> > > > > > > > > >   * We conduct one trial of 100 hyperparameter sampling iterations for each dataset, according to the protocol described in Section 4.2. We use one random seed for each run. We agree that it would be ideal to run our protocol multiple times. We also note that training the 23,700 models in our main results alone required 50k GPU-hours.
> > > > > > > > > >   * In order to provide some evidence regarding variability over random seeds, we provide a set of additional experimental results in a separate post below to address the reviewers’ concern. These results suggest that variation in both ID and OOD accuracy tends to be small, and the intervals overlap substantially across random seeds.
> > > > > > > > > >
> > > > > > > > > > * **(1) In-depth Data Analysis:**
> > > > > > > > > > > “[I]s there flexibility in defining alternate targets within the TableShift framework?”
> > > > > > > > > >   * Since TableShift is a benchmarking toolkit, we aimed to provide an interface to a specific set of benchmark tasks. The API does not currently support tasks other than the benchmark prediction tasks. Since the framework is fully open-source, users are free to adapt it as desired.
> > > > > > > > > >
> > > > > > > > > > * **(3) Improved Report Presentation:**
> > > > > > > > > > > “Upon closer examination, I noted that the performance metrics for the FICO HELOC dataset across all models are strikingly similar, corresponding to an indistinctive point in Figure 1. Could you perhaps shed light on any underlying reason for this uniformity?”
> > > > > > > > > >   * We cannot conclusively determine the cause of this uniformity. However, we note that this finding is consistent with that of Rudin et al., who showed that properly-tuned simple models could perform competitively with more complex models on this dataset. We hypothesize that this uniformity could be due to the underlying credit scoring algorithm being dependent on the factors being used here (i.e., the credit score may be a series of rules or thresholds applied to certain features, which can be learned by many models). However, we cannot verify this claim, as the scoring algorithms are proprietary.
> > > > > > > > > >
> > > > > > > > > >   > “Furthermore, I would strongly advocate for the presentation of results for each dataset in separate tables. Consolidating results in this manner enhances clarity and facilitates a more straightforward discussion.”
> > > > > > > > > >   * We will gladly incorporate this suggestion. To avoid too many additional changes to table numbering prior to the review deadline and to facilitate communication with the other reviewers, we hope the reviewer would be willing to allow us to leave the tables in their current form until the camera-ready preparation phase, as this is strictly a formatting suggestion and the data itself will not change.

---

> > > > > > > > > > > ### Author Response · Authors · 2023-08-31
> > > > > > > > > > > **Follow-up on "(0) Models x Methods Benchmark: results with additional random seeds"**
> > > > > > > > > > >
> > > > > > > > > > > Below, in order to address the reviewers’ question about the impact of multiple random seeds, we provide results from a set of multiple trials using different random seeds, on a subset of TableShift tasks.  Following  [1, 2, 3, 4], we conduct three separate trials with different random seeds. We select a subset of the highest-performing models (measured by average PMA-OOD across tasks; Figure 4a), along with MLP, which we believe to be a relevant baseline due to its use in combination with several robust learning methods in our work.
> > > > > > > > > > >
> > > > > > > > > > > For each random seed, we perform the exact same protocol described in Section 4.2: we perform 100 iterations of hyperparameter tuning from the full grid for each model, and select the best model according to the (in-distribution) validation accuracy. We compute 95% Clopper-Pearson confidence intervals for both the ID and OOD accuracy. All values are rounded to 5 decimal places. Note that the first iteration, 0, of each row is the original result from our paper, since it follows the same protocol.
> > > > > > > > > > >
> > > > > > > > > > > The results show strong consistency across models. Indeed, we did not identify any cases where the ID or OOD confidence intervals did not all overlap at some region. These provide further support that our main results are not likely to be affected by varying the random seed, and demonstrate the utility of using large test sets and conducting rigorous hyperparameter tuning.
> > > > > > > > > > >
> > > > > > > > > > > Not all experiments with FT-Transformer were able to complete in time, due to the significantly higher compute cost of Transformer-based models. However, we provide the results for FT-Transformer we were able to obtain.
> > > > > > > > > > >
> > > > > > > > > > >
> > > > > > > > > > > | Task              | Estimator       | Iteration | ID Test Accuracy | ID Test Accuracy 95% Clopper Pearson CI | OOD Test Accuracy 95% Cloper Pearson CI | OOD Test Accuracy 95% Clopper Pearson CI |
> > > > > > > > > > > |-------------------|-----------------|-----------|------------------|---------------------------------------|----------------------------------------|----------------------------------------|
> > > > > > > > > > > | College Scorecard | CatBoost        | 0         | 0.9575           | (0.95367, 0.96089)                    | 0.8846                                 | (0.86637, 0.90116)                     |
> > > > > > > > > > > |                   |                 | 1         | 0.9585           | (0.95485, 0.96198)                    | 0.8794                                 | (0.86088, 0.89633)                     |
> > > > > > > > > > > |                   |                 | 2         | 0.9592           | (0.95553, 0.9626)                     | 0.8817                                 | (0.86323, 0.8984)                      |
> > > > > > > > > > > |                   | FT-Transformer  | 0         | 0.9478           | (0.94373, 0.95167)                    | 0.8587                                 | (0.83901, 0.87687)                     |
> > > > > > > > > > > |                   |                 | 1         | 0.9459           | (0.94172, 0.94979)                    | 0.8499                                 | (0.82968, 0.86849)                     |
> > > > > > > > > > > |                   |                 | 2         | 0.9404           | (0.9361, 0.94454)                     | 0.8299                                 | (0.80877, 0.84955)                     |
> > > > > > > > > > > |                   | LightGBM        | 0         | 0.9393           | (0.93492, 0.94344)                    | 0.8217                                 | (0.79951, 0.84109)                     |
> > > > > > > > > > > |                   |                 | 1         | 0.9426           | (0.93836, 0.94666)                    | 0.8395                                 | (0.81883, 0.85868)                     |
> > > > > > > > > > > |                   |                 | 2         | 0.9430           | (0.93878, 0.94705)                    | 0.8365                                 | (0.81573, 0.85587)                     |
> > > > > > > > > > > |                   | MLP             | 0         | 0.9466           | (0.94247, 0.9505)                     | 0.8454                                 | (0.82503, 0.86429)                     |
> > > > > > > > > > > |                   |                 | 1         | 0.9485           | (0.94449, 0.95237)                    | 0.8595                                 | (0.83979, 0.87757)                     |
> > > > > > > > > > > |                   |                 | 2         | 0.9448           | (0.94063, 0.94877)                    | 0.8587                                 | (0.83901, 0.87687)                     |
> > > > > > > > > > > |                   | XGBoost         | 0         | 0.9420           | (0.93769, 0.94603)                    | 0.8303                                 | (0.80877, 0.84955)                     |
> > > > > > > > > > > |                   |                 | 1         | 0.9460           | (0.9418, 0.94987)                     | 0.8417                                 | (0.82115, 0.86078)                     |
> > > > > > > > > > > |                   |                 | 2         | 0.9472           | (0.94306, 0.95104)                    | 0.8447                                 | (0.82425, 0.86359)                     |

---

> > > > > > > > > > > > ### Author Response · Authors · 2023-08-31
> > > > > > > > > > > > **[additional results table continued, dataset 2/3]**
> > > > > > > > > > > >
> > > > > > > > > > > > | Task              | Estimator       | Iteration | ID Test Accuracy | ID Test Accuracy 95% Clopper Pearson CI | OOD Test Accuracy 95% Cloper Pearson CI | OOD Test Accuracy 95% Clopper Pearson CI |
> > > > > > > > > > > > |-------------------|-----------------|-----------|------------------|---------------------------------------|----------------------------------------|----------------------------------------|
> > > > > > > > > > > > | Food Stamps       | CatBoost        | 0         | 0.8494           | (0.84692, 0.85194)                    | 0.8246                                 | (0.82122, 0.8280)                      |
> > > > > > > > > > > > |                   |                 | 1         | 0.8499           | (0.84737, 0.85238)                    | 0.8241                                 | (0.82073, 0.8275)                      |
> > > > > > > > > > > > |                   |                 | 2         | 0.8493           | (0.84676, 0.85177)                    | 0.8237                                 | (0.82025, 0.82703)                     |
> > > > > > > > > > > > |                   | FT-Transformer  | 0         | 0.8432           | (0.84062, 0.84572)                    | 0.8156                                 | (0.81218, 0.81907)                     |
> > > > > > > > > > > > |                   |                 | 1         | 0.8444           | (0.84188, 0.84696)                    | 0.8171                                 | (0.8136, 0.82047)                      |
> > > > > > > > > > > > |                   | LightGBM        | 0         | 0.8359           | (0.83327, 0.83846)                    | 0.8082                                 | (0.80462, 0.81162)                     |
> > > > > > > > > > > > |                   |                 | 1         | 0.8436           | (0.84101, 0.8461)                     | 0.8179                                 | (0.81448, 0.82135)                     |
> > > > > > > > > > > > |                   |                 | 2         | 0.8430           | (0.84047, 0.84557)                    | 0.8171                                 | (0.81364, 0.82051)                     |
> > > > > > > > > > > > |                   | MLP             | 0         | 0.8410           | (0.83841, 0.84354)                    | 0.8151                                 | (0.81164, 0.81854)                     |
> > > > > > > > > > > > |                   |                 | 1         | 0.8448           | (0.84226, 0.84734)                    | 0.8172                                 | (0.81378, 0.82066)                     |
> > > > > > > > > > > > |                   |                 | 2         | 0.8438           | (0.84127, 0.84635)                    | 0.8112                                 | (0.80768, 0.81464)                     |
> > > > > > > > > > > > |                   | XGBoost         | 0         | 0.8444           | (0.84187, 0.84695)                    | 0.8205                                 | (0.81706, 0.82389)                     |
> > > > > > > > > > > > |                   |                 | 1         | 0.8428           | (0.84028, 0.84538)                    | 0.8188                                 | (0.81533, 0.82218)                     |
> > > > > > > > > > > > |                   |                 | 2         | 0.8449           | (0.84231, 0.84739)                    | 0.8198                                 | (0.8164, 0.82324)                      |

---

> > > > > > > > > > > > > ### Author Response · Authors · 2023-08-31
> > > > > > > > > > > > > **[additional results table continued, dataset 3/3]**
> > > > > > > > > > > > >
> > > > > > > > > > > > > | Task              | Estimator       | Iteration | ID Test Accuracy | ID Test Accuracy 95% Clopper Pearson CI | OOD Test Accuracy 95% Cloper Pearson CI | OOD Test Accuracy 95% Clopper Pearson CI |
> > > > > > > > > > > > > |-------------------|-----------------|-----------|------------------|---------------------------------------|----------------------------------------|----------------------------------------|
> > > > > > > > > > > > > | Hypertension      | CatBoost        | 0         | 0.6704           | (0.66477, 0.67601)                    | 0.5987                                 | (0.59741, 0.60008)                     |
> > > > > > > > > > > > > |                   |                 | 1         | 0.6708           | (0.66511, 0.67635)                    | 0.5985                                 | (0.59721, 0.59988)                     |
> > > > > > > > > > > > > |                   |                 | 2         | 0.6713           | (0.66567, 0.6769)                     | 0.5997                                 | (0.59834, 0.60101)                     |
> > > > > > > > > > > > > |                   | FT-Transformer  | 0         | 0.6662           | (0.66051, 0.67178)                    | 0.6042                                 | (0.60282, 0.60548)                     |
> > > > > > > > > > > > > |                   | LightGBM        | 0         | 0.6776           | (0.67201, 0.68319)                    | 0.6341                                 | (0.63276, 0.63538)                     |
> > > > > > > > > > > > > |                   |                 | 1         | 0.6716           | (0.66596, 0.67719)                    | 0.6361                                 | (0.6348, 0.63742)                      |
> > > > > > > > > > > > > |                   |                 | 2         | 0.6722           | (0.66659, 0.67782)                    | 0.6281                                 | (0.62678, 0.62941)                     |
> > > > > > > > > > > > > |                   | MLP             | 0         | 0.6641           | (0.65846, 0.66976)                    | 0.5831                                 | (0.58177, 0.58445)                     |
> > > > > > > > > > > > > |                   |                 | 1         | 0.6687           | (0.66307, 0.67432)                    | 0.5974                                 | (0.59605, 0.59873)                     |
> > > > > > > > > > > > > |                   |                 | 2         | 0.6676           | (0.66199, 0.67325)                    | 0.5980                                 | (0.59663, 0.5993)                      |
> > > > > > > > > > > > > |                   | XGBoost         | 0         | 0.6711           | (0.66544, 0.67668)                    | 0.5885                                 | (0.58713, 0.58981)                     |
> > > > > > > > > > > > > |                   |                 | 1         | 0.6694           | (0.66374, 0.67498)                    | 0.5856                                 | (0.58422, 0.58691)                     |
> > > > > > > > > > > > > |                   |                 | 2         | 0.6694           | (0.66374, 0.67498)                    | 0.5856                                 | (0.58422, 0.58691)                     |

---

> > > > > > > > > > > > > > ### Comment · Reviewer_yyVo · 2023-08-31
> > > > > > > > > > > > > >
> > > > > > > > > > > > > > Thank you for providing additional details. Nonetheless, my initial concerns largely remain unaddressed. It's unclear what kinds of distribution shifts occur, and how severe they are ((1) In-depth Data Analysis). The evaluation, anchored to a single seed, lacks the comprehensiveness of the approach in [1] ((0) Models x Methods Benchmark). Additionally, the conflation of methods and model evaluations yields results that may be challenging to interpret and compare ((4) Models vs. Methods Clarification). Finally, The extensive use of datasets and the visual representation of results make it particularly arduous for subsequent studies to draw meaningful comparisons ((2) Clarification of Dataset Selection and (3) Improved Report Presentation).
> > > > > > > > > > > > > >
> > > > > > > > > > > > > > While I appreciate the underlying intentions behind this work, the results appear convoluted due to an indiscriminate juxtaposition of diverse elements. To the best of my understanding, this complicates the path forward for future research. Notably, the benchmark predominantly serves binary classification problems, though the name "TableShift" suggests a broader applicability.
> > > > > > > > > > > > > >
> > > > > > > > > > > > > > I would recommend the authors either sharpen their focus or segment the paper's findings into distinct, well-defined areas.
> > > > > > > > > > > > > >
> > > > > > > > > > > > > > - [1] Gulrajani, Ishaan, and David Lopez-Paz. ‘In Search of Lost Domain Generalization’. arXiv, 2 July 2020. [http://arxiv.org/abs/2007.01434](http://arxiv.org/abs/2007.01434)

---

### Official Review · Reviewer_ZSf5 · 2023-07-20
**A relevant benchmark for out-of-distribution evaluation of tabular machine learning**

**Rating:** 7
**Confidence:** 5

**Strengths:**

- The paper addresses the relevant problem of distribution shift in tabular data and offers a great foundation for future endeavors in improving out-of-distribution accuracy in tabular data. Although the problem of out-of-distribution accuracy has been addressed in computer vision, this paper offers a new benchmark for tabular data.

- The domains in which the datasets are selected are relevant. For instance, finance is a domain where distribution shifts occur.


**Additional Feedback:**

- The shape of the paper can be improved, especially the figures. The figures are not ordered: Figure 4 is placed after Figure 5. Furthermore, Figure 5 is referenced before Figures 2 and 3. Figures 2 and 3 are referenced by the keyword “Figs” which makes it hard to find (especially considering the aforementioned misplacements).

- Figures 2 and 3 are hard to read for micro comparison, even on a computer screen, though they are suitable to identify the trend.

- The results of Figure 7 are described in the main paper but the Figure is only present in the supplementaries. We recommend adding it to the main paper.


**Clarity:**

The paper is generally well written. However, the clarity of the presentation can be improved (see Additional Feedback)

**Correctness:**


- The dataset construction is sound.

- The evaluation methods and experiment design are sound


**Documentation:**

- The paper provides in appendices, for each dataset, the detail of its construction for the task of out-of-distribution evaluation. In particular, it explains the choice of the feature on which the in/out-of-distribution split is made.

- The authors provide a well-documented code repository that gives enough details to reproduce the benchmark.

- The datasets are downloaded from various sources that may not be permanent. Having a copy of the dataset (for which the license allows it) and a permalink would guarantee long-term availability.


**Ethics:**

No ethical concerns.

**Limitations:**

- The paper clearly states the limitation of some datasets that do not have 2 or more training domains which makes domain generalization models unusable.

- It also shows the limitation of the evaluated out-of-distribution methods in the experimental section.


**Opportunities For Improvement:**

There is an opportunity to extend the dataset collection in the domain of computer security. For instance, [1] and [2] propose datasets with the task of detecting malware and botnet respectively. In computer security, the attacker constantly evolves its strategy. Therefore, there is a strong need for out-of-distribution generalization.

[1] Alesia Chernikova and Alina Oprea. Fence: Feasible evasion attacks on neural
networks in constrained environments.

[2]  Hojjat Aghakhani, Fabio Gritti, Francesco Mecca, Martina Lindorfer, Stefano Ortolani,
Davide Balzarotti, Giovanni Vigna, and Christopher Kruegel. When malware is packin’heat; limits of machine learning classifiers based on static analysis features.


**Relation To Prior Work:**

The paper bridges the gap between out-of-distribution studies in the domain of computer vision and NLP and tabular data Machine Learning.

**Summary And Contributions:**

This paper proposes a benchmark for distribution shifts in tabular data.

It compiles 15 datasets in various domains (e.g. finance, health, education) including 10 that are composed of multiple sub-domain. The presence of sub-domains in this dataset is crucial to evaluate the generalization of models to distribution shift and this constitutes a contribution on its own.

The paper studies the in-distribution and out-distribution accuracy of 19 architecture and training methods, including classical ML models (e.g. random forest), neural networks (e.g. SAINT), and state-of-the-art domain generalization methods (e.g. DRO).

In addition, the authors package their work in a comprehensive and open-source framework that contributes to open science.

---

> ### Author Response · Authors · 2023-08-22
> **Response to Reviewer ZSf520**
>
> We are grateful for the reviewers’ feedback, and in particular of their recognition that “dataset construction is sound,” “evaluation methods and experiment design are sound,” and that the paper is well-written.
>
> The reviewer shared several pieces of useful feedback regarding the clarity of presentation or “shape of the paper”. We have incorporated all of the reviewer’s suggestions. In particular, we corrected the positioning and numbering of the relevant Figures. In the caption for each figure, we also added explicit references to tables containing the exact data values for that figure. In particular, Table 6 lists the ID/OOD values for each method and dataset. As the reviewer notes, of course, the scatter plots contain quite a bit of data and are mostly intended to serve as general illustrations of a trend, not precise readouts of data points. The detailed results in the supplement corresponding to each figure will serve as a reference for precise values.
>
> The reviewer also makes an excellent suggestion regarding hosting of the datasets. We will consult the licenses of the various datasets, but where available, we will not only plan to host the original dataset, but also a preprocessed version, in a public repository (Hugging Face Datasets) to accompany the benchmark. This will have the added benefit of reducing the computational overhead of working with the large public datasets that require preprocessing. However, since the track call for papers strongly emphasizes the importance of authors' honoring the original data license, we would appreciate the extra time to review each dataset's license in order to determine whether this is a possible route prior to camera-ready.

---

> > ### Comment · Reviewer_ZSf5 · 2023-08-30
> >
> > I thank the authors for their answer.
> >
> > I believe that guaranteeing long-term access to the datasets is essential for the long-term impact of your work. I appreciate the challenge you raised regarding licensing etc. A reasonable solution would be to host the datasets with permissive license. This would solve the problem partially.

---

> > > ### Author Response · Authors · 2023-08-31
> > >
> > > We are grateful for the reviewers' emphasis on ensuring long-term impact of our work! We are eager to ensure this as well. We are also eager to do so in a way that follows the licenses and restrictions of the datasets comprising the benchmark.
> > >
> > > Ultimately, we are bound by the license of each of the original data sources -- we cannot give a permissive license to datasets which already have restrictive licenses. We appreciate the reviewers' understanding of this issue. We can assure the reviewer that, prior to the camera-ready deadline, we will make publicly available preprocessed versions of any datasets which have licenses that allow for such sharing; however, reviewing the dataset licenses takes time and consultation with experts in this area.

---

### Official Review · Reviewer_9358 · 2023-07-21
**Significant contributions to the OOD problem in tabular data.**

**Rating:** 6
**Confidence:** 4
**Correctness:** Yes.
**Clarity:** The paper is well-written and organized.

**Strengths:**

1. This work contributes significantly to domain shift in tabular data, which is a very important research topic but gains not enough attention.
2. The experiments are extensive. They conduct experiments on 5 main categories of models and 15 tasks.
3. The paper is very well organized and written.

**Additional Feedback:**

None.

**Documentation:**

The benchmark code is provided.

**Ethics:**

None.

**Limitations:**

See the details in "Opportunities For Improvement".

**Opportunities For Improvement:**

1. The test-time adaptation can be considered in this framework as an extension.
2. For the API usage, authors can add some figures to illustrate the API use more clearly.
3. The limitation of this paper should be discussed in a more direct way.

**Relation To Prior Work:**

There is a corresponding discussion of the related work.

**Summary And Contributions:**

This paper presents a benchmark for domain shift in tabular data. The contribution of this paper can be summarized in three folds. The first is a TabularShift benchmark. The second is extensive experiments on the domain shift in tabular data. The third is publicly accessible API and baselines.

---

> ### Author Response · Authors · 2023-08-22
> **Response to Reviewer 935820**
>
> Thanks to the reviewer for their feedback; we are glad to see the reviewer recognizes our three main contributions in this area “which is a very important research topic but gains not enough attention” -- we agree!
>
> * **Test-time adaptation:** We agree that test-time adaptation is potentially relevant to this problem. However, we consider an investigation of test-time adaptation methods beyond the scope of our paper; this is why we frame the task in Section 2 as “we assume that no information about the target Dtest is available”. We hope that future work utilizing the benchmark conducts such studies and have already been engaging with colleagues in this area interested in using TableShift.
>
> * **Illustrating API use in the paper:** We recognize the reviewers’ feedback about more clearly illustrating the API use. We attempted to focus the paper on our experiments, datasets, and empirical results, as these will not change over time. However, we were reluctant to include specific examples of API use in the paper itself, in case the API evolves over time. So, we have focused our efforts on documenting the API on the github page hosting the project, and on the TableShift website. This ensures that the paper remains a reliable source of information over time, and allows us the flexibility to improve and evolve the API while being able to keep information about TableShift up to date. Please also see our response to reviewer XmM225 where we discuss several additional improvements made to the documentation, including improvements to the environment setup, instructions on benchmarking new algorithms and reproducing our experiments, and improved documentation in the code itself. We hope that these provide sufficient demonstration of the API use, while decoupling it from the paper itself.
>
> * **Limitations:** We will add an additional discussion of the limitations to the paper with the extra camera-ready page, with an explicit “limitations” section. In addition to our existing limitations, we plan to specifically add: (i) more datasets could always be added to the benchmark, and our results reflect the specific datasets included in our benchmark. More empirical validation is needed, including studies comparing our findings on the TableShift benchmark to other tabular distribution shifts. (ii) our work does not consider “hybrid” methods (which have never been tried in the literature, to our knowledge) and which may mitigate shift gaps. (iii) while our work demonstrates strong empirical evidence of various relationships (e.g. between label shift and shift gap), it does not establish theoretical connections between these. Future work in this direction would be useful, as would more general methods for measuring and understanding distribution shift beyond the metrics ($\Delta_y$, $\Delta_x$, $\Delta_{y|x}$) from our paper. If the reviewer has other specific limitations in mind beyond those here, we are also open to additional limitations to highlight in the paper.

---

> > ### Author Response · Authors · 2023-08-26
> >
> > As a follow up, we would like to note that we have added a "Limitations" section to the paper (Section 6 in the revision).

---

### Official Review · Reviewer_dJDE · 2023-07-21
**Paper fills important tabular gap in distribution shift**

**Rating:** 8
**Confidence:** 4

**Strengths:**

- The datasets are diverse and the shift gap is clearly documented. The criteria for inclusion is also clear.

- The paper provides a simple and uniform API for handling data loading and preprocessing, which is more complicated than most image-based pipelines.

- The paper is very clear about its experimental setup and the methods chosen, which are diverse and representative.

- The results are clearly explained with key takeways.



**Additional Feedback:**

- It might be interesting to have a leaderboard for each dataset like WILDS.

**Clarity:**

- Figure 1 is interesting but a bit confusing with all the symbols. I guess the main point is the color and overall scatterplot which is interesting.

- Typo on line 101.

- Tables and Figures should only be displayed after they are mentioned in the text (eg.., Table 1 should be on the bottom of the page or on the next page). Also table captions should be on top.


**Correctness:**

- Methods seem correct and reproducible.


**Documentation:**

- Benchmark is well-documented.

**Ethics:**

- N/A

**Limitations:**

- Perhaps discussing the computational cost of each method and/or dataset would help users know what they should test.


**Opportunities For Improvement:**

- No significant weaknesses.

**Minor:**
- What do you conjecture is the reason for label methods not eliminating the label gap?  Is it that there is some unknown confounding factor that changes both the label distribution and the covariate distribution?  Perhaps an experiment that artificially creates label gaps while the covariate distribution is the same might be interesting to disentangle this.

- Also, it might be helpful to have a subset of representative datasets that researchers could try before applying their method to all datasets, like 3-4 representative datasets that exhibit the diversity of the datasets as a whole.



**Relation To Prior Work:**

- Good comparison to prior shift benchmarks.


**Summary And Contributions:**

The paper presents a benchmark suite for distribution shifts in *tabular* datasets. The paper selects a diverse set of real tabular datasets and a diverse set of models to understand tabular shift. The proposed API makes dataset access simple. Finally, an extensive experimental comparison on datasets and methods is used to draw several high-level conclusions about tabular dataset shift.

---

> ### Author Response · Authors · 2023-08-22
> **Response to Reviewer dJDE21**
>
> We are grateful for the reviewers’ feedback and glad they found the work to be “very clear”, and the shift gaps to be “clearly documented”. We are also glad the reviewer highlighted that “[t]he datasets are diverse” and that we used “a diverse set of models” -- this was our goal, to bring together a diverse suite of both datasets and models to conduct a thorough initial comparison while also providing a benchmark for future comparisons in the tabular space.
>
> The reviewer asks: “What do you conjecture is the reason for label methods not eliminating the label gap?” We believe the answer is quite simple: there is “no free lunch” for achieving label robustness, without additional information (such as information about the shift, or unlabeled data from the test distribution). Since our experiments are concerned with the setting where there is no such additional information, these methods did not result in uniform gains. However, we still felt it was important to include relevant label-shift robustness baselines in order to confirm this intuition empirically, and given our emphasis on label shift in explaining shift gaps in the results. We will add a clearer discussion of this point to the conclusion section with the extra camera-ready page.
>
> We have addressed the reviewers’ feedback regarding formatting and positioning of tables and figures in the paper.

---

> > ### Comment · Reviewer_dJDE · 2023-08-25
> >
> > Thank you for the response. Do you have suggestions about a smaller subset of datasets for prototyping that represents the diversity of datasets?

---

> > > ### Author Response · Authors · 2023-08-26
> > >
> > > We certainly understand that using a smaller subset of datasets would be useful for speedy development or prototyping new methods. While it is difficult to recommend a “best” subset for development, if the emphasis is on diversity of datasets, we would specifically suggest considering diversity along the following axes:
> > > * Dataset size
> > > * Baseline Gap (values shown in Table 1)
> > > * Label Shift ($\Delta_y$, values shown in Table 3)
> > >
> > > Of course, diversity over domain would also be useful (some datasets are drawn from similar sources, such as the various datasets derived from MIMID). We would also suggest choosing publicly available datasets, as these are quickest to get started with.
> > >
> > > A subset that seems to meet these criteria might be: College Scorecard; FICO HELOC; Hospital Readmission; Voting (not publicly available, but easy to access via free/instant account creation); Public Health Insurance.
> > >
> > > If this does not provide the guidance the reviewer had in mind, please feel free to follow up and we can provide further suggestions or discussion.

---

> > > > ### Comment · Reviewer_dJDE · 2023-08-29
> > > >
> > > > Thanks for the followup. Yes, this sounds like a reasonable subset. Please consider adding this to the discussion or at least the appendix.

---

> > > > > ### Author Response · Authors · 2023-08-30
> > > > >
> > > > > Thank you for the suggestion -- we have added this to the "benchmarking guide" in the GitHub repo, as we feel it will be most useful to users of the TableShift package there!
> > > > >
> > > > > https://github.com/mlfoundations/tableshift/blob/main/benchmarking_guide.md#finding-a-suitable-subset-for-development

---

### Official Review · Reviewer_XmM2 · 2023-07-25
**Well written paper with a comprehensive benchmark filing a gap in the literature, but with reproducibility issues.**

**Rating:** 7
**Confidence:** 3
**Correctness:** To my knowledge, all claims are correct.
**Clarity:** Yes.

**Strengths:**

- Well-written paper
- Comprehensive benchmark with interesting conclusions
- Facilitates access to public datasets

**Additional Feedback:**

Comments:
- I will raise my score if more care is given to reproducibility (e.g. if I can install the environment), documentation, and instructions for running the benchmarks.
- Typo line 197 (pptimization)
- In Figure 1, the green point near (0.7, 0.42) has a marker that doesn't seem to be in the legend.
- In Figure 2, it seems hard to distinguish between MLP and catboost.

**Documentation:**

The documentation could be improved (see notes above in opportunities for improvement).

**Ethics:**

No ethics concerns.

**Limitations:**

I'm satisfied with the discussion of limitations and societal impact.

**Opportunities For Improvement:**

- I tried to install the conda environments on three different systems without success (Mac M1, Linux cluster with Rocky 8, Linux container from docker://continuumio/miniconda3); therefore, I couldn't test the package's functionality.
- At the very least, you should include the versions of the packages used for the conda environment.  A Docker container or similar would be ideal.
- I did not find instructions for reproducing the paper results. I can see that all the code is there, but it's unclear what command to run. Reproducibility is critical for this track.
- The paper will be way more impactful and support the development of new methods if clear instructions for benchmarking a *new *algorithm are given.
- Some instructions for downloading the data are embedded in the code and are unclear. For example, data_source.py states: "see the TableShift instructions for accessing/placing the FICO HELOC dataset..." and heloc.py states: "Data dictionary .xslx file can be accessed after filling out the data agreement at https://community.fico.com/s/explainable-machine-learning-challenge. After reading these instructions, I still don't know what and where to download the data. It'd be preferable to have centralized documentation with clear instructions.

**Relation To Prior Work:**

To my knowledge, prior work is discussed adequately.

**Summary And Contributions:**

TableShift conducts a comprehensive benchmark for domain shift with tabular data. The paper is well-written and easy to follow. The selection of baselines and methods seems appropriate, although I am not an expert in the domain shift literature. Some interesting results show that no method specifically aiming at robustness to domain shift outperforms standard ML pipelines. Unfortunately, aspects of documentation and reproducibility seem incomplete (e.g., I tried to install the provided conda environment on three different systems without success, so I could never reproduce any part of the code). This work can be an impactful paper if the benchmarks and code are made easily reproducible and generalizable to new methods, supporting the development of new methodologies.

---

> ### Author Response · Authors · 2023-08-22
> **Response to Reviewer XmM225**
>
> We are grateful to the reviewer for their thoughtful feedback and for pointing out some issues with usability and reproducibility of the software package, which we have now resolved. We apologize for the technical issues -- these reflect the difficulties of large-scale benchmarking, particularly in the tabular domain where many frameworks are widely used -- and we agree with the reviewer that a major goal of an effective benchmarking toolkit should be to ease those difficulties for researchers.
>
> * **Installing the package**: We made several improvements to ease the overall setup process, including:
>   - Pinned exact Python package versions in a requirements.txt file. In addition to ensuring long-term reoroducibility, the requirements.txt file better supports non-conda environments (such as Python virtual environments).
>   - Created a docker environment to support tableshift. (The dockerfile is also now included in the code, along with a README describing how to run the pre-built image or build it locally.) In fact, we have adjusted our instructions to recommend using the docker environment due to the non-Python dependencies of some of our core dependencies (e.g. LightGBM requires specific C libraries that we cannot ensure proper installation of in Conda).
>   - Verified the environment setup process and running of the sample script on both Linux Ubuntu 22.04 and Mac OS 13, using the docker container. Our git repo now includes CI tests for these environments. We also tested the installation process on Linux and Mac (M1 Max) machines.
>
> * **Instructions for benchmarking new algorithms:** We thank the reviewer for the suggestion to add directions on benchmarking new algorithms, as this is the main intended use of our benchmark. We added a benchmarking guide (benchmarking_guide.md) to the github repo. This contains a comprehensive set of instructions for a user interested in benchmarking a new algorithm.
>
> * **Instructions for downloading data:**
>   - We added instructions for accessing each dataset in the benchmark to our documentation at tableshift.org. Each individual data use agreement is also linked directly in the Github README, but we acknowledge that this may not have been sufficiently user-friendly on its own and are grateful to the reviewer for pointing this out. The complexity and confusion of bringing together a rich and diverse suite of datasets from their original sources is something we seek to alleviate with the benchmark and software!
>   - We also updated the headers of all Python files in the tableshift.datasets module with links to our datasets documentation and a description of the access level of the dataset (public, public credentialized), for benchmark datasets. While we don’t intend for the code to be a source of documentation itself, we hope that now it will direct users to the right information on our website and repo.
>
> * **Paper formatting and figures:**
> We addressed the typos the reviewer flagged in the paper. We note that the marker the reviewer identified in Figure 1 is actually multiple overlapping markers (FT-Transformer, NODE). We also hope that our addition of references to the exact data values in the plots (in each figure’s caption) helps clarify some of the overlapping data points.
> Reproducing our experiments: We added a reproduction guide to the github repo (reproduction_guide.md). This contains step-by-step instructions for launching the script to reproduce our experiments and run e.g. a hyperparameter sweep for a specific model and dataset. We note that the added docker environment should be especially helpful here.
>
> If the reviewer experiences any further issues in setting up the environment and running the code, we would be grateful to know the commands the reviewer executed in order to ensure that we directly address any issues. The command below should allow the reviewer to create and verify the environment and run a test script:
> ```
> # pull the image from github container registry
> $ docker pull ghcr.io/jpgard/tableshift:latest
> # run the image; the default entrypoint will run tableshift/examples/run_expt.py
> $ docker run ghcr.io/mlfoundations/tableshift:latest
> # OPTIONAL: run the image in interactive mode; scripts can also be launched from here
> $ docker run -it --entrypoint=/bin/bash ghcr.io/mlfoundations/tableshift:latest
> ```

---

### Decision · Program_Chairs · 2023-09-22

**Decision:**

Accept (Poster)

**Comment:**

This paper presents a benchmark on tabular data under distribution shifts, encompassing a variety of datasets and algorithms. The documentation is well-structured, facilitating ease of use. After a careful review of both the manuscript and the rebuttal, a major concern is the insufficient analysis and specification of the distribution shift patterns. This omission renders the empirical results and algorithmic comparisons somewhat nebulous. For a benchmark paper of this nature, delineating the specific types of shifts under consideration is pivotal in aiding users to accurately test and evaluate their methods. I would strongly encourage the authors to address this fundamental aspect in future iterations. Furthermore, a deeper level of analysis would significantly enhance the paper’s contribution to the field.